# Methane, carbon dioxide and nitrous oxide emissions from two clear-water and two turbid-water urban ponds in Brussels (Belgium)

Thomas Bauduin [1,2], Nathalie Gypens [1], Alberto V. Borges [2]

[1]Ecology of Aquatic Systems, Université Libre de Bruxelles, Belgium
[2]Chemical Oceanography Unit, University of Liège, Belgium

Correspondence to: Thomas Bauduin (thomas.bauduin@ulb.be)

**Abstract.** Shallow ponds can occur either in a clear-water state dominated by macrophytes or a turbid-water state dominated by phytoplankton, but it is unclear if and how these two states affect the emission to the atmosphere of greenhouse gases (GHGs) such as carbon dioxide ($CO_2$), methane ($CH_4$) and nitrous oxide ($N_2O$). We measured on 46 occasions over 2.5 years (between June 2021 and December 2023) the dissolved concentration of $CO_2$, $CH_4$, and $N_2O$ from which the diffusive air-water fluxes were computed, in four urban ponds in the city of Brussels (Belgium): two clear-water macrophyte-dominated ponds (Silex and Tenreuken), and two turbid-water phytoplankton-dominated ponds (Leybeek and Pêcheries). $CH_4$ ebullitive fluxes were measured with bubble traps in the four ponds during deployments in spring, summer, and fall, totalling 48 days of measurements. To characterize methanogenic pathways (acetoclastic or hydrogenotrophic) and quantify water column methane oxidation (MOX) we measured the $^{13}C/^{12}C$ ratio of $CH_4$ ($\delta^{13}C$-$CH_4$) from gas trapped in the bubble traps, from bubbles deliberately released by the perturbation of the sediments, and in dissolved $CH_4$ in the water column. Measured ancillary variables include water temperature, oxygen saturation level (%$O_2$), concentrations of chlorophyll-$a$ (Chl-$a$), total suspended matter (TSM), soluble reactive phosphorus (SRP), nitrite ($NO_2^-$), nitrate ($NO_3^-$) and ammonium ($NH_4^+$). The turbid-water and clear-water ponds did not differ significantly in terms of diffusive emissions of $CO_2$ and $N_2O$. Clear-water (macrophyte-dominated) ponds exhibited higher values of annual ebullitive $CH_4$ fluxes compared to turbid-water (phytoplankton-dominated) ponds, most probably in relation to the delivery to sediments of organic matter from macrophytes. At seasonal scale, $CH_4$ emissions exhibited a temperature dependence in all four ponds, with ebullitive $CH_4$ fluxes having a stronger dependence to temperature than diffusive $CH_4$ fluxes. The temperature sensitivity of ebullitive $CH_4$ fluxes decreased with increasing water depth. In summer, the $\delta^{13}C$-$CH_4$ values of sediment bubbles indicated that the hydrogenotrophic methanogenesis pathway seemed to dominate in clear-water ponds and acetoclastic methanogenesis pathway seemed to dominate in turbid-water ponds. The $\delta^{13}C$-$CH_4$ values of bubbles traps suggested a seasonal shift from the acetoclastic methanogenesis pathway in spring-summer to the hydrogenotrophic methanogenesis pathway in fall. The $\delta^{13}C$-$CH_4$ of dissolved $CH_4$ indicated higher rates of MOX in turbid-water ponds compared to clear-water ponds, with an overall positive correlation with TSM and Chl-$a$ concentrations. The presence of suspended particles putatively enhanced MOX by reducing light inhibition of MOX and/or by serving as substrate in the water column for fixed methanotrophic bacteria. Total $CH_4$ emissions (diffusive+ebullitive) in $CO_2$ equivalents either equalled or exceeded those of $CO_2$, while $N_2O$ emissions were negligible compared to the other two GHGs. Total annual GHG emissions in $CO_2$ equivalents from all four ponds increased from 2022 to 2023 due to higher $CO_2$ diffusive fluxes, likely driven by higher annual precipitation in 2023 compared to 2022, possibly in response to the intense El Niño event of 2023.

## 1. Introduction

Greenhouse gas (GHG) emissions from inland water (rivers, lakes, and reservoirs) to the atmosphere such as carbon dioxide ($CO_2$), methane ($CH_4$) and nitrous oxide ($N_2O$) are quantitatively important for global budgets (Lauerwald et al., 2023). GHG emissions from lakes are lower than from rivers for $CO_2$ (Raymond et al., 2013) and $N_2O$ (Lauerwald et al., 2019; Maavara et al., 2019). However, emissions of $CH_4$ from lakes (Rosentreter et al., 2021; Johnson et al., 2022) are significant compared to rivers (Stanley et al., 2016; Rocher-Ros et al., 2023). Emissions of $CO_2$ and $CH_4$ from lakes to the atmosphere represent 1.25 to 2.30 Pg $CO_2$ equivalents ($CO_2$-eq) annually with a significant proportion from $CH_4$ emissions, and represent nearly 20% of global $CO_2$ emissions from fossil fuels (Delsontro et al., 2018). The contribution of $CO_2$ and $CH_4$ emissions from small lentic water bodies (small lakes and ponds) can be disproportionately high compared to large systems (Holgerson and Raymond, 2016) as small lakes and ponds are the most abundant of all water body types in number (Verpoorter et al., 2014, Cael et al., 2017), and flux intensities (per $m^2$) are usually higher in smaller water bodies. The emissions of GHGs from artificial water bodies such as agricultural reservoirs, urban ponds, and storm-water retention basins could be higher than those from natural systems (Martinez-Cruz et al., 2017; Grinham et al., 2018; Herrero Ortega et al., 2019; Gorsky et al., 2019; Ollivier et al., 2019; Peacock et al., 2019, 2021; Webb et al., 2019; Bauduin et al., 2024). These higher emissions seem to result from higher external inputs of anthropogenic carbon and nitrogen in artificial systems such as rainfall runoff that brings organic matter and dissolved inorganic nitrogen (DIN), but might also reflect other differences compared to natural systems such as in hydrology (Clifford and Heffernan, 2018). Among artificial systems, urban ponds are the subject of a growing body of literature (Singh et al., 2000; Natchimuthu et al., 2014; van Bergen et al., 2019; Audet et al., 2020; Peacock et al., 2021; Goeckner et al., 2022; Ray and Holgerson, 2023; Bauduin et al., 2024). Urban areas can have numerous small artificial water bodies mostly associated to green spaces such as public parks, and their number is increasing due to rapid urbanisation worldwide (Brans et al., 2018; Audet et al., 2020; Gorsky et al., 2024; Rabaey et al., 2024). Urban ponds are generally small, shallow, and usually their catchment consists in majority of impervious surfaces with a smaller contribution from soils (Davidson et al., 2015; Peacock et al., 2021).

In shallow ponds and lakes, including urban ponds, aquatic primary production is either dominated by submerged macrophytes or by phytoplankton, corresponding to two alternate states (Scheffer et al., 1993). These two alternative states correspond to clear waters (macrophyte-dominated) or turbid waters (phytoplankton-dominated), during the productive period of the year (spring and summer in mid-latitudes). Submerged macrophytes and phytoplankton regulate $CO_2$ dynamic directly through photosynthesis that can be more or less balanced by community respiration in the water column. However, it is not clear whether the presence of macrophytes increases or decreases the $CO_2$ emissions from ponds and lakes. Some studies have shown a decrease of $CO_2$ emissions with increasing macrophyte density (Kosten et al., 2010; Ojala et al., 2011; Davidson et al., 2015), but other studies showed the opposite pattern (Theus et al., 2023). In phytoplankton-dominated lakes, $CO_2$ concentrations depend in part on the development stage of the phytoplankton, with the growth and peak phases generally coinciding with lower $CO_2$ concentrations due to intense photosynthesis (Grasset et al., 2020; Vachon et al., 2020). $CH_4$ emissions have been reported to increase with the concentration of chlorophyll-*a* (Chl-*a*) in phytoplankton-dominated lakes (DelSontro et al., 2018; Borges et al., 2022). The presence of macrophytes strongly affects $CH_4$ cycling in freshwaters (Bastviken et al., 2023) and vegetated littoral zones of lakes exhibit higher $CH_4$ emissions than non-vegetated zones (Desrosiers et al., 2022; Theus et al., 2023). Macrophytes influence organic matter decomposition processes in sediments depending on the quality and quantity of plant matter they release into their environment (Reitsema et al., 2018; Grasset et al., 2019; Harpenslager et al., 2022; Theus et al., 2023). Yet, few studies have consistently compared $CH_4$ emissions in clear-water and turbid-water ponds (Hilt et al., 2017). A study in Argentina reported higher dissolved $CH_4$ concentrations in clear-

water ponds with submerged macrophytes compared to turbid-water phytoplankton-dominated ponds, but no differences in
measured $CH_4$ emissions (Baliña et al., 2023). The production of $N_2O$ predominantly occurs through microbial nitrification
and denitrification that depend on DIN and $O_2$ levels (Codispoti and Christensen, 1985; Mengis et al., 1997). Competition
for DIN between primary producers and $N_2O$-producing microorganisms can impact $N_2O$ production. Additionally, the
transfer of labile phytoplankton organic matter to sediments fuels benthic denitrification. Combined, these two processes
could explain that some lakes can act as sinks of $N_2O$ under elevated Chl-$a$ concentrations (Webb et al., 2019; Borges et al.,
2022).The presence of macrophytes also strongly influences nitrogen cycling in sediments of lakes and ponds (Barko et al.,
1991; Choudhury et al., 2018; Deng et al., 2020; Dan et al., 2021) and should in theory also affect $N_2O$ emissions, although
seldom investigated, and available studies provide contradictory conclusions. $N_2O$ emissions has been showed to follow
diurnal cycles of $O_2$ concentrations in areas dominated by submerged macrophytes in Lake Wuliangsuhai (China) (Ni et al.,
2022) and the seasonal cycle of aboveground biomass of emerged macrophytes (*Phragmites*) in Baiyangdian Lake (China)
(Yang et al., 2012). On the contrary, some studies showed there were no significant differences of $N_2O$ production in
sediments of macrophyte-rich (n=10) and macrophyte-free (n=12) lakes in subtropical China (Liu et al., 2018). There have
been a very limited number of studies investigating systematically how emissions differ between ponds dominated by
phytoplankton and those dominated by macrophytes (Harpenslager et al., 2022; Baliña et al., 2023), and none investigating
simultaneously $CO_2$, $CH_4$, and $N_2O$ emissions including both diffusive and ebullitive components.
The emissions from aquatic systems of $CO_2$ and $N_2O$ are exclusively through diffusion across the air-water interface
(diffusive flux), while $CH_4$ can be additionally emitted as bubbles released from sediments to the atmosphere (ebullitive
flux). At annual scale, ebullitive $CH_4$ flux usually represents more than half of total (diffusive+ebullitive) $CH_4$ emissions
from shallow lakes (Wik et al., 2013; Deemer and Holgerson, 2021), although the relative contribution of ebullitive and
diffusive $CH_4$ emissions is highly variable seasonally (*e.g.* Wik et al., 2023; Ray and Holgerson, 2023). Ebullitive $CH_4$
fluxes are particularly high in the littoral zone of lakes at depths <5 m (Wik et al., 2013; DelSontro et al., 2016; Borges et al.,
2022) and strongly increase in response to temperature (DelSontro et al., 2016; Aben et al., 2017), as well as organic matter
availability (DelSontro et al., 2016; 2018). Ebullitive $CH_4$ fluxes tend to be higher in small and shallow water bodies
(Deemer and Holgerson, 2021) but are notoriously variable in time and space, and are difficult to estimate reliably
(DelSontro et al., 2011).
The two primary metabolic pathways for $CH_4$ production in sediments by methanogenic archaea are the fermentation of
acetate (acetoclastic pathway) and the reduction of carbon dioxide by $H_2$ (hydrogenotrophic pathway) (Whiticar et al., 1986;
Conrad, 1989). $CH_4$ produced by these two pathways exhibits distinct $^{13}C/^{12}C$ ratios ($\delta^{13}C$-$CH_4$) (Whiticar et al., 1986) and
can be used to discriminate which pathway is dominant. When $CH_4$ diffuses from sediments to the water column, it can be
oxidized by methanotrophic bacteria who preferentially consume $CH_4$ with $^{12}C$ over $^{13}C$, resulting in an increase of $\delta^{13}C$-$CH_4$
of the residual $CH_4$ in the water column (Bastviken et al., 2002). Fractionation models then allow estimating methane
oxidation (MOX) from measurements of $\delta^{13}C$-$CH_4$ of dissolved $CH_4$ in the water column. Bastviken et al. (2008) report that
30 to 99% of the $CH_4$ produced in sediments of freshwater lakes can be removed by MOX that is as a significant $CH_4$ sink in
these water bodies. MOX is known to be inhibited by light (Dumestre et al., 1998) and increases with the presence
suspended particles (Abril et al. 2007) so that MOX might vary between clear and turbid waters (Morana et al. 2020).
Here, we report a dataset of $CO_2$, $CH_4$, and $N_2O$ dissolved concentrations in four shallow and small urban ponds (Leybeek,
Pêcheries, Silex, and Tenreuken) in the city of Brussels (Belgium) (Fig. 1), with data collected 46 times at regular intervals
(between June 2021 and December 2023) on each pond. The air-water diffusive fluxes of $CO_2$, $CH_4$, and $N_2O$ were
calculated from dissolved concentrations and the gas transfer velocity, while the ebullitive $CH_4$ fluxes were measured with
inverted funnels during 8 deployments (totalling 48 days) in the four ponds. The $\delta^{13}C$-$CH_4$ in the sedimentary bubbles and in
the water provides additional information on $CH_4$ dynamics such as the methanogenesis pathway (acetoclastic or
hydrogenotrophic) and MOX. We test the hypothesis that the two alternative states in shallow lakes (a clear-water state
dominated by macrophytes, or a turbid-water state dominated by phytoplankton) drive differences in the $CO_2$, $CH_4$, and $N_2O$
dissolved concentration and diffusive emissions from the four studied artificial ponds, that have similar depth, surface area,
and catchment urban coverage, and that mainly differ by the phytoplankton-macrophyte dominance. We also test the
hypothesis that the two alternative states in shallow lakes drive differences in the ebullitive $CH_4$ emissions, water column
MOX, and sedimentary methanogenesis pathway (acetoclastic or hydrogenotrophic) in the four studied ponds. The final
objective of the present work is to determine the relative contribution of $CO_2$, $CH_4$, and $N_2O$ to the total GHG emissions in
$CO_2$-eq and to test the hypothesis that the relative contribution of each GHG differs according to the two alternative states in
shallow lakes.

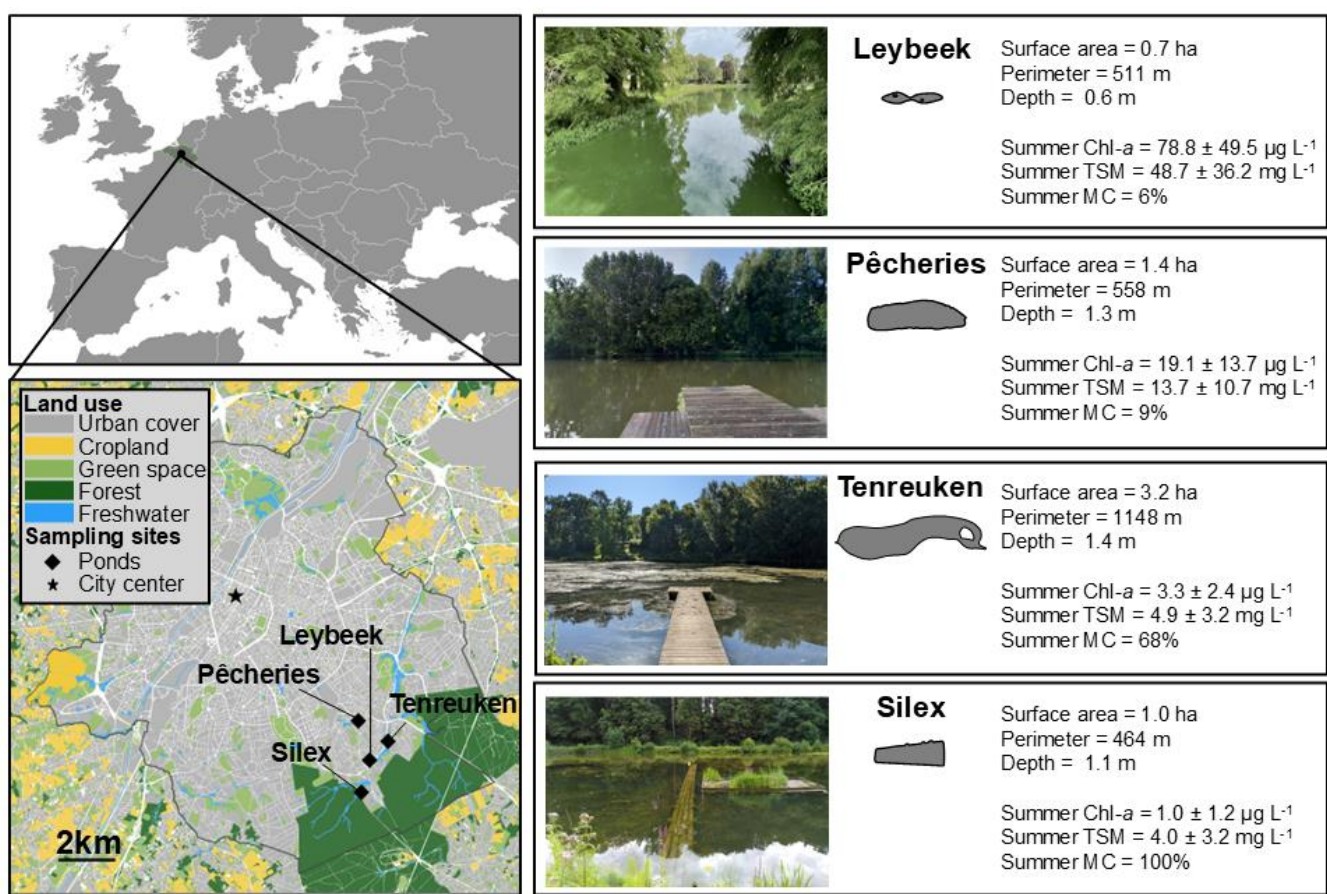

**Figure 1: Location of the four sampled ponds in Brussels (Belgium, Europe). Bottom left map shows the metropolitan area of the**
**region of Brussels delineated by the black line and the surrounding region of Flanders in Belgium, showing land cover and**
**sampled urban ponds (black diamonds). The star corresponds to the center of the city (50.8504°N, 4.3487°E). Additional**
**information for each pond is indicated on right panels: shapes of the ponds, surface area (ha), perimeter (m), average depth (m),**
**mean±standard deviation of summer chlorophyll-*a* (Chl-*a*, in μg $L^{-1}$) and summer total suspended matter (TSM, in mg $L^{-1}$) from**
**21 June to 21 September in 2021, 2022, 2023, and summer total macrophyte cover (MC, in %) (Table S1).**

## 2. Material and Methods

### 2.1. Field sampling and meteorological data

Sampling was carried out from a pontoon in the four ponds on the same day between 9am and 11am, 46 times on each pond between June 2021 and December 2023 at a frequency ranging from one (winter) to three (summer) times per month at a single fixed station in each of the four ponds. Water was sampled 5cm below the surface with 60ml polypropylene syringes for analysis of dissolved concentrations of $CO_2$, $CH_4$, and $N_2O$. Samples for $CH_4$ and $N_2O$ were transferred from the syringes with a silicone tube into 60 ml borosilicate serum bottles (Weathon), preserved with 200 µl of a saturated solution of $HgCl_2$, sealed with a butyl stopper and crimped with aluminium cap, without a headspace, samples were stored at ambient temperature protected from direct light prior to analysis in laboratory. The partial pressure of $CO_2$ ($pCO_2$) was measured directly in the field, within 5 minutes of sample collection, with a Li-Cor Li-840 infrared gas analyser (IRGA) based on the headspace technique with 4 polypropylene syringes (Borges et al., 2019). A volume of 30 ml of sample water was equilibrated with 30 ml of atmospheric air within the syringe by shaking vigorously for 5 minutes. The headspace of each syringe was then sequentially injected into the IRGA and a fifth syringe was used to measure atmospheric $CO_2$. The final $pCO_2$ value was computed taking into account the partitioning of $CO_2$ between water and the headspace, as well as equilibrium with $HCO_3^-$ (Dickson et al., 2007) using water temperature measured in-situ and after equilibration, and total alkalinity (data not shown). Samples for total alkalinity were conditioned, stored and analysed as described by Borges et al. (2019). The IRGA was calibrated in the laboratory with ultrapure $N_2$ and a suite of gas standards (Air Liquide Belgium) with $CO_2$ mixing ratios of 388, 813, 3788 and 8300 ppm. The precision of $pCO_2$ measurements was ±2.0%. Water temperature, specific conductivity, and oxygen saturation level (%$O_2$) were measured in-situ with VWR MU 6100H probe 5cm below the surface. A 2 liter polyethylene water container was filled with surface water for conditioning the samples for other variables at the laboratory in Université Libre de Bruxelles.

Surveys to identify and quantify visually the relative coverage of emerged and submerged macrophytes were conducted in summer 2023 (Table S1). The resulting list of macrophyte species agreed with past studies in Brussels ponds (Peretyatko et al., 2009).

Three bubble traps were deployed 50 cm apart for measuring ebullitive $CH_4$ flux. The bubble traps consisted of inverted polypropylene funnels (diameter 23.5 cm) mounted with 60 ml polypropylene syringes, with three way stop valves allowing to collect the gas without contamination from ambient air. The polypropylene funnel was attached with steel rods to a polystyrene float. The volume of gas collected in the funnels was sampled with graduated polypropylene 60 ml syringes every 24 hours. The value of the collected volume of gas was logged, and the gas was transferred immediately after collection to pre-evacuated 12 ml vials (Exetainers, Labco, UK) that were stored at ambient temperature protected from direct light prior to the analysis of $CH_4$ concentration and $\delta^{13}C$-$CH_4$ in the laboratory. The time-series of measurement were longer at the Silex pond than the other three ponds, because the Silex pond is closed to the public during the week, while the other three ponds are open to the public all the time.

In summer 2023, the bubbles present in the sediment were directly collected with bubble traps by physically perturbing the sediment below the traps with a wooden rod. The gas collected in the funnels was stored in pre-evacuated 12 ml vials (Exetainers, Labco, UK) that were stored at ambient temperature protected from direct light prior to the analysis of $\delta^{13}C$-$CH_4$ in the laboratory. These samples are referred hereafter to as from "perturbed sediments." The samples collected in the bubble traps during the ebullition measurements are referred to as from "trapped bubbles."

Air temperature, precipitation, wind speed, and atmospheric pressure, were retrieved from https://wow.meteo.be/en for the
meteorological station of the Royal Meteorological Institute of St-Lambert (50.8408°N, 4.4234°E) in Brussels, located
between 2.5 and 5.0 km from the surveyed ponds. Air temperature, wind speed and atmospheric pressure were averaged over
24 h to obtain a daily mean value. Precipitation was integrated each day to obtain cumulated daily rainfall.

### 2.2. Laboratory analysis

#### 2.2.1. Chlorophyll-*a*, total suspended matter, and dissolved inorganic nutrients

Water was filtered through Whatman GF/F glass microfiber filters (porosity 0.7 μm) with a diameter of 47 mm for total
suspended matter (TSM) and Chl-*a*. Filters for TSM were dried in an oven at 50 °C and filters for Chl-*a* were kept frozen (-
20 °C). The weight of each filter was determined before and after filtration of a known volume of water using an Explorer™
Pro EP214C analytical microbalance (accuracy ±0.1 mg) for determination of TSM concentration. Chl-*a* concentration was
measured on extracts with 90% acetone by fluorimetry (Kontron model SFM 25) (Yentsch and Menzel, 1963) with a limit of
detection of 0.01 μg $L^{-1}$. Filtered water was stored frozen (-20 °C) in 50 ml polypropylene bottles for analysis of dissolved
nutrients. Soluble reactive phosphorus (SRP) was determined by the ammonium molybdate, ascorbic acid and potassium
antimony tartrate staining method (Koroleff, 1983), with a limit of detection of 0.1 μmol $L^{-1}$. Ammonium ($NH_4^+$) was
determined by the nitroprusside-hypochlorite-phenol staining method (Grasshoff and Johannsen, 1972), with a limit of
detection of 0.05 μmol $L^{-1}$. Nitrite ($NO_2^-$) and nitrate ($NO_3^-$) were determined before and after reduction of $NO_3^-$ to $NO_2^-$ by a
cadmium-copper column, using the Griess acid reagent staining method (Grasshoff and Kremling, 2009), with a detection
limit of 0.01 and 0.1 μmol $L^{-1}$, respectively. Concentration of dissolved inorganic nitrogen (DIN) was calculated as the sum
$NH_4^+$, $NO_2^-$ and $NO_3^-$ concentrations in μmol $L^{-1}$.

#### 2.2.2. $CH_4$ and $N_2O$ measurements by gas chromatography and $\delta^{13}C$-$CH_4$ by cavity ring-down spectrometry

Measurements of $N_2O$ and $CH_4$ concentrations dissolved in water and in the gas samples from bubbles were made with the
headspace technique (Weiss 1981) with an headspace volume of 20ml of ultra-pure $N_2$ (Air Liquid Belgium) and a gas
chromatograph (GC) (SRI 8610C) with a flame ionisation detector for $CH_4$ and an electron capture detector for $N_2O$
calibrated with $CH_4$:$N_2O$:$N_2$ gas mixtures (Air Liquide Belgium) with mixing ratios of 1, 10 and 30 ppm for $CH_4$,  and 0.2,
2.0 and 6.0 ppm for $N_2O$. The precision of measurement based on duplicate samples was ±3.9% for $CH_4$ and ±3.2% for $N_2O$.
The $CO_2$ concentration is expressed as partial pressure ($pCO_2$) in parts per million (ppm) and $CH_4$ as dissolved concentration
(nmol $L^{-1}$), as frequently used in topical literature. $CH_4$ concentration were systematically and distinctly above saturation
level (2-3 nmol $L^{-1}$) and $pCO_2$ values were only five times below saturation out of the 187 measurements. The $N_2O$
concentrations fluctuated around atmospheric equilibrium, so data are presented as percent of saturation level (%$N_2O$, where
atmospheric equilibrium corresponds to 100%). The equilibrium with atmosphere for $N_2O$ was calculated from the average
air mixing ratios of $N_2O$ provided by the Global Monitoring Division (GMD) of the National Oceanic and Atmospheric
Administration (NOAA) Earth System Research Laboratory (ESRL) (Dutton and Hall, 2023), and using the Henry's
constant given by Weiss and Price (1980).
The $\delta^{13}C$-$CH_4$ was measured in the gas of the headspace (20 ml of synthetic air, Air Liquid Belgium) equilibrated with the
water sample (total volume 60 ml) for water samples and directly in the gas stored in Exetainers for samples from the bubble
traps. The gas samples were diluted to obtain a final partial pressure of $CH_4$ in the cavity below 10 ppm (target value of 6
ppm) to fall within the operational concentration range of the instrument recommended by the manufacturer, prior to
injection into a cavity ring-down spectrometer (G2201i, Isotopic Analyzer, Picarro) with a Small Sample Introduction
Module 2 (SSIM, Picarro). Data were corrected with curves of $\delta^{13}$C-CH$_4$ as a function of concentration based on two gas
standards from Airgas Specialty Gases with certified $\delta^{13}$C-CH$_4$ values of -23.9±0.3 ‰ and -69.0±0.3 ‰.

### 2.3. Calculations

#### 2.3.1. Diffusive GHG emissions

The diffusive air-water $CO_2$, $CH_4$, or $N_2O$ fluxes ($F_G$) were computed according to:
$$F_G = k\Delta[G] \,,\tag{1}$$
where $k$ is the gas transfer velocity and $\Delta[G]$ is the air-water gas concentration gradient.
The atmospheric $pCO_2$ was measured in the field with the Li-Cor Li-840. For $CH_4$, the global average present day
atmospheric mixing ratio of 1.9 ppm was used (Lan et al., 2024). Atmospheric $N_2O$ concentration was calculated from the
average air mixing ratios of $N_2O$ provided by the GMD of the NOAA ESRL (Dutton et al., 2017). $k$ was computed from a
value normalized to a Schmidt number of 600 ($k_{600}$) and from the Schmidt number of $CO_2$, $CH_4$ and $N_2O$ in freshwater
according to the algorithms as function of water temperature given by Wanninkhof (1992). $k_{600}$ was calculated from the
parameterization as a function of wind speed of Cole and Caraco (1998). $CH_4$ and $N_2O$ emissions were converted into $CO_2$
equivalents ($CO_2$-eq) considering a 100-year timeframe, using global warming potential of 32 and 298 for $CH_4$ and $N_2O$,
respectively (Myrhe et al., 2013).

#### 2.3.2. Ebullitive flux

Bubble flux (ml m$^{-2}$ d$^{-1}$) measured with the inverted funnels was calculated according to:
$$F_{bubble} = \frac{V_g}{A \times \Delta t} \,,\tag{2}$$
where $V_g$ is the volume of gas collected in the funnels (ml), $A$ is the cross-sectional area of the funnel (m$^2$), $\Delta t$ is the
collection time (d).
A multiple linear regression model of $F_{bubble}$ dependent on water temperature and drops of atmospheric pressure was fitted to
the data according to:
$$\log_{10}(F_{bubble}) = \alpha \times T_w + \beta \times \Delta p \,,\tag{3}$$
where $\alpha$ and $\beta$ are the slope coefficients of the multiple linear regression model, $T_w$ is the water temperature (°C), and $\Delta p$
quantifies the drops in atmospheric pressure (atm), calculated according to Zhao et al. (2017):
$$\Delta p = -\frac{1}{\Delta t}\int_0^t p - p_0 \;\; ; \quad \forall \; p < p_0 \,,\tag{4}$$
where $p$ is the atmospheric pressure (atm), $p_0$ a threshold pressure fixed at 1 atm and $\Delta t$ the time interval between two
measurements (d) (Fig. S1).
Ebullitive $CH_4$ fluxes (mmol m$^{-2}$ d$^{-1}$) were calculated according to:
$$E_{CH4} = [CH_4] \times F_{bubble} \,,\tag{5}$$
where $[CH_4]$ is the $CH_4$ concentration in bubbles (mmol ml$^{-1}$).
The methane ebullition $Q_{10}$ represents the proportional change in the ebullitive $CH_4$ flux per 10°C change in water
temperature (DelSontro et al., 2016) and was computed according to:
$$Q_{10} = 10^{10b} ,$$ (6)
where b is the slope of the linear regression between the logarithm of the ebullitive $CH_4$ flux ($E_{CH4}$) and $T_w$, and c is the y-
intercept, according to:
$$\log_{10}(E_{CH4}) = b \times T_w + c ,$$ (7)
The flux of $CH_4$ from dissolution of rising bubbles was computed using the model of McGinnis et al. (2006) implemented in
the SiBu-GUI graphical user interface (Greinert and McGinnis, 2009).

### 2.3.3. Methane oxidation

The fraction of $CH_4$ oxidized (FOX) was calculated with a closed-system Rayleigh fractionation model (Liptay et al., 1998)
according to:
$$\ln(1 - \text{FOX}) = \frac{\ln(\delta^{13}C\text{-}CH_{4\_initial} + 1000) - \ln(\delta^{13}C\text{-}CH_4 + 1000)}{\alpha - 1} ,$$ (8)
where $\delta^{13}C$-$CH_{4\_initial}$ is the $^{13}C/^{12}C$ ratio of dissolved $CH_4$ as produced by methanogenesis in sediments, $\delta^{13}C$-$CH_4$ is the
$^{13}C/^{12}C$ ratio of dissolved $CH_4$ in-situ, and $\alpha$ is the fractionation factor.
We used a value of 1.02 for $\alpha$ based on laboratory culture experiments carried out at 26°C (Coleman et al., 1981) and field
measurements in three Swedish lakes (Bastviken et al., 2002) and one tropical lake (Morana et al., 2015). The $\alpha$ values
gathered in the three Swedish lakes were independent of season and temperature according to Bastviken et al. (2002).
For $\delta^{13}C$-$CH_{4\_initial}$, we used a value of -69‰ for spring and summer, and -83‰ for fall based on average of measured $\delta^{13}C$-
$CH_4$ in trapped bubbles (see Sect. 3.5). For winter we used a value of -76‰ corresponding to the average of the fall and
spring/summer values.
MOX was computed from $FOX$ and the $F_G$ of $CH_4$ ($F_{CH4}$) according to (Bastviken et al., 2002):
$$\text{MOX} = F_{CH_4} \times \frac{\text{FOX}}{1 - \text{FOX}} ,$$ (9)

### 2.4. Statistical analysis

Statistical analysis was conducted with R version 4.4.1. Pearson's linear correlation coefficients and the r² coefficient were
used to assess relationships between log-transformed variables within each pond and across the dataset, to identify potential
pond-specific and overall direct relationships between variables and GHGs. Statistical significance was determined using
Fisher's F test and the associated $p$-value. This approach was also applied to study the relationships between $\delta^{13}C$-$CH_4$, FOX
and MOX with Chl-$a$ and TSM. To assess the impact of Chl-$a$ concentration, macrophyte cover in summer, water depth, and
lake surface area on diffusive and ebullitive $CH_4$ fluxes, the ratio of ebullitive $CH_4$ to total $CH_4$ flux, and $CO_2$ and $N_2O$
fluxes, both linear and quadratic relationships were applied to log-transformed averaged data. This approach allowed for the
observation of trends between explanatory and dependent variables. For N₂O fluxes, additional explanatory variables
included $NO_2^-$, $NO_3^-$, $NH_4^+$, and DIN concentrations.
A two-way repeated measures analysis of variance (ANOVA) was used to test for differences in categorical variables, with
the four seasons and the four ponds serving as independent factors, pond was set as a random effect to account for repeated
measurements. A one-way repeated measures ANOVA was used to test for differences in $\delta^{13}C\text{-}CH_4$ from "perturbed
sediments" with the four ponds serving as independent factors. After conducting an ANOVA and establishing significant
differences among at least two groups ($p<0.05$), Tukey's Honestly Significant Difference (HSD) post-hoc test was employed
to perform pairwise comparisons across all groups. Statistical outcomes are visually represented on boxplots, where upper-
and lower-case letters are used to denote significant differences ($p<0.05$). Different lower- and upper-case letters indicate
significant differences between groups.
## 3. Results and discussion
### 3.1. Seasonal variations of meteorological conditions and GHG concentrations
Belgium has a west coast marine climate with mild weather year-round, and evenly distributed abundant rainfall totalling on
average 837 mm annually for the reference period 1991-2020. The average annual air temperature was 11°C, with summer
average of 17.9 °C and winter average of 4.1 °C for the reference period 1991-2020. During the sampling period, from June
2021 to December 2023, water temperature in the surface of the four sampled ponds (Leybeek, Pêcheries, Silex, and
Tenreuken; Fig. 1) tracked closely the air temperature that ranged between -1.5 and 30.0°C following the typical seasonal
cycle at mid-latitudes in the Northern Hemisphere (Fig. S2). Years 2022 and 2023 were about 1 °C warmer than the average
for the period 1991-2020 (11 °C), while year 2021 was closer to the long-term average (Fig. 2). Year 2022 was warmer and
drier than 2021 and 2023 (Fig. 2), with positive temperature anomalies observed evenly throughout the year (9 months out of
12) and negative precipitation anomalies in summer, fall and early winter (Fig. S2). Year 2021 had warmer and drier months
in June and September, colder and wetter months in July and August, and was overall wetter and colder than 2022 (Fig. 2).
Year 2023 was marked by both positive temperature and precipitation anomalies (Fig. S2), resulting in a wetter and warmer
year than normal and compared to 2021 and 2022. (Fig. 2). Daily wind speed was generally low (<1 m s⁻¹) except for a
windier period in spring 2022 (up to 5.8 m s⁻¹, corresponding to the Eunice storm) and in fall 2023 (up to 9.7 m s⁻¹,
corresponding to the Ciarán storm) (Fig. S2).

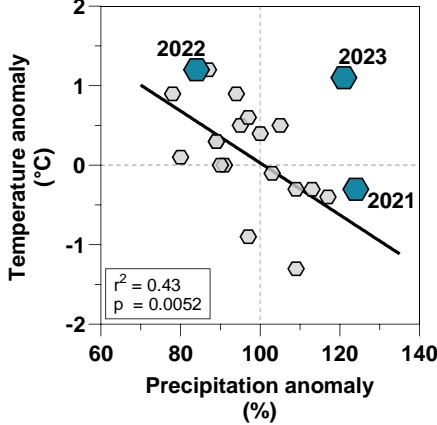


**Figure 2: Temperature anomaly (difference between the average annual temperature and the normal annual temperature for the**
**reference period 1991-2020 (11 °C), in °C) plotted against precipitation anomaly (ratio between annual precipitation and normal**

annual precipitation for the reference period 1991-2020 (837 mm), in %) from 2003 to 2023. Each small grey hexagon represents values for years from 2003 to 2020 and larger blue hexagons represent the years of sampling from this study (2021, 2022 and 2023). Linear regression for years 2003-2020 is shown by a black solid line ($Y = 3.29 - 0.03 \cdot X$, n=20, Table S11). Note the anomalous rainy year in 2023 relative to the pattern as function of temperature for the other years, possibly in response to the strong El Niño event of 2023 (Chen et al., 2024).

The four sampled ponds are situated in the periphery of the city of Brussels, with the Silex pond bordered by the Sonian Forest (Fig. 1). The four ponds are relatively small (0.7-3.2 ha) and shallow (60-140 cm) and have not been drained or dredged since at least 2018 (Table S2). The four studied ponds had significantly different Chl-*a* concentration values during summer, with the Leybeek pond having higher Chl-*a* (78.8±49.5 µg L$^{-1}$), followed by the Pêcheries pond (19.1±13.7 µg L$^{-1}$), the Tenreuken pond (3.3±2.4 µg L$^{-1}$), and the Silex pond (1.0±1.2 µg L$^{-1}$) (Tukey's HSD test p ≤0.0001 for each pair of comparisons, Figs. 1, 3). The Leybeek and Pêcheries ponds with higher summer Chl-*a* concentration had turbid-water (summer TSM = 48.7±36.2 and 13.7±10.7 mg L$^{-1}$, respectively), and undetectable submerged macrophyte cover in summer (Fig. 1, Table S1). The Tenreuken and Silex ponds with lower summer Chl-*a* concentrations had clear-water (summer TSM = 4.9±3.2 and 4.0±3.2 mg L$^{-1}$, respectively), and a high total macrophyte cover during summer (68 and 100%, respectively, Fig. 1, Table S1). Values of Chl-*a* were higher in summer than in winter in the turbid-water Leybeek and Pêcheries ponds (Tukey's HSD test p=0.0107 for the Leybeek pond, p=0.0211 for the Pêcheries pond) related to summer algal blooms. Values of Chl-*a* were higher in winter than in summer in the clear-water Tenreuken and Silex ponds (Tukey's HSD test=0.0296 for the Tenreuken pond, p=0.0056 for the Silex pond), probably related to competition for inorganic nutrients from macrophytes, with the Silex pond showing lower summer Chl-*a* (Tukey's HSD test p<0.0001), lower summer TSM concentrations (Tukey's HSD test p<0.0001) and higher summer total macrophyte cover compared to the Tenreuken pond (Fig. 1).

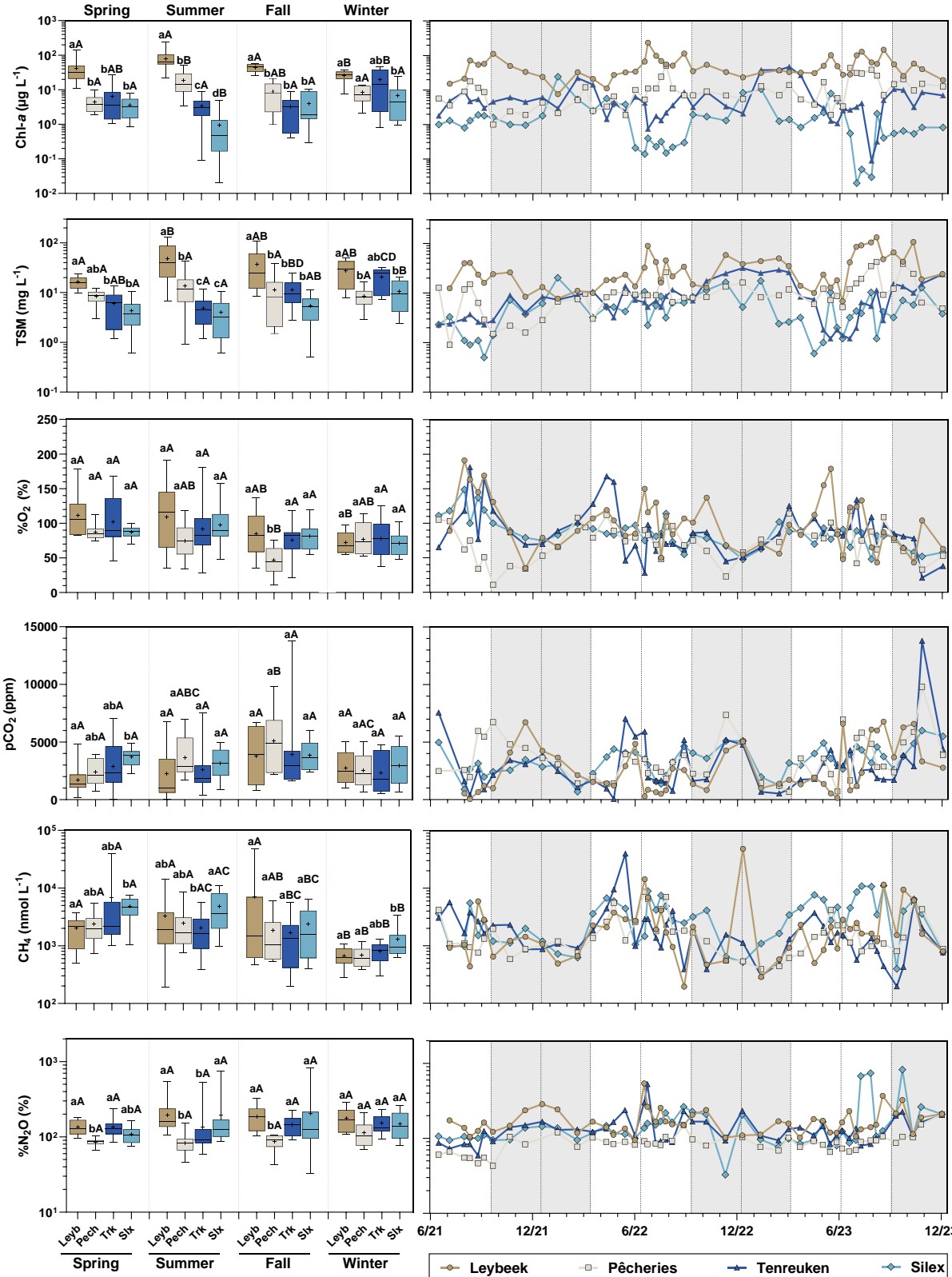

320

**Figure 3: Seasonal variations of Chlorophyll-*a* (Chl-*a*, in µg L$^{-1}$), total suspended matter (TSM, in mg L$^{-1}$), oxygen saturation (%O$_2$, in %), partial pressure of CO$_2$ (pCO$_2$ in ppm), dissolved CH$_4$ concentration (CH$_4$, in nmol L$^{-1}$), and N$_2$O saturation level (%N$_2$O, in %) in four urban ponds (Leybeek (Leyb), Pêcheries (Pech), Tenreuken (Trk), and Silex (Slx)) in the city of Brussels (Belgium) from June 2021 to December 2023. Box plots show median (horizontal line), mean (cross), and 25–75% percentiles (box limits). Whiskers extend from minimum to maximum values. White and grey bands in the graphs on the right correspond to the autumn/winter and spring/summer periods, respectively, and dotted vertical bars represent the first days of each season. ANOVA results of the multiple comparison between boxplots are summarized in Tables S4 and S5. Different lower-case letters indicate significant differences between ponds within a season and different upper-case letters indicate significant differences between seasons for a given pond.**

The $\%O_2$ values ranged from 11 to 191% (Fig. 3). The highest $\%O_2$ values in the four ponds were observed in spring and
summer compared to fall and winter owing to aquatic primary production. In summer, $\%O_2$ was statically higher in the
Leybeek pond (109±46 %) characterized by higher Chl-*a* concentration compared to the Pêcheries pond (75±23 %) (Tukey's
HSD test p=0.0037). The lowest average $\%O_2$ was observed in fall in the Pêcheries pond (46±22 %) and was statistically
lower than in the Leybeek (85±34%, Tukey's HSD test p=0.0302), Tenreuken (76±26 %, Tukey's HSD test p=0.0488), and
Silex ponds (81±19 %, Tukey's HSD test p=0.0132).
The $pCO_2$ values ranged from 40 to 13,804 ppm (Fig. 3), within the range of values typically observed in ponds (Holgerson
and Raymond, 2016; Peacock et al., 2019; Audet et al., 2020). Undersaturation of $CO_2$ with respect to atmospheric
equilibrium was only observed on five occasions out of the 187 measurements, three times in the turbid-water Leybeek pond
in summer (40 ppm on 13 August 2021, 220 ppm on 27 June 2022 and 149 ppm on 13 June 2023), and twice in the clear-
water Tenreuken pond in spring and summer (383 ppm on 13 August 2021 and 55 ppm on 2 May 2022). Low values of
$pCO_2$ were generally observed in spring and summer probably due to uptake of $CO_2$ by primary production from either
phytoplankton or submerged macrophytes. High values of $pCO_2$ were observed in fall in the four ponds and probably reflect
the release of $CO_2$ from degradation of organic matter due to the senescence of phytoplankton or macrophytes (Fig. 3). A
general control of $pCO_2$ by biological activity (primary production and respiration) was confirmed by the strong negative
correlation with $\%O_2$ observed in all four ponds (*e.g.* Holgerson, 2015), as well as a positive correlation with DIN observed
in three ponds, and a positive correlation with SRP observed in the two clear-water ponds (Table S3; Figs S3, S4, S5, S6). A
negative correlation between $pCO_2$ and Chl-*a* was only observed in the turbid-water Leybeek pond (Table S3; Fig S5),
which showed the highest average Chl-*a* concentration, and no correlation was found in clear-water ponds, where aquatic
primary production was presumably mainly related to submerged macrophytes (Table S3; Figs S3, S4). In all four ponds,
$pCO_2$ strongly correlated positively to precipitation (Table S3; Figs S3, S4, S5, S6) suggesting a control of external inputs of
carbon either as organic carbon sustaining internal degradation of organic matter or as soil $CO_2$ (*e.g.* Marotta et al., 2011).
The $CH_4$ dissolved concentrations ranged from 194 to 48,380 nmol $L^{-1}$ (Fig. 3), within the range of values typically observed
in ponds (Holgerson and Raymond, 2016; Peacock et al., 2019; Audet et al., 2020). Dissolved $CH_4$ concentration was
positively correlated to water temperature in all four ponds (Table S3; Figs S3,S4,S5,S6), most probably reflecting the
increase of sedimentary methanogenesis with temperature (Schulz and Conrad, 1996), with higher dissolved $CH_4$
concentrations observed in spring (3160±5989 nmol $L^{-1}$) and summer (3979±2993 nmol $L^{-1}$) than in fall (2645±7315 nmol $L^{-1}$
) and winter (868±601 nmol $L^{-1}$) (Tukey's HSD test: spring versus fall, p=0.0954; spring versus winter, p<0.0001; summer
versus fall, p=0.0154; summer versus winter, p<0.0001). In individual ponds, dissolved $CH_4$ concentration was negatively
correlated to precipitation and DIN in the Pêcheries pond (Table S3; Fig S6), and positively correlated to SRP in the Silex
pond (Table S3; Fig S4). These relationships between $CH_4$ and other variables probably indirectly reflect the seasonal
variations of these other variables that showed correlations with temperature, as DIN was negatively correlated to
temperature in the Pêcheries pond ($r^2$=0.11, p=0.0028), and SRP was positively correlated to temperature in the Silex pond
($r^2$= 0.10, p=0.0103). Dissolved $CH_4$ concentration was negatively correlated to Chl-*a* in the Silex pond (Table S3; Fig S4)
and to TSM in the Tenreuken pond (Table S3; Fig S3). These relationships probably reflect the negative relationship
between Chl-*a* and temperature in the Silex pond ($r^2$=0.13, p=0.0008) and the negative relationship between TSM and
temperature in the Tenreuken pond ($r^2$=0.36, p<0.0001) because of the primary production from macrophytes peaks in
summer in the two clear-water ponds.

The correlations between $pCO_2$ and precipitation, and between dissolved $CH_4$ concentration and temperature observed in all four ponds individually were also observed when pooling together the data for all four ponds ("All" in Table S3; Fig S7). The slopes of these correlations were not significantly different between ponds and were not correlated with surface area, depth, or dominance of type of primary producers (phytoplankton or macrophyte) (Table S6). These results suggest that the effect of precipitation on $pCO_2$ and the impact of temperature on dissolved $CH_4$ concentration outweigh other factors in explaining seasonal variations.

The $\%N_2O$ values ranged from 32 to 826% (Fig. 3), within the range of values observed in other ponds (Audet et al., 2020; Rabaey and Cotner, 2022). The $\%N_2O$ values did not show significant seasonal variations in any of the four sampled ponds (ANOVA $F_{(3,174)}=1,127$, $p=0.4091$) (Fig. 3). In individual ponds, $\%N_2O$ correlated negatively to temperature in the Tenreuken pond and Chl-$a$ in the Silex pond, and positively to SRP in the Silex pond and TSM concentration in the Tenreuken pond (Table S3; Fig S3, S4). We do not have a clear explanation for these correlations that might be spurious. The correlations with Chl-$a$ and TSM were surprising since they were observed in the two clear-water ponds and might indirectly reflect seasonal variations (with minimal values of these two quantities in summer). More surprisingly, $\%N_2O$ was not correlated with DIN (Table S3; Fig S3, S4, S5, S6) nor with individual forms of DIN ($NH_4^+$, $NO_2^-$, $NO_3^-$) in the four ponds individually or when all the data were pooled together for the individual forms of DIN (Table S3; Fig S7). In a previous study of the variation of GHGs in 22 urban ponds in the city of Brussels sampled only once during each season, $\%N_2O$ correlated positively with DIN, $NH_4^+$, $NO_2^-$, and $NO_3^-$. The range of variation of DIN and $\%N_2O$ across these 22 ponds (2 to 625 $\mu$mol L$^{-1}$ for DIN, and 0 to 10,354% for $\%N_2O$) was wider than the one observed in the present study of only four ponds (1 to 135 $\mu$mol L$^{-1}$ for DIN, and 32 to 826% for $\%N_2O$) (Fig. S8). The four ponds studied here are located at the periphery of the city and most probably receive less atmospheric nitrogen deposition than closer to the city center. A lower atmospheric nitrogen deposition in the periphery than in the city center is consistent with the correlation between $\%N_2O$ and atmospheric nitrogen dioxide ($NO_2$), and the correlation between $\%N_2O$ and the distance from the city center (Fig. S8). Atmospheric nitrogen deposition has been shown to enhance denitrification and $N_2O$ production in lakes (McCrackin and Elser, 2010; Palacin-Lizarbe et al., 2020).

The relationships between GHG dissolved concentrations and other variables were similar in clear-water macrophyte-dominated ponds and turbid-water phytoplankton-dominated ponds. $pCO_2$ was positively correlated with precipitation, and dissolved $CH_4$ concentration was positively correlated with temperature, while no significant correlation was found between $\%N_2O$ and other variables in the four ponds taken individually. The negative correlation between $pCO_2$ and $\%O_2$ reflected the photosynthesis-respiration balance independently from the community driving aquatic primary production (macrophytes in clear-water ponds and phytoplankton in turbid-water ponds).

### 3.2. Drivers of bubble flux

The bubble flux measured with inverted funnels in the four sampled ponds in the city of Brussels ranged between 0 and 2078 ml m$^{-2}$ d$^{-1}$ and strongly increased with water temperature (Fig. 4) and were overall higher in summer (837$\pm$434 mL m$^{-2}$ d$^{-1}$) than in spring (198$\pm$170 mL m$^{-2}$ d$^{-1}$) and fall (106$\pm$63 mL m$^{-2}$ d$^{-1}$) (Tukey's HSD test $p<0.0001$ for summer versus spring and summer versus fall). The bubble flux values in the four sampled ponds in the city of Brussels were within the range of values reported in lentic systems of equivalent size by Wik et al. (2013) (0 to 2772 mL m$^{-2}$ d$^{-1}$), Delsontro et al. (2016) (11 to 748 mL m$^{-2}$ d$^{-1}$) and Ray and Holgerson (2023) (0 to 2079 mL m$^{-2}$ d$^{-1}$). The mean $CH_4$ content of the bubbles in the four sampled ponds in the city of Brussels was 31$\pm$21%, and comparable to the values obtained by Wik et al. (2013) (35$\pm$25%), Delsontro et al. (2016) (58$\pm$25%) and Ray and Holgerson (2023) (25$\pm$13%) in lentic systems of equivalent size. The $CH_4$ content of the

bubbles increased with bubble flux (Fig. 4). These patterns between bubble flux and temperature and %CH$_4$ were most
probably related to the strong dependence of methanogenesis on temperature (Schulz and Conrad, 1996). The increase of
methanogenesis with temperature leads to the build-up of gas bubbles in sediments that are richer in CH$_4$, and consequently
to higher bubble fluxes with a higher CH$_4$ content at higher temperatures.

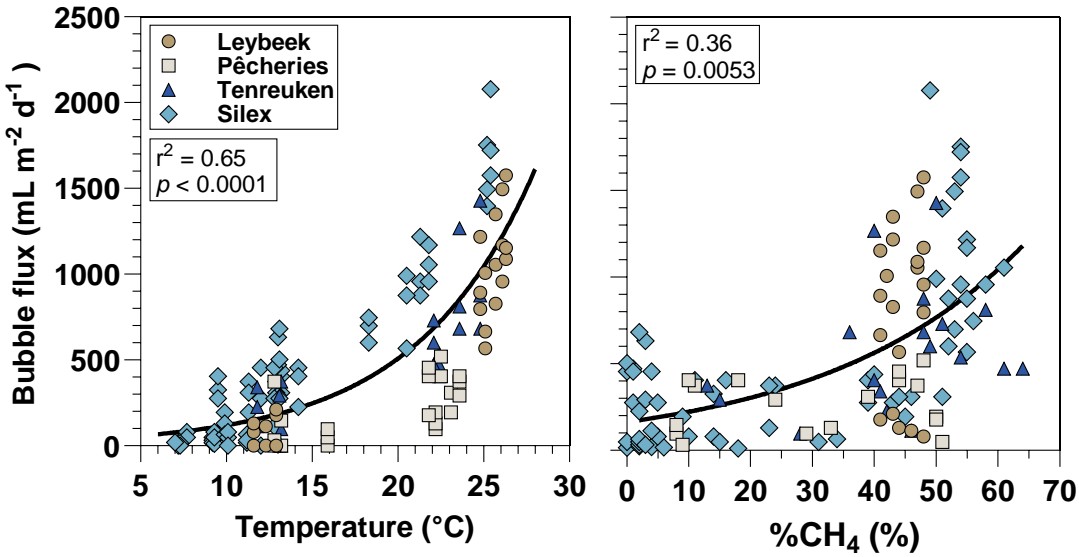


**Figure 4: Bubble flux (ml m$^{-2}$ d$^{-1}$) as a function of water temperature (°C) and the relative CH$_4$ content in bubbles (%CH$_4$, in %)**
**in four urban ponds (Leybeek, Pêcheries, Tenreuken, and Silex) in the city of Brussels (Belgium). Bubbles fluxes were measured**
**with three bubble traps in spring, summer, and fall of 2022 and 2023, totalling 8 days in the Leybeek, Pêcheries, and Tenreuken**
**ponds and 24 days in the Silex pond. Given the shallowness of the sampled systems (<1.5 m, Fig. 1) we assume that sediments**
**experience the same temperature as surface waters. Solid lines represent exponential regression fit of bubble flux as function of**
**temperature ($Y = 28 \cdot e^{0.14 \cdot X}$, n=139), and as function of relative CH$_4$ content in the bubbles ($Y = 164 \cdot e^{0.03 \cdot X}$, n=123) (Table**
**S11).**

Bubbling events are known to also be triggered by a decrease of hydrostatic pressure on the sediments due to water level
fluctuations or changes in atmospheric pressure. Drops in atmospheric pressure have been documented to trigger bubble
fluxes from lake sediments (Tokida et al., 2007; Scandella et al., 2011; Varadharajan and Hemond, 2012; Wik et al., 2013;
Taoka et al., 2020; Zhao et al., 2021). The bubble fluxes were measured during more lengthy series at the Silex pond than
the other three ponds for logistical reasons allowing investigating in more the detail the effects of temperature and
atmospheric pressure variations on bubble fluxes. In spring 2022, the bubble flux at the Silex pond increased during events
of drops in atmospheric pressure (depressions) (Fig. 5). There was no relation between wind speed and peaks of bubble flux
($r^2 = 0.01$, p=0.4629) as shown in Gatun Lake (Keller and Stallard, 1994), suggesting a more important role of changes of
atmospheric pressure than wind speed in the Silex pond in spring 2022. The bubble flux at the Silex pond was higher in
summer (1152±433 mL m$^{-2}$ d$^{-1}$) than during spring (198±170 mL m$^{-2}$ d$^{-1}$) (Tukey's HSD test p<0.0001), and the temporal
changes tracked those of water temperature. The bubble flux was modelled as function of temperature alone or as function of
both temperature and pressure changes (Figs. 5, S9). The inclusion of the term of pressure drops in addition to temperature
improved the performance of the model compared to the original data, for periods of low temperature (<15°C) but not for
warmer periods (>15°C) (Figs. 5, S9) when bubbling fluxes were quantitatively more important. The inclusion of the term of
pressure changes only improved the performance of the model compared to the original data very marginally when
comparing the full temperature range (<15°C and >15°C) (Fig. S9), showing that the intensity of bubble flux was mainly
driven by temperature change at yearly scales, in agreement with previous studies (*e.g.* Wik et al., 2013; DelSontro et al.,
2016; Aben et al., 2017; Ray and Holgerson, 2023).

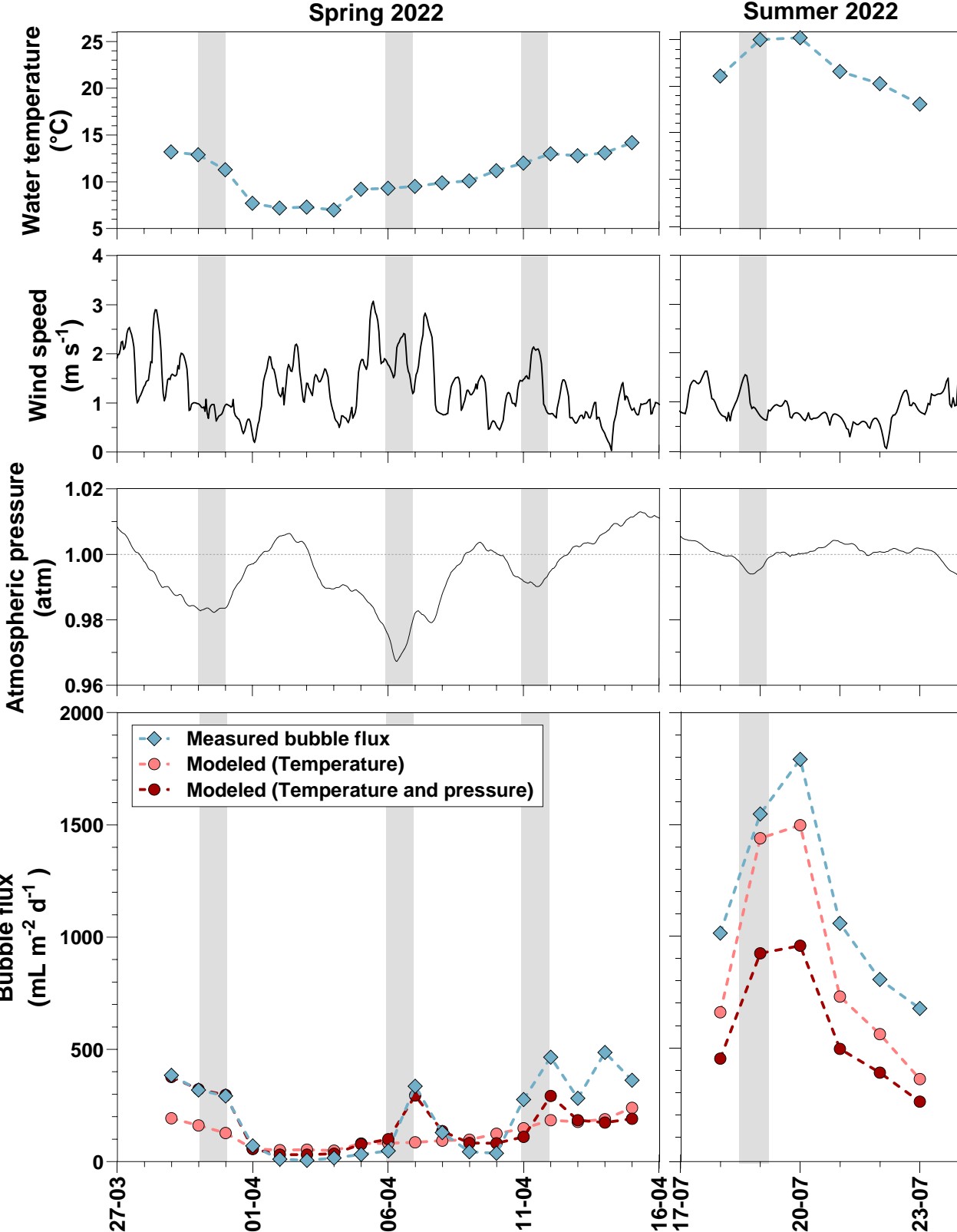


**Figure 5: Water temperature (°C), wind speed (m s⁻¹), atmospheric pressure (atm), and measured and modeled bubble flux (ml m⁻² d⁻¹) in the Silex pond from the 29 March 2022 to the 15 April 2022 and from the 18 July 2022 to the 23 July 2022. The bubble flux was modelled from a fit to data based on temperature alone and from both temperature and drops in atmospheric pressure.**

## 3.3. Drivers of methane ebullitive fluxes

Ebullitive $CH_4$ fluxes in the four ponds ranged between 0 and 59 mmol m$^{-2}$ d$^{-1}$, within the range reported in lentic systems (*e.g.* Deemer and Holgerson, 2021) and were positively related to temperature (Fig. 6) as shown previously in other small lentic systems (*e.g.* Wik et al., 2013; DelSontro et al., 2016; Aben et al., 2017; Ray and Holgerson, 2023; Rabaey and Cotner, 2024). The fitted relations between ebullitive $CH_4$ fluxes and temperature were specific to each pond and encompassed the fitted relations established in similar systems: four small ponds in Québec (DelSontro et al., 2016) and a small urban pond in the Netherlands (Aben et al., 2017). The $Q_{10}$ of $CH_4$ ebullition values ranged between 4.4 in the deeper Pêcheries pond and 26.9 in the shallower Leybeek pond, respectively (Table S7). The $Q_{10}$ of $CH_4$ ebullition in the four studied ponds of the city of Brussels, in Québec (DelSontro et al., 2016), and in the Netherlands (Aben et al., 2017) were negatively related to water depth (Fig. 6). An increase in water temperature leads to a smaller increase in $CH_4$ ebullitive fluxes (lower $Q_{10}$) in deeper ponds as the impact of hydrostatic pressure on sediments is higher in deeper ponds compared to shallow ponds, restricting bubble formation and release (*e.g.* DelSontro et al., 2016).

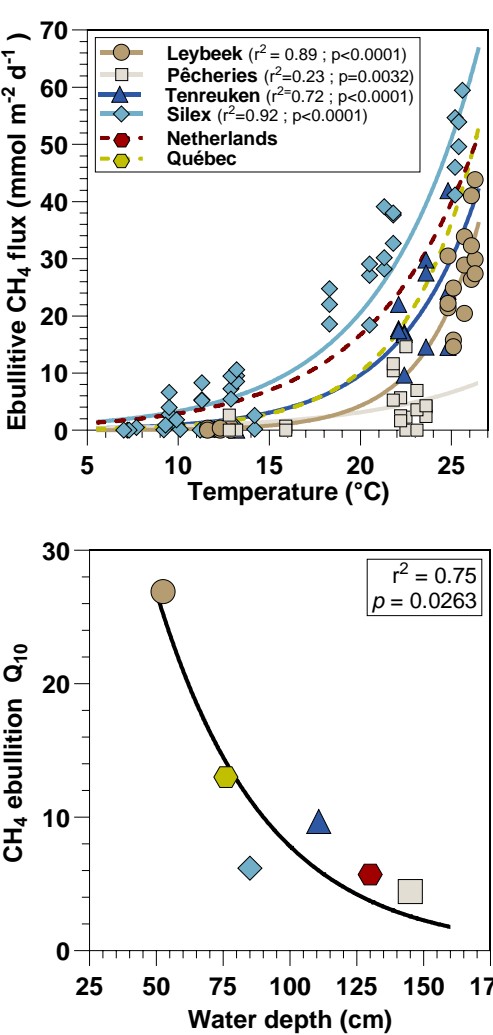

**Figure 6: Measured ebullitive $CH_4$ fluxes (mmol m$^{-2}$ d$^{-1}$) as function of water temperature (°C) in four urban ponds (Leybeek, Pêcheries, Tenreuken, and Silex) in the city of Brussels (Belgium), in spring, summer, and fall of 2022 and 2023, totalling 8 days in the Leybeek, Pêcheries, and Tenreuken ponds and 24 days in the Silex pond, with three bubble traps. Solid lines represent exponential fit for the Leybeek ($Y = 0.01 \cdot e^{0.32 \cdot X}$, n=22), Pêcheries ($Y = 0.16 \cdot e^{0.15 \cdot X}$, n=22), Tenreuken ($Y = 0.10 \cdot e^{0.23 \cdot X}$, n=19), Silex ($Y = 0.54 \cdot e^{0.18 \cdot X}$, n=72) ponds (Table S7) dashed lines represent exponential fit established in similar systems: four small ponds in Québec ($Y = 0.06 \cdot e^{0.25 \cdot X}$) (DelSontro et al., 2016) and a small urban pond in the Netherlands ($Y = 0.51 \cdot e^{0.17 \cdot X}$) (Aben et al., 2017). Each exponential curve allows to determine a $Q_{10}$ of $CH_4$ ebullition, plotted against water depth; solid line represents exponential regression fit ($Y = 92 \cdot e^{-0.02 \cdot X}, n = 6$) (Table S11).**

## 3.4. Relative contribution of methane ebullitive and diffusive fluxes

Diffusive $CH_4$ fluxes computed from dissolved $CH_4$ concentration and $k$ derived from wind speed ranged between 0.1 and 19.7 mmol m$^{-2}$ d$^{-1}$ (Fig. 7) within the range reported in lentic systems (*e.g.* Deemer and Holgerson, 2021). The diffusive $CH_4$ fluxes tended to be higher in summer and spring than in fall and winter owing to the strong positive dependency between $CH_4$ and water temperature (Fig. 3; Table S3). In addition, wind speed only showed small seasonal variations during sampling (0.6±0.6m s$^{-1}$ in spring, 0.3±0.2 m s$^{-1}$ in summer, 0.7±0.7 m s$^{-1}$ in fall, and 0.6±0.2 m s$^{-1}$ in winter) (Fig. 3). Ebullitive $CH_4$ fluxes were calculated from the relations with temperature for each pond given in Figure 6 from the temperature data coincident with the diffusive $CH_4$ fluxes (Fig. 7). The resulting calculated ebullitive $CH_4$ fluxes allowed to compare and integrate seasonally both components of $CH_4$ emissions to the atmosphere, and to calculate the relative contribution of ebullition to total (diffusive+ebullitive) $CH_4$ emissions.

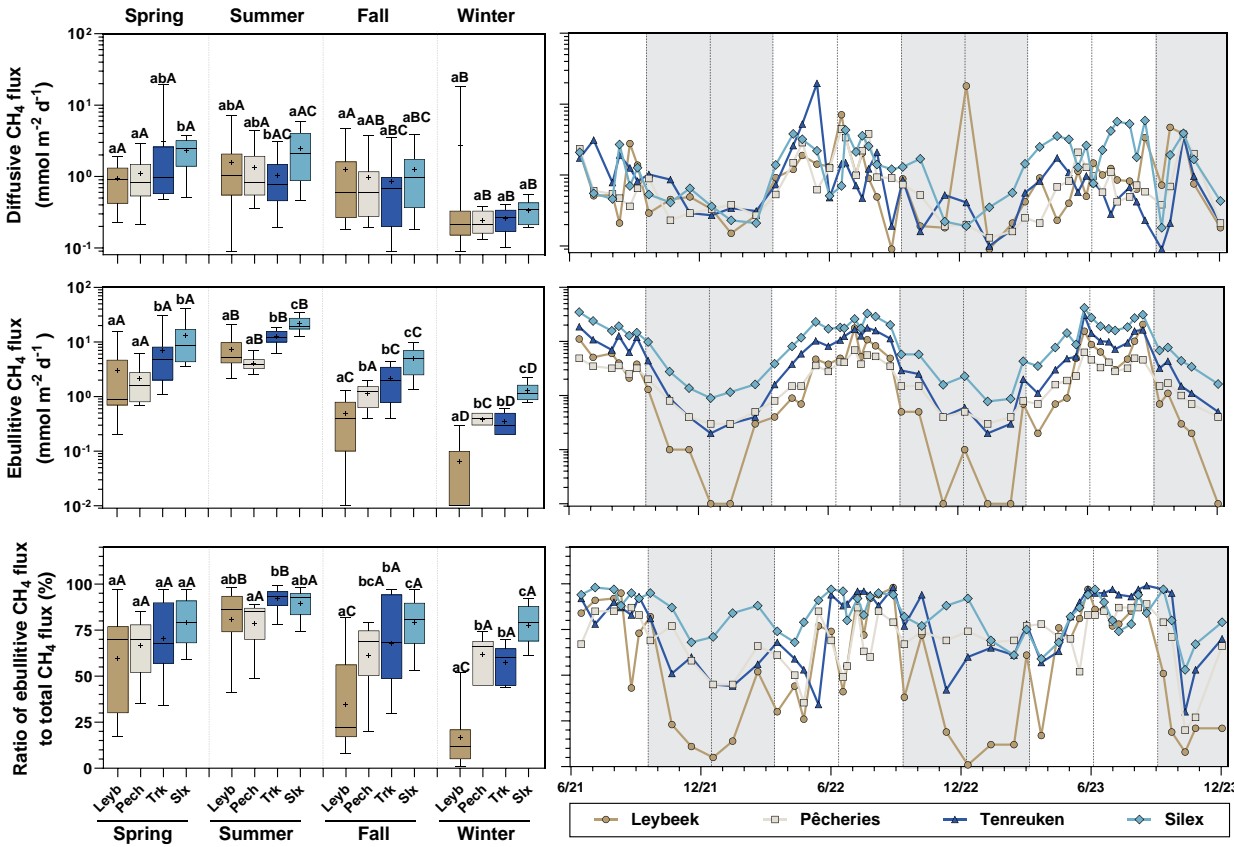

**Figure 7: Seasonal variations of diffusive and ebullitive $CH_4$ fluxes (mmol m$^{-2}$ d$^{-1}$), and the ratio of ebullitive $CH_4$ flux to total (ebullitve+diffusive) $CH_4$ flux (%) in four urban ponds (Leybeek (Leyb), Pêcheries (Pech), Tenreuken (Trk), and Silex (Slx)) in the city of Brussels (Belgium) from June 2021 to December 2023. Diffusive fluxes were calculated from $CH_4$ concentration and gas transfer velocity derived from wind speed. Ebullitive $CH_4$ fluxes were calculated from the relations with temperature for each pond (Fig. 6; Table S7) from the temperature data coincident with the diffusive $CH_4$ fluxes. Box plots show median (horizontal line), mean (cross), and 25–75% percentiles (box limits). Whiskers extend from minimum to maximum values. White and grey bands in the graphs on the right correspond to the autumn/winter and spring/summer periods, respectively, and dotted vertical bars represent the first days of each season. ANOVA results of the multiple comparison between boxplots are summarized in Tables S4 and S5. Different lower-case letters indicate significant differences between ponds within a season and different upper-case letters indicate significant differences between seasons for a given pond.**

The relative contribution of ebullition to total $CH_4$ emissions ranged between 1 and 99% in the four sampled ponds in the city of Brussels (Fig. 7), within the range reported in lentic systems (*e.g.* Deemer and Holgerson, 2021). Owing to the strong dependency of ebullitive $CH_4$ fluxes to temperature (Table S7; Fig. 6), the mean relative contribution of ebullition to total $CH_4$ emissions for all data pooled together was higher in summer (85±7 %) compared to spring (69±14 %, Tukey's HSD test

p=0.0104), fall (61±18 %, Tukey's HSD test p<0.0001), and winter (53±8 %, Tukey's HSD test p<0.0001) (Fig. 7). This finding is consistent with other studies showing that ebullitive $CH_4$ fluxes can account for more than half of total $CH_4$ emissions in small and shallow lentic systems (*e.g.* Wik et al., 2013; Deemer and Holgerson, 2021; Ray and Holgerson, 2023; Rabaey and Cotner, 2024). The relative contribution of ebullition to total $CH_4$ emissions was lowest during the other seasons, especially in the Leybeek pond (Fig. 7). Owing to the strong dependency of ebullitive $CH_4$ fluxes to temperature, the relative contribution of ebullition to total $CH_4$ emissions was related to temperature in the four ponds (Fig. S10), as previously also shown in Québec ponds (DelSontro et al., 2016).

The values of $Q_{10}$ of diffusive $CH_4$ fluxes were lower than those for ebullitive $CH_4$ fluxes, less variable (1.2 in the Pêcheries pond to 2.9 in the Silex pond), and less statistically significant (Table S7). Other studies have also reported higher $Q_{10}$ for $CH_4$ ebullition than for $CH_4$ diffusion in lentic systems (DelSontro et al., 2016; Xun et al., 2024). The lower dependence to temperature of $CH_4$ diffusion compared to $CH_4$ ebullition might be related to a lower relative change of $CH_4$ concentrations and $k$ to temperature change. $CH_4$ concentrations in surface water are very strongly affected by MOX (see hereafter). A relative increase of $CH_4$ production in sediments by methanogenesis will lead to a stronger increase of $CH_4$ emission by ebullition than by diffusion because of a mitigation by MOX on $CH_4$ diffusion. Additionally, $k$ depends on wind speed, but the warmer periods of the year (summer) tended to be less windy (~0.3 m s$^{-1}$) than the other seasons (>0.6 m s$^{-1}$) also contributing to lower dependence on temperature of $CH_4$ diffusion compared to ebullition and lower $Q_{10}$ values.

The annual averaged diffusive and ebullitive fluxes of $CH_4$ in the four ponds in the city of Brussels were plotted against Chl-*a* concentration, total macrophyte cover in summer, water depth, and lake surface area (Fig. 8) that are frequent predictors of variations of $CH_4$ fluxes among lakes (Holgerson and Raymond, 2016; DelSontro et al., 2018, Deemer and Holgerson, 2021; Casas-Ruiz et al., 2021; Borges et al., 2022). The annual diffusive $CH_4$ flux was significantly lower in the slightly deeper Pêcheries pond (130 cm depth) than the two slightly shallower ponds (Leybeek (60 cm depth) and Silex (110 cm depth) ponds) (Tukey's HSD test p=0.0007 for Pêcheries versus Leybeek, p<0.0001 for Pêcheries versus Silex), and the annual ebullitive $CH_4$ flux was significantly lower in the Pêcheries pond than the Silex pond (Tukey's HSD test p<0.0001) but was not significantly different than the Leybeek pond (Tukey's HSD test p=0.3847). No other significant differences in annual diffusive and ebullitive $CH_4$ fluxes related to water depth or surface area were observed. The narrow range of variation of water depth (50 to 150 cm) and surface area (0.7 to 3.2 ha) could explain the lack of a clear decrease of diffusive and ebullitive $CH_4$ fluxes with increasing depth or surface that are frequent predictors of variations of $CH_4$ fluxes among ponds (*e.g.* Holgerson, 2015; Holgerson and Raymond, 2016; Ray et al., 2023; Theus et al., 2023) and lakes (*e.g.* Kankaala et al., 2013; DelSontro et al., 2018, Deemer and Holgerson, 2021; Casas-Ruiz et al., 2021; Borges et al., 2022). Correlations between $CH_4$ fluxes and depth or lake surface area have been shown among lakes across much larger ranges of variation of lake depth (Borges et al., 2022) and surface area (Kankaala et al., 2013; Holgerson and Raymond, 2016; Casas-Ruiz et al., 2021).

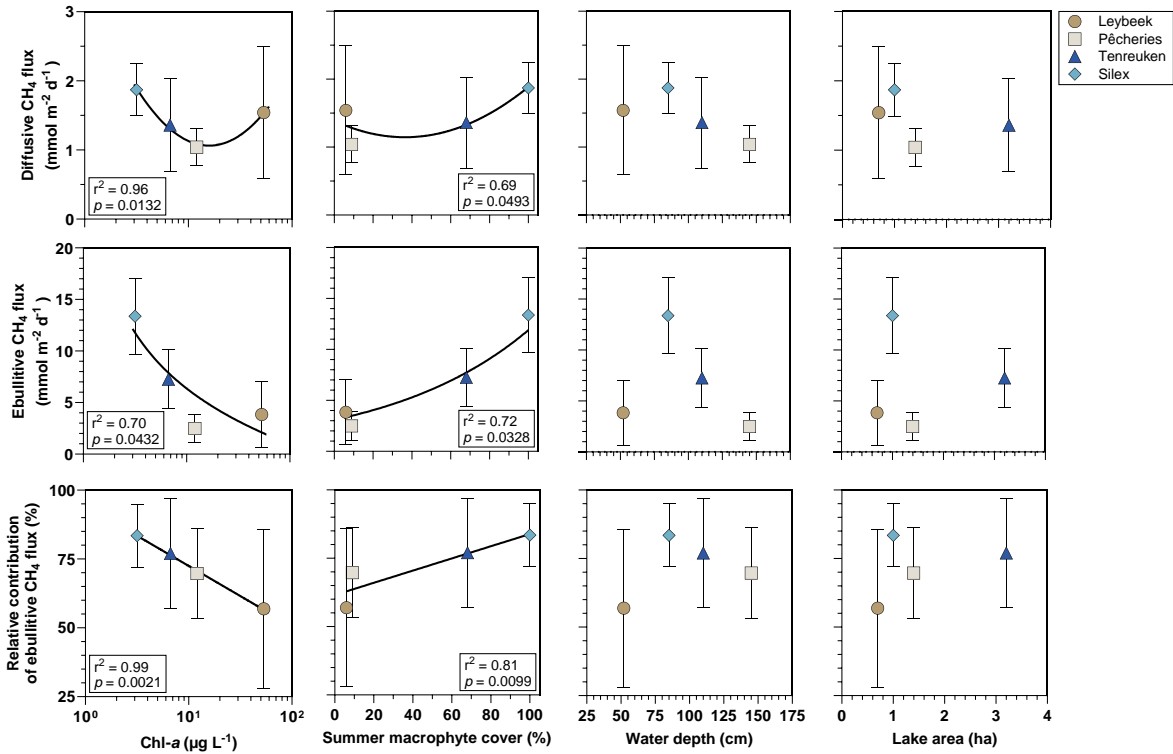

519

**Figure 8:** Mean diffusive and ebullitive $CH_4$ fluxes (mmol m$^{-2}$ d$^{-1}$) and mean ratio of ebullitive $CH_4$ flux to total (diffusive+ebullitive) $CH_4$ flux (%) versus chlorophyll-*a* (Chl-*a*, in μg L$^{-1}$), total macrophyte cover in summer (%), water depth (cm), and lake surface area (ha) in four ponds (Leybeek, Pêcheries, Tenreuken, and Silex) in the city of Brussels (Belgium) from June 2021 to December 2023. Error bars indicate the standard deviation. Dashed lines indicate trends in relationship between variables (Table S11).

The annual ebullitive $CH_4$ fluxes were higher in the two clear-water ponds (7.3±2.9 and 13.4±3.7 m$^{-2}$ d$^{-1}$ in the Tenreuken and Silex ponds, respectively) than the two turbid-water ponds (3.8±3.2 and 2.5±1.4 m$^{-2}$ d$^{-1}$ in the Leybeek and Pêcheries ponds, respectively) (Tukey's HSD test p<0.0001 for each comparison between a clear-water pond and a turbid-water pond). The annual ebullitive $CH_4$ fluxes were significantly higher in the Silex pond than the Tenreuken pond (Tukey's HSD test p<0.0001) that showed a higher macrophyte cover during summer (100% in the Silex pond and 68% in the Tenreuken pond) and were not significantly different in the two turbid-water Leybeek and Pêcheries ponds (Tukey's HSD test p=0.3847) that showed similar macrophyte cover during summer (6 and 9% in the Leybeek and Pêcheries ponds, respectively) (Fig. 8). The annual ebullitive $CH_4$ fluxes were overall positively correlated to macrophyte cover and negatively correlated to Chl-*a* (Fig. 8). The higher ebullitive $CH_4$ emissions from the clear-water ponds would suggest that the delivery of organic matter to sediments from macrophytes sustained a larger methane production than from phytoplankton. This finding is consistent with the notion that vegetated littoral zones of lakes are hot spots of $CH_4$ production and emission (*e.g.* Hyvönen et al., 1998; Huttunen et al., 2003; Juutinen et al., 2003; Desrosiers et al., 2022). In other small lentic systems, the $CH_4$ dissolved concentrations and diffusive fluxes have also been shown to correlate positively with macrophyte cover (*e.g.* Ray et al., 2023; Theus et al., 2023).

The annual diffusive $CH_4$ flux was higher in the two clear-water ponds (1.4±0.7 and 1.9±0.4 mmol m$^{-2}$ d$^{-1}$ in the Tenreuken and Silex ponds, respectively) than in the turbid-water Pêcheries pond (1.0±0.3 mmol m$^{-2}$ d$^{-1}$) (Tukey's HSD test p=0.0404 for Tenreuken versus Pêcheries, and p<0.0001 for Silex versus Pêcheries), which was consistent with the pattern of higher ebullitive $CH_4$ emissions from clear-water ponds (Fig. 8). In the four sampled urban ponds, annual $CH_4$ diffusive fluxes were significantly higher in the pond with the highest total macrophyte cover in the clear-water ponds, and significantly higher in the pond with highest Chl-*a* concentration in the turbid-water ponds (Fig. 8). An increase in methane production with

phytoplankton biomass in turbid-water ponds has also been reported by other studies in lakes (*e.g.* Yan et al., 2019; Bartosiewicz et al., 2021). Since total macrophyte cover and Chl-*a* were anti-correlated, we hypothesize that the variations of $CH_4$ diffusive fluxes follow a U-shaped relation with either Chl-*a* or macrophyte cover. Higher values of annual $CH_4$ diffusive fluxes occurred at the extreme values of Chl-*a* or of macrophyte cover (minimum or maximum), and lower values occurred at the intermediate values of Chl-*a* or macrophyte cover. The relative contribution of ebullitive $CH_4$ fluxes to the total flux was higher in the clear-water Silex pond, which had the highest macrophyte cover, compared to the two turbid-water ponds with lower macrophyte cover (Tukey's HSD test $p<0.0001$ for Silex versus Leybeek, $p=0.0056$ for Silex versus Pêcheries), and was higher in the clear-water Tenreuken pond than in the turbid-water Leybeek pond (Tukey's HSD test $p<0.0001$) (Fig. 8). The relative contribution of ebullitive $CH_4$ fluxes to the total $CH_4$ flux seems to increase concomitantly with the macrophyte cover (Fig. 8), and was overall strongly positively correlated to macrophyte cover and negatively to Chl-*a* (Fig. 8). These patterns are consistent with the idea of an increase of ebullition relative to diffusive $CH_4$ emissions in vegetated sediments compared to unvegetated sediments (e.g. Desrosiers et al., 2022; Ray et al., 2023; Theus et al., 2023).

The annual diffusive and ebullitive fluxes in the four ponds in the city of Brussels were within the range of values for ponds of similar surface area (0.4 to 4.0 ha) compiled by Deemer and Holgerson (2021) (Fig. S11). The linear regression of ebullitive $CH_4$ fluxes as a function of diffusive $CH_4$ fluxes allows comparing the data of ebullitive $CH_4$ fluxes from the four Brussels ponds "normalized" to the diffusive $CH_4$ fluxes. The ebullitive $CH_4$ fluxes from the two turbid-water ponds (Pêcheries and Leybeek) were very close to the linear regression showing they were characterized by ebullitive $CH_4$ fluxes equivalent to those in the ponds compiled by Deemer and Holgerson (2021) when normalized by the diffusive fluxes. The ebullitive $CH_4$ fluxes from the two clear-water ponds (Tenreuken and Silex) were above the linear regression showing they were characterized by ebullitive $CH_4$ fluxes above those in the ponds compiled by Deemer and Holgerson (2021) when normalized by the diffusive fluxes. We hypothesize the relatively higher ebullitive fluxes in the two clear-water ponds were related to enhancement of ebullition from organic matter subsidized by macrophytes. This hypothesis is consistent with the two clear-water ponds in Brussels having higher ebullitive fluxes than in the ponds compiled by Deemer and Holgerson (2021) at equivalent Chl-*a* values (Fig. S11). The observed high ebullitive fluxes in the clear-water ponds would suggest that Chl-*a* concentration alone fails to predict ebullitive fluxes in macrophyte-dominated clear-water ponds. Consequently, global scaling of $CH_4$ fluxes in lentic systems using Chl-*a* as a predictor as used in lakes (*e.g.* DelSontro et al., 2018) might under-estimate ebullitive $CH_4$ emissions due to a misrepresentation of macrophyte-dominated clear-water ponds.

The annual averaged diffusive fluxes of $CO_2$ ($F_{CO2}$) and $N_2O$ ($F_{N2O}$) in the four ponds in the city of Brussels were also plotted against Chl-*a* concentration, total macrophyte cover in summer, water depth, and lake surface area, as well as DIN for $N_2O$ fluxes (Figs. S12, S13, S14). Annual $F_{CO2}$ did not show significant differences between the four studied ponds (Tukey's HSD test: $p>0.05$ for each comparison), and $F_{CO2}$ did not significantly correlate to the other variables (Chl-*a* concentration, total macrophyte cover, water depth, and lake surface area). This might be surprising since other studies have reported lower $CO_2$ fluxes in more productive lentic systems (*e.g.* Sand-Jensen and Staehr 2007; Borges et al. 2022). We hypothesize that given that the four systems were either phytoplankton-dominated or macrophyte-dominated, in both cases, the ponds had an important submerged productivity resulting in a relatively invariant $F_{CO2}$ as function of either Chl-*a* or macrophyte cover. Annual mean $F_{CO2}$ was also uncorrelated to water depth and lake area (Fig. S12). This might have resulted from the relative similarity of depth and surface area of the four studied ponds, as it is well established that $CO_2$ emissions strongly increase with decreasing size of ponds (Holgerson and Raymond, 2016). Annual $F_{N2O}$ was not significantly different between clear-water and turbid-water ponds. $F_{N2O}$ was significantly lower in the slightly deeper Pêcheries pond than the two slightly shallower Leybeek and Silex ponds (Fig. S13) (Tukey's HSD test $p=0.0012$ for Pêcheries vs. Leybeek, and $p=0.0052$ for

Pêcheries vs. Silex), and $F_{N2O}$ showed a significant negative relationship with water depth (Fig. S13). We hypothesize that
this might reflect a larger dilution of $N_2O$ diffusing from sediments in the deeper systems. $F_{N2O}$ did not correlate to DIN,
$NH_4^+$, $NO_2^-$, and $NO_3^-$ (Fig. S14). We hypothesize that this reflects the rather narrow range of annual DIN average values in
the four studied ponds (~24 to ~29 µmol $L^{-1}$), as DIN, $NH_4^+$, $NO_2^-$, and $NO_3^-$ were not statistically different between ponds
(Tukey's HSD test p>0.05 for every comparison).

### 3.5. Methanogenesis pathway inferred from δ¹³C-CH₄ in bubbles

$\delta^{13}C$-$CH_4$ was measured in bubbles trapped during the ebullition flux measurements and in bubbles collected by perturbing
the sediments. The variations of $\delta^{13}C$-$CH_4$ suggest that there could have been variations of the relative importance of
hydrogenotrophic versus acetoclastic pathways of methanogenesis among different ponds but also seasonally.
Methanogenesis by the hydrogenotrophic pathway produces $CH_4$ with more negative $\delta^{13}C$-$CH_4$ values (-100‰ to -60‰)
compared to the acetoclastic pathway (-65‰ to -50‰) (Whiticar et al., 1986). Yet, it remains unclear which environmental
factors determine the relative importance of hydrogenotrophic and acetoclastic methanogenesis pathways (Conrad et al.,

597 2011).

The $\delta^{13}C$-$CH_4$ values in the trapped bubbles for the all dataset were statistically more negative in fall (-83.2±5.2 ‰) than
summer (-69.5±3.2 ‰) and spring (-68.2±4.4 ‰) (Fig. 9; Table S8) (Tukey's HSD test p<0.0001 for fall versus summer, and
fall versus spring), suggesting a dominance of hydrogenotrophic methanogenesis in fall compared to spring and summer
when acetoclastic methanogenesis seemed dominant. Hydrogenotrophic methanogenesis occurs at higher temperatures than
acetoclastic methanogenesis (Schulz and Conrad, 1996; Schulz et al., 1997), however, temperature in fall (11.9±3.7 °C) was
lower than in summer (21.1±1.9 °C) (Tukey's HSD test p<0.0001). A shift from acetoclastic methanogenesis to
hydrogenotrophic methanogenesis has been documented in response to the increase of $NH_4^+$ concentration (Ni et al., 2022;
Wang et al., 2022) and the decrease of pH (Kotsyurbenko et al., 2007) expected in response to an increase of $CO_2$. An
increase of $NH_4^+$ and decrease of pH in pore waters in fall compared to summer and spring would be consistent with the
sustained benthic organic matter degradation leading to a gradual change of pore water chemistry from spring to fall.

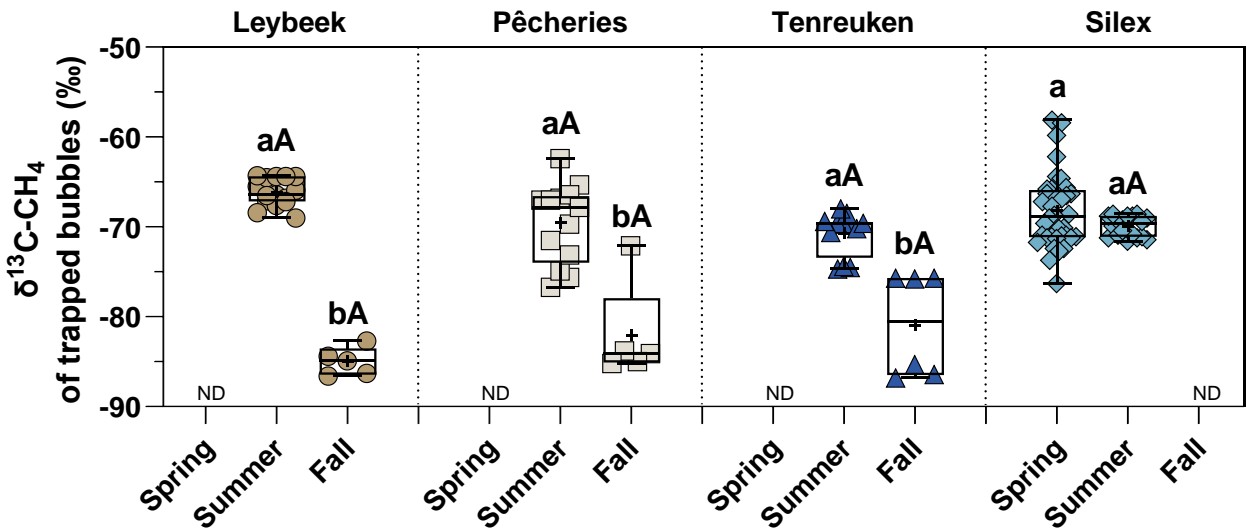


**Figure 9: ¹²C/¹³C ratio of CH₄ (δ¹³C-CH₄, in ‰) in bubbles collected during ebullitive flux measurements ("trapped bubbles") in**
**four urban ponds (Leybeek, Pêcheries, Tenreuken, and Silex) in the city of Brussels (Belgium), measured in spring, summer, and**
**fall in 2022 and 2023 (September 2023 and October 2023 in the Leybeek pond; July 2023 and October 2023 in the Pêcheries pond;**
**August 2023 and October 2023 in the Tenreuken pond; April 2022 and July 2022 in the Silex pond). Box plots show median**
**(horizontal line), mean (cross), and 25–75% percentiles (box limits). Whiskers extend from minimum to maximum values. ND = no**

In summer 2023, a survey of all four ponds was made to simultaneously sample bubbles by perturbation of the sediment for the determination of the $\delta^{13}$C-CH$_4$ in the released bubbles. The $\delta^{13}$C-CH$_4$ values of perturbed sediments were more negative in the clear-water macrophyte-dominated ponds (-80.1±0.1 ‰ and -78.4±1.2 ‰ in the Tenreuken and Silex ponds, respectively) than in the turbid-water phytoplankton-dominated ponds (-69.7±0.7 ‰ and -70.7±0.4 ‰ in the Leybeek and Pêcheries ponds, respectively) (Tukey's HSD test p<0.0001 for each comparison between a clear pond and a turbid pond) (Fig. 10). This pattern of $\delta^{13}$C-CH$_4$ of perturbed sediments could suggest a higher contribution of the hydrogenotrophic methanogenesis pathway compared to the acetoclastic pathway in the clear-water ponds where organic matter for methanogenesis was assumed to be mainly related to macrophytes rather than phytoplankton. Based on gene expression during incubations, Wang et al. (2023) suggested that acetoclastic methanogenesis pathway was stimulated by macrophyte organic carbon compared to phytoplankton organic matter in lakes Chaohu and Taihu in China. The distribution of $\delta^{13}$C-CH$_4$ data in the four urban ponds of the city of Brussels suggests the opposite pattern, with macrophyte organic carbon stimulating the hydrogenotrophic methanogenesis pathway. This pattern seems consistent with the more refractory nature of macrophyte organic carbon compared to the more labile nature of phytoplankton organic carbon. Organic matter from macrophytes has a large share of molecules difficult to degrade such as cellulose unlike organic matter from phytoplankton that is rich in polysaccharides and proteins (West et al., 2015; Berberich et al., 2020). In presence of more refractory organic matter, a partial fermentation would favour the production of H$_2$ over acetate which would favour hydrogenotrophic methanogenesis over acetoclastic methanogenesis (Liu et al., 2017).

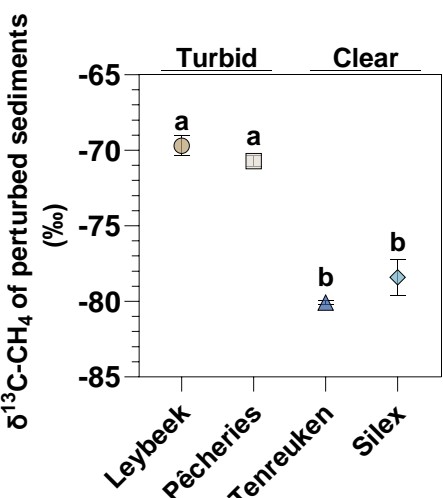

Figure 10: Mean ± standard deviation $^{13}$C/$^{12}$C ratio of CH$_4$ ($\delta^{13}$C-CH$_4$, in ‰) in bubbles released from sediments after physical perturbation ("perturbed sediments") in four ponds (Leybeek, Pêcheries, Tenreuken, and Silex) in the city of Brussels (Belgium) in summer 2023 (4th September 2023). Error bars indicate standard deviation on the mean. ANOVA results of the multiple comparison between boxplots are summarized in Table S9. Different lower-case letters indicate significant differences between ponds.

## 3.6. Methane oxidation

The $\delta^{13}$C-CH$_4$ of dissolved CH$_4$ in surface waters in the four sampled ponds in the city of Brussels ranged between -16 and -64 ‰ (Fig. 11). The $\delta^{13}$C-CH$_4$ of dissolved CH$_4$ in surface waters were generally higher than in sediments based on trapped bubbles during the ebullition measurements (-55 to -87 ‰; Fig. 9). The $^{13}$C enriched values of dissolved CH$_4$ in surface waters samples probably resulted from MOX. FOX in surface waters in the four sampled ponds in the city of Brussels

ranged between 22 and 97‰. MOX in surface waters in the four sampled ponds in the city of Brussels ranged between 0.1
and 73.0 mmol m$^{-2}$ d$^{-1}$ (Fig. 11).

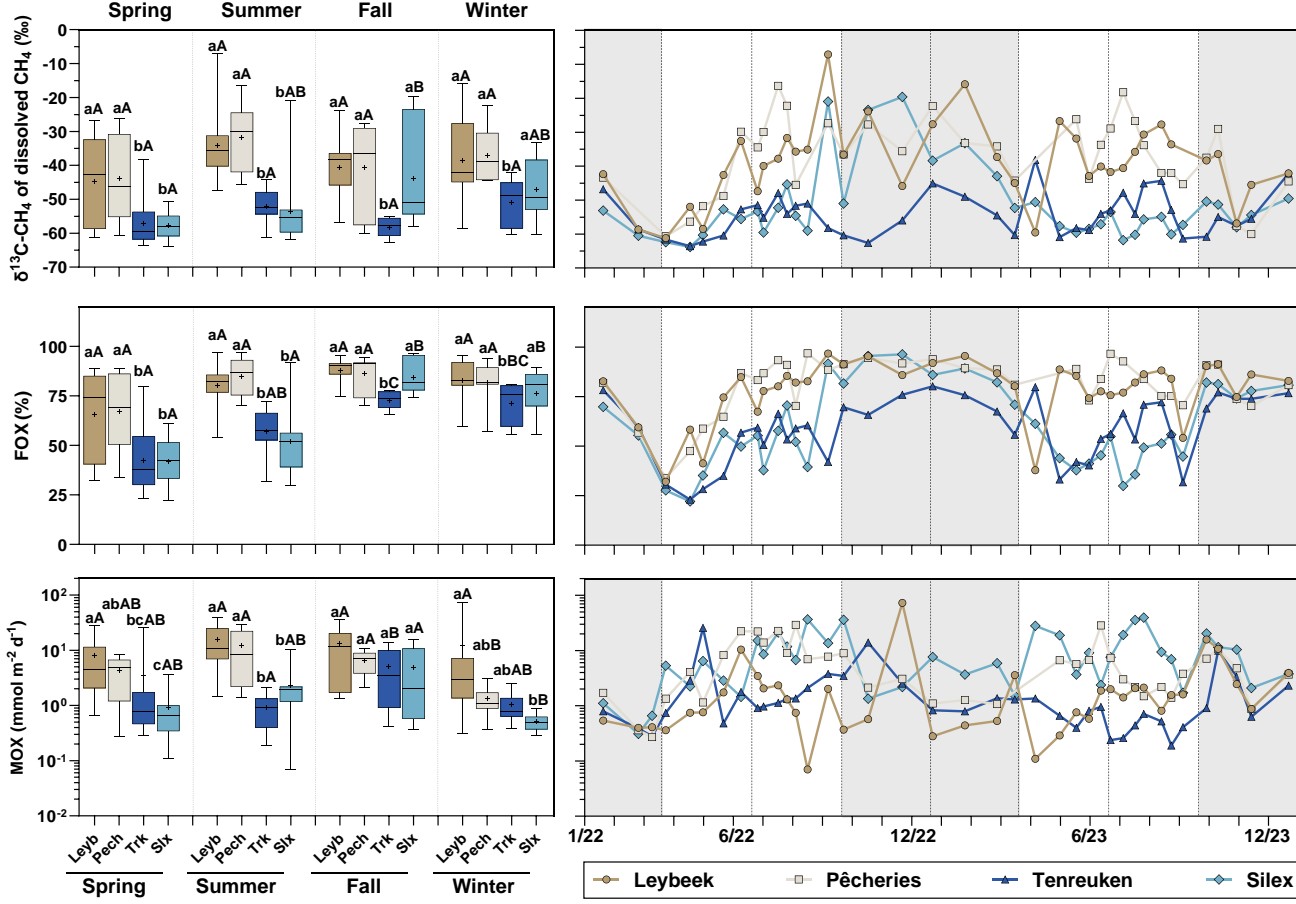


**Figure 11: Seasonal variations of $^{13}$C/$^{12}$C ratio of dissolved CH$_4$ in surface waters ($\delta^{13}$C-CH$_4$ of dissolved CH$_4$, in ‰), fraction of**
**CH$_4$ removed by methane oxidation (FOX, in %), and methane oxidation (MOX, in mmol m$^{-2}$ d$^{-1}$) in four urban ponds (Leybeek**
**(Leyb), Pêcheries (Pech), Tenreuken (Trk), and Silex (Slx)) in the city of Brussels (Belgium) from January 2022 to December 2023.**
**Box plots show median (horizontal line), mean (cross), and 25–75% percentiles (box limits). Whiskers extend from minimum to**
**maximum values. White and grey bands in the graphs on the right correspond to the fall/winter and spring/summer periods, and**
**dotted vertical bars represent the first days of each season. ANOVA results of the multiple comparison between boxplots are**
**summarized in Tables S4 and S5. Different lower-case letters indicate significant differences between ponds within a season and**
**different upper-case letters indicate significant differences between seasons for a given pond.**
FOX and MOX followed the same seasonal variations as $\delta^{13}$C-CH$_4$ of dissolved CH$_4$ since both quantities were derived from
isotopic models that include $\delta^{13}$C-CH$_4$ of dissolved CH$_4$. $\delta^{13}$C-CH$_4$ of dissolved CH$_4$, FOX, and MOX showed no significant
differences between seasons in the two turbid-water ponds except in the Pêcheries pond where MOX was lower in winter
(1.3±0.86 mmol m$^{-2}$ d$^{-1}$) than in summer (12.3±10.5 mmol m$^{-2}$ d$^{-1}$, Tukey's HSD test p=0.0010) and fall (6.5±3.0 mmol m$^{-2}$
d$^{-1}$, Tukey's HSD test p=0.0254) (Fig. 11). In the clear-water Silex pond, FOX was lower in spring (42±12 %) and summer
(52±16 %) than in fall (84±9 %) and winter (76±12 %) (Tukey's HSD test p< 0.0001 for spring or summer versus fall or
winter). In the clear-water Tenreuken pond, FOX was higher in fall (73±5 %) than in spring (42±17 %, Tukey's HSD test
p<0.0001) and summer (57±11 %, Tukey's HSD test p=0.0324), and higher in winter (71±10 %) than in spring (42±17 %,
Tukey's HSD test p<0.0001). $\delta^{13}$C-CH$_4$ of dissolved CH$_4$ and FOX were statistically higher in the turbid-water ponds
(Leybeek and Pêcheries) than in the clear-water ponds (Tenreuken and Silex) during spring and summer (Fig. 11) and than in
the Tenreuken pond during fall and winter (Fig. 11; Tables S4 and S5). These seasonal differences led to an annual MOX
that was statistically higher in the turbid-water ponds (10.8 and 7.2 mmol m$^{-2}$ d$^{-1}$ in the Leybeek and Pêcheries ponds,
respectively) than the clear-water ponds (2.4 and 4.4 mmol m$^{-2}$ d$^{-1}$ in the Tenreuken and Silex ponds, respectively) (Tukey's
HSD test p=<0.0001 for each turbid-water pond versus each clear-water pond). TSM and Chl-$a$ concentrations were higher
in the turbid-water ponds than in the clear-water ponds, particularly during productive phytoplanktonic periods of spring and
summer (Fig. 3), when the highest difference of $\delta^{13}$C-CH$_4$ of dissolved CH$_4$, FOX, and MOX were observed between the
turbid-water and the clear-water ponds (Fig. 11).
$\delta^{13}$C-CH$_4$ of dissolved CH$_4$, FOX, and MOX positively correlated to TSM and Chl-$a$ concentrations (Fig. 12). These patterns
could reflect the increase of micro-organisms including methanotrophs fixed on particles leading to an increase of MOX in
parallel to an increase of TSM concentration (Abril et al., 2007). Fixed micro-organisms can grow on inorganic particles and
aggregates of organic matter (Kirchman and Mitchell, 1982), but also on aggregates of living cyanobacteria (Li et al., 2021).
An increase of particles in the water column increases light attenuation in the water column which would alleviate the
inhibition of MOX by light (Dumestre et al., 1999; Murase and Sugimoto 2005; Morana et al., 2020), also possibly
contributing to a positive relation between MOX and TSM and Chl-$a$, along the turbidity gradient. Both processes could co-
occur contributing to the observed positive patterns between MOX and TSM and Chl-$a$ concentrations.

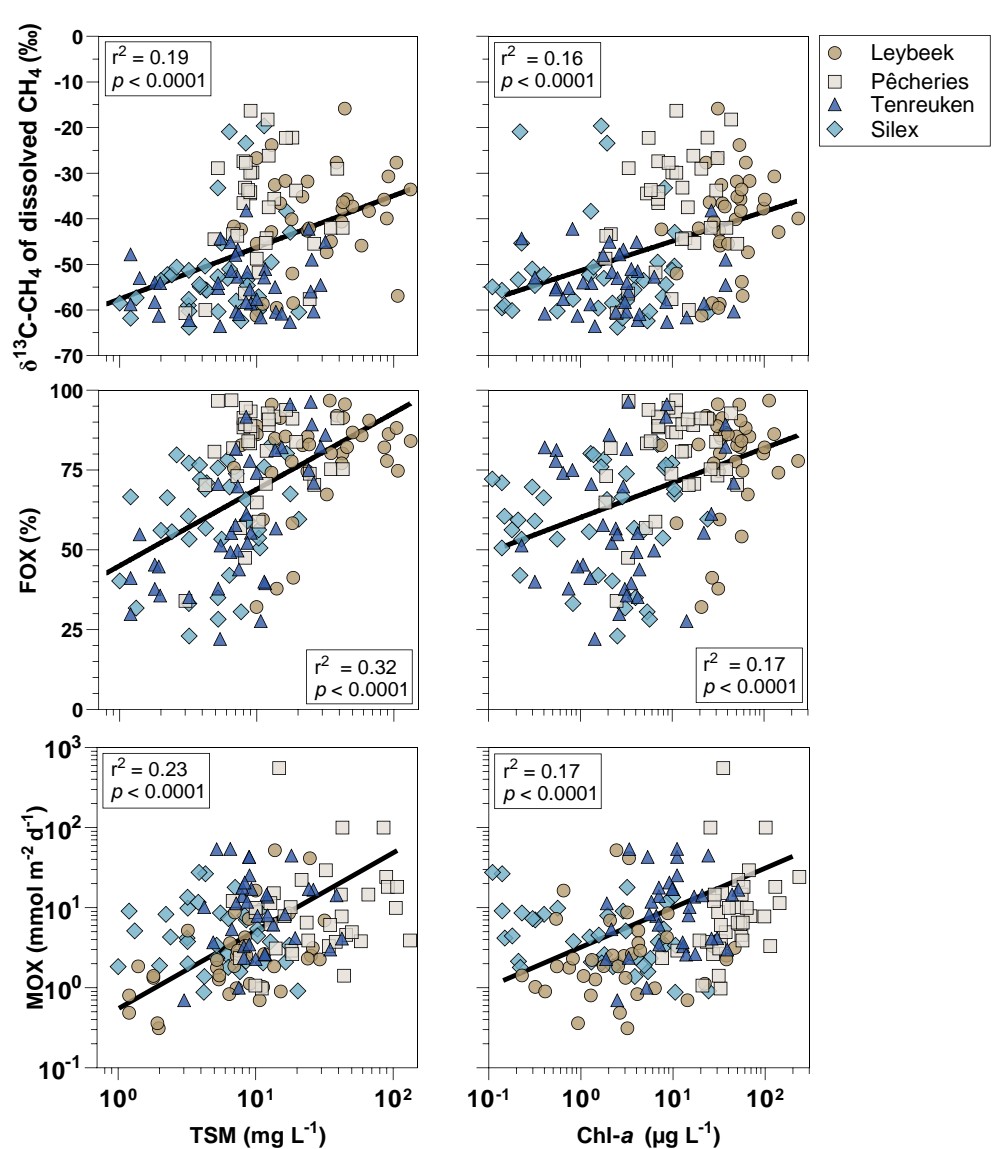


**Figure 12: $^{12}$C/$^{13}$C ratio of CH$_4$ in surface waters ($\delta^{13}$C-CH$_4$, in ‰), fraction of CH$_4$ removed by methane oxidation (FOX, in %),**
**and methane oxidation flux (MOX, in mmol m$^{-2}$ d$^{-1}$) versus total suspend matter concentration (TSM, in mg L$^{-1}$) and chlorophyll-$a$**

 concentration (Chl-*a*, in µg L[-1]) in four urban ponds (Leybeek, Pêcheries, Tenreuken, and Silex) in the city of Brussels (Belgium)

 from January 2022 to December 2023. Linear regression shown as black solid line (Table S11).

Figure S15 compares the main fluxes of dissolved $CH_4$ in the water column: MOX, diffusive $CH_4$ emissions, bubble
dissolution that were derived from measurements, and the sedimentary diffusive $CH_4$ flux that was computed as a closing
term (assuming a steady state) for comparative purposes. The dissolution of bubbles was a significantly smaller input term of
dissolved $CH_4$ compared to the diffusive sedimentary flux that represented $88\pm18$ % of the total input of $CH_4$ to the water
column (Tukey's HSD test p<0.0001 in each pond). The low contribution of dissolution of bubbles resulted from the
shallowness of the studied ponds because bubble dissolution depends on the time spent by the bubble in the water column
during ascent, which is directly proportional to depth (McGinnis et al., 2006). MOX was a larger sink of dissolved $CH_4$ than
the diffusive $CH_4$ emission to the atmosphere in the four ponds, representing $80\pm19$ % and $80\pm14$ % of the total dissolved
$CH_4$ removal in the turbid-water Leybeek and Pêcheries ponds respectively (Tukey's HSD test p<0.0001 for the two ponds),
and $59\pm21$ % and $51\pm27$ % in the clear-water Tenreuken and Silex ponds respectively (Tukey's HSD test p=0.3429 for the
Tenreuken pond, and p=0.7634 for the Silex pond). For all four ponds, MOX accounted for $66\pm26$ % of the total $CH_4$
dissolved removal from the water column, in agreement with other studies in lentic systems (Kankaala et al., 2006;
Bastviken et al., 2008; Morana et al., 2020; Reis et al., 2022).
**3.7. Relative contribution of $CO_2$, $CH_4$ and $N_2O$ emissions**
The emissions in $CO_2$-eq for the 3 GHGs averaged per season for both 2022 and 2023 peaked seasonally in summer with 2.9
and 1.7 mg $CO_2$-eq m[-2] d[-1] in the Silex and the Tenreuken ponds, respectively, and 1.1 mg $CO_2$-eq m[-2] d[-1] in the Leybeek
pond (Fig. 13). The GHG fluxes peaked in fall in the Pêcheries pond, with 1.3 mg $CO_2$-eq m[-2] d[-1]. The higher value of the
total GHG emissions in fall compared to other seasons in the Pêcheries pond was due to an increase of $CO_2$ emissions in fall
that surpassed the peak of $CH_4$ emissions in summer. The GHG fluxes were the lowest in winter with 1.3 and 0.9 mg $CO_2$-eq
m[-2] d[-1] in the Silex and the Tenreuken ponds, respectively, and 0.8 and 0.6 mg $CO_2$-eq m[-2] d[-1] in the Pêcheries and the
Leybeek ponds, respectively. The relative contribution of ebullitive $CH_4$ fluxes peaked in summer in all four ponds, 73.8%
and 70.9% in the Silex and the Tenreuken ponds, respectively, and 23.6% and 58.3% in the Pêcheries and the Leybeek
ponds, respectively. The relative contribution of ebullitive $CH_4$ fluxes was lowest in winter with 22.1% and 10.0% in the
Silex and the Tenreuken ponds, respectively, and 6.7% and 1.0% in the Pêcheries and the Leybeek ponds, respectively.
The annual emissions in $CO_2$-eq of the three GHGs ($CO_2$, $CH_4$, and $N_2O$) in 2022 and 2023 were higher in the two clear-
water ponds ($1.3\pm0.5$ and $1.8\pm0.9$ mg $CO_2$-eq m[-2] d[-1] in the Tenreuken and Silex ponds, respectively) than in the two turbid-
water ponds ($1.0\pm0.2$ and $0.9\pm0.5$ mg $CO_2$-eq m[-2] d[-1] in the Leybeek and Pêcheries ponds, respectively) (Fig. 13) (Tukey's
HSD test p<0.0001 for Silex versus Pêcheries, p<0.0001 for Silex versus Leybeek, p=0.0107 for Tenreuken versus
Pêcheries, and p=0.0467 for Tenreuken versus Leybeek) due to higher total $CH_4$ emissions (diffusive+ebullitive) in clear-
water ponds ($0.7\pm0.4$ and $1.2\pm0.5$ mg $CO_2$-eq m[-2] d[-1] in the Tenreuken and Silex ponds, respectively) than in turbid-water
ponds ($0.2\pm0.2$ and $0.4\pm0.3$ mg $CO_2$-eq m[-2] d[-1] in the Leybeek and Pêcheries ponds, respectively) (Tukey's HSD test
p<0.0001 for Silex versus Pêcheries, p<0.0001 for Silex versus Leybeek, p=0.0005 for Tenreuken versus Pêcheries, and
p=0.0164 for Tenreuken versus Leybeek), as there were no significant differences between the four ponds for $CO_2$ emissions
in 2022 and 2023 (Tukey's HSD test p>0.05 for each comparison). $N_2O$ emissions were significantly lower in the Pêcheries
pond than the Leybeek and Silex ponds (Tukey's HSD test p=0.0012 for Pêcheries versus Leybeek, and p=0.0052 for
Pêcheries versus Silex). The contribution of $N_2O$ to the total GHG emissions was marginal and did not affect the differences
in total GHG fluxes between ponds, with the highest contribution observed in the Leybeek pond, with a contribution of

723 1.7%.

The majority of GHG emissions in $CO_2$-eq was related to $CO_2$ and $CH_4$ (diffusive+ebullitive) in the four ponds. In turbid-
water ponds $CO_2$ represented the largest fraction of GHG emissions (68.5% (2022) and 79.3% (2023) in the Pêcheries pond,
and 49.0% (2022) and 58.3% (2023) in the Leybeek pond). In clear-water ponds $CH_4$ represented the largest fraction of
GHG emissions (66.5% (2022) and 63.3% (2023) in the Silex pond, and 60.8% (2022) and 50.0% (2023) in the Tenreuken
pond). The higher annual GHG emissions in $CO_2$-eq from the two clear-water ponds than the turbid-water ponds were
related to the higher contribution of ebullitive $CH_4$ fluxes.

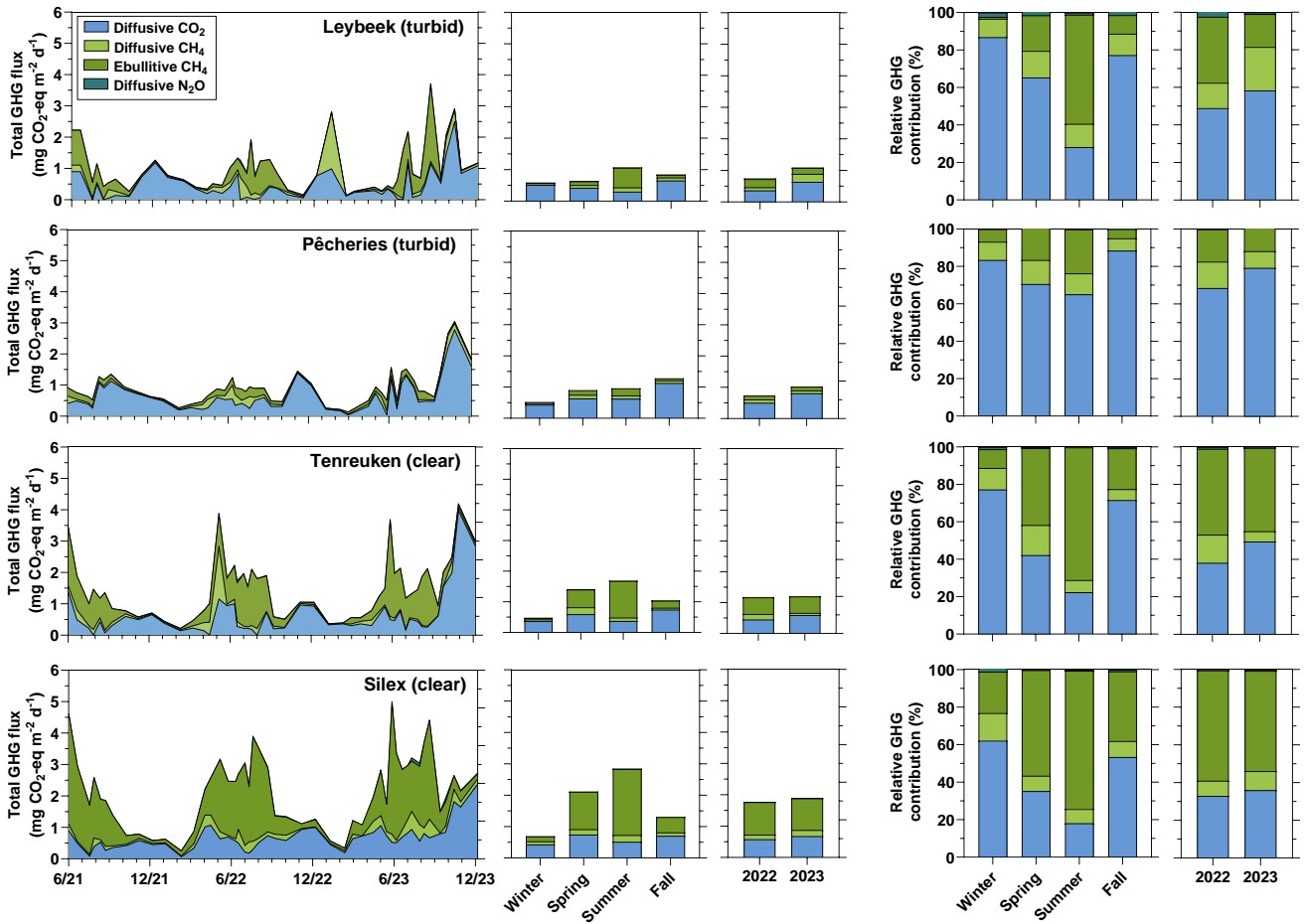


**Figure 13: Temporal evolution and relative contribution of emissions to the atmosphere of $CO_2$ (diffusive), $CH_4$ (diffusive and**
**ebullitive), and $N_2O$ (diffusive) expressed in $CO_2$ equivalents (in mg $CO_2$-eq m$^{-2}$ d$^{-1}$), in four urban ponds (Leybeek, Pêcheries,**
**Tenreuken, and Silex) in the city of Brussels (Belgium) from June 2021 to December 2023. Averages per season include data from**
**2021, 2022, and 2023. Year 2023 had a higher annual precipitation (1011 mm) than year 2022 (701 mm).**
The annual GHG fluxes increased from 2022 to 2023 due to an increase in relative contribution of $CO_2$ diffusive emissions
in all four ponds. Diffusive $CO_2$ emissions averaged annually in all four ponds 0.5 mg $CO_2$ m$^{-2}$ d$^{-1}$ in 2022 and 0.7 mg $CO_2$
m$^{-2}$ d$^{-1}$ in 2023. Diffusive $CO_2$ emissions were two times higher in summer 2023 than in summer 2022, and 2.5 times higher
in fall 2023 than in fall 2022, for similar values between 2023 and 2022 in spring and winter (1.1 higher and 1.1 lower,
respectively). Air temperatures were similar in both years (annual average of 12.2°C in 2022 and 12.1°C in 2023) with
winter, spring and summer marginally colder in 2023 than in 2022 (-0.5, -1.1°C and -0.4°C, respectively), and fall
marginally warmer in 2023 than 2022 (+0.6°C). Spring and summer were rainier in 2023 than 2022 (2.2 and 2.5 and times,
respectively) but fall and winter precipitations were relatively similar in both years (1.4 times wetter and 1.2 times drier in
2023 than 2022, respectively). Higher precipitations are likely to increase the inputs of organic and inorganic carbon from
soils to ponds by ground-waters, soil-waters, and surface runoff, as previously shown in other lentic systems (*e.g.* Marotta et

al., 2011; Holgerson, 2015). The highest seasonal increase of diffusive $CO_2$ emissions was observed in fall 2023. While this hypothesis is only based on the comparison of two years, the increase of the relative contribution of $CO_2$ diffusive emissions was observed in all four ponds which suggests a common uniform driver that would be consistent with a large variation weather such as annual precipitation. The El Niño event in 2023 induced low-level cyclonic wind anomalies and higher precipitation over Western Europe, including Belgium (Chen et al., 2024).

## 4. Conclusions

We found very marked differences in $CH_4$ dynamics between the two clear-water macrophyte-dominated ponds (Tenreuken and Silex) and the two turbid-water phytoplankton-dominated ponds (Pêcheries and Leybeek) of the city of Brussels. MOX was more important in the two turbid-water ponds compared to the clear-water ponds. MOX correlated to TSM and Chl-*a* concentrations possibly owing to a higher abundance of methanotrophs in the water column fixed to particles and/or an attenuation of light limitation of MOX. Ebullitive $CH_4$ emissions were higher in the two clear-water ponds than the two turbid-water ponds, possibly related to high availability of macrophyte organic matter. The annual diffusive $N_2O$ and $CO_2$ fluxes in 2022-2023 were not statistically different in the two clear-water ponds (Tenreuken and Silex) and in the two turbid-water ponds (Pêcheries and Leybeek). Other studies have found no difference in $N_2O$ sedimentary production in lakes with high and low density of submerged macrophytes. We hypothesize that in human impacted system such as the urban ponds in the city of Brussels, the strong range of variations of DIN was the main driver of $N_2O$ levels and over-rides other possible drivers such as presence or absence of macrophytes. Such a hypothesis was consistent with an overall positive relation between $\%N_2O$ and DIN in the urban ponds of the city of Brussels irrespective of presence or absence of macrophytes (Bauduin et al., 2024; this study). We hypothesize that $CO_2$ fluxes were relatively invariant among the four sampled ponds because of they were of similar size and depth, and that they were all relatively productive irrespective of whether from phytoplankton or submerged macrophytes.

The total (diffusive and ebullitive) $CH_4$ emissions represented $57.7\pm28.9$ % (ranging seasonally from 4.9 to 99.9%) of total GHG emissions in $CO_2$-eq in the two clear-water ponds compared to $41.0\pm28.7$ % (ranging seasonally from 2.8 to 99.9%) in the two turbid-water ponds. $CO_2$ represented nearly all the remainder of total GHG emissions in $CO_2$-eq, and $N_2O$ represented a very marginal fraction ($0.8\pm1.6$ %, ranging from 0.0% to 14.9%, with the maximum coinciding with minimal total $CO_2$-eq GHG flux in the Leybeek pond). The seasonal variations of GHG emissions were dominated by $CH_4$ ebullitive seasonal variations that peaked in summer (both quantitatively and relatively), as $CH_4$ ebullition was strongly related to temperature. The $pCO_2$ values in the four sampled ponds increased with precipitation at seasonal scale, probably in relation to higher inputs of organic and inorganic carbon by surface runoff. Years 2022 and 2023 were abnormally dry and wet, respectively, and consequently, the GHG emissions were higher in 2023 mainly due to an increase in the relative contribution of $CO_2$ emissions, probably in response to a strong El Niño event. This would suggest that variations of precipitation also affected year-to-year variations of $CO_2$ emissions in addition to partly regulating seasonal variations of $CO_2$ emissions from the four studied ponds.

**Data availability.** Timestamped and georeferenced data-set is available at 10.5281/zenodo.11103556.

**Author contributions.** AVB and NG conceived the study; TB collected field samples; TB and AVB made the laboratory analysis; TB and AVB jointly interpreted data and drafted the manuscript with substantial inputs from NG.

**Competing interests.** The authors declare that they have no conflict of interest.

**Acknowledgements.** We thank Ozan Efe (University of Liège) and Adriana Anzil (Université Libre de Bruxelles) for
analytical assistance, Florence Charlier (Université Libre de Bruxelles) for help in macrophyte identification and density
quantification (Table S1), Bruxelles Environnement for providing information on history of operations in the ponds (Table
S2), and Cédric Morana (University of Liège) for help and advice in setting up the Picarro G2201-i isotopic analyzer, two
anonymous reviewers for comments and suggestions on the initial manuscript.
**Financial support.** TB received funding from the Brussels-Capital Region's institute for the encouragement of scientific
research and innovation (Innoviris) as part of the Smartwater project (RBC/2020-EPF-6 h) and from the "Fonds pour la
formation à la Recherche dans l'Industrie et dans l'Agriculture" (FRIA, Belgium). The Picarro G2201-i isotopic analyzer
was funded by FRS-FNRS (U.N005.21). AVB is a Research Director at the FRS-FNRS.

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
