# Peer review of "Methane, carbon dioxide, and nitrous oxide emissions from two"

_EGUsphere, 2024_

## Author Comment (AC1)

This study focuses on intra- and interannual trends of GHG emissions from 4 small urban ponds and also addresses their underlying drivers. I believe that the study is a great addition to inland water studies, especially since long-term data on small pond GHG emissions is currently scarce. I commend the authors for curating such a comprehensive dataset. However, the current structure of the paper needs to be revised to make it easier for the reader to follow and also for possible final publications to Biogeosciences. Below are my brief and detailed comments for each subsection for possible improvement of the manuscript.

Reply: We warmly thank the reviewer for the positive evaluation of our work and for the detailed and useful comments for improvement.

Brief comments

Abstract and introduction

While the introduction was well compiled with the key motivation of the study, the end part did not include clear objectives and hypotheses for the work, which would have guided the reader better throughout the whole manuscript. I suggest that the authors include this and also adjust the intro to highly key gaps that would be addressed later on in the manuscript

Reply: The reviewer is right; we have revised the text providing objectives and working hypotheses at the end of the discussion. Text now reads L137: "We test the hypothesis that the two alternative states in shallow lakes (a clear-water state dominated by macrophytes, or a turbid-water state dominated by phytoplankton) drive differences in the $CO_2$, $CH_4$, and $N_2O$ dissolved concentration and diffusive emissions from the four studied artificial ponds, that have similar depth, surface area, and catchment urban coverage, and that mainly differ by the phytoplankton-macrophyte dominance. We also test the hypothesis that the two alternative states in shallow lakes drive differences in the ebullitive $CH_4$ emissions, water column MOX, and sedimentary methanogenesis pathway (acetoclastic or hydrogenotrophic) in the four studied ponds. The final objective of the present work is to determine the relative contribution of $CO_2$, $CH_4$, and $N_2O$ to the total GHG emissions in $CO_2$-eq and to test the hypothesis that the relative contribution of each GHG differs according to the two alternative states in shallow lakes."

Materials and methods

Although the description of the analysis of the GHG and associated parameters was well done, the statistical analysis part was too short and lacked enough detail. For example, the authors said they used a one-way ANOVA, yet they had a two-factor problem. i.e., seasonal and also pond-type influences. Also, what post hoc tests were used, what correlative analyses were used, and the main aims of this analysis are either lacking or not clearly stated. I suggest that more details should be added addressing the points above.

Reply: The reviewer is right; we have revised the text to provide additional information of the statistical treatment of the data. Text now reads L307: "Statistical analysis was conducted with R version 4.4.1. Pearson's linear correlation coefficients and the $r^2$ coefficient were used to assess relationships between log-transformed variables within each pond and across the dataset, to identify potential pond-specific and overall direct relationships between variables and GHGs. Statistical significance was determined using Fisher's F test and the associated $p$-value. This approach was also applied to study the relationships between $\delta^{13}C$-$CH_4$, FOX and MOX with Chl-$a$ and TSM. To assess the impact of Chl-$a$ concentration, macrophyte cover in summer, water depth, and lake surface area on diffusive and ebullitive $CH_4$ fluxes, the ratio of ebullitive $CH_4$ to total $CH_4$ flux, and $CO_2$ and $N_2O$ fluxes, both linear and quadratic relationships were applied to log-transformed averaged data. This approach allowed for the observation of trends between explanatory and dependent variables. For $N_2O$ fluxes, additional explanatory variables included $NO_2^-$, $NO_3^-$, $NH_4^+$, and DIN concentrations.

A two-way repeated measures analysis of variance (ANOVA) was used to test for differences in categorical variables, with the four seasons and the four ponds serving as independent factors, pond was set as a random effect to account for repeated measurements. A one-way repeated measures ANOVA was used to test for differences in $\delta^{13}C$-$CH_4$ from "perturbed sediments" with the four ponds serving as independent factors. After conducting an ANOVA and establishing significant differences among at least two groups (p<0.05), Tukey's Honestly Significant Difference (HSD) post-hoc test was employed to perform pairwise comparisons across all groups. Statistical outcomes are visually represented on boxplots, where upper- and lower-case letters are used to denote significant differences (p<0.05). Different lower- and upper-case letters indicate significant differences between groups."

Results and discussion

While combining the results and discussion is an acceptable practice, it sometimes leads to sectors of the manuscript that are not well described. For example, the trends of CO2 and N2O were only mentioned as results and not well substantiated by findings from other studies. In contrast, CH4 trends were well described by the authors, with proper discussions also related to other studies. I suggest either sticking to methane alone or giving also equal focus to the other GHGs.

Reply: The reviewer is right that there is more emphasis on $CH_4$ than $CO_2$ and $N_2O$ in the manuscript. This reflects the amount of collected data and variables that was different for each of the three GHGs. For $CO_2$ and $N_2O$ only dissolved concentrations were collected; for $CH_4$ additional variables were collected (ebullitive fluxes, $^{13}C/^{12}C$ ratios in water column, and in sedimentary bubbles). As such, we do not see how to provide an equivalent amount of discussion for each of the three GHGs. However, we think that $CO_2$ and $N_2O$ data bring invaluable added value to the manuscript because it allows us to quantify the relative importance of $CH_4$ emissions in $CO_2$ equivalents compared to the other two gases, in the context of the two alternative states, as well as at seasonal and inter-annual scales (former Fig. 14 = new Fig. 13). We feel this is a major outcome of our work, because there are relatively few studies that report simultaneously the diffusive emissions of three GHGs and even fewer with the addition of ebullitive $CH_4$ fluxes. So, we preferred to keep the data-set of the three GHGs in the paper.

But we followed the reviewer's suggestion, and we have expanded the description of the variations of $CO_2$ and $N_2O$ diffusive fluxes with new supplemental figures (Figs. S12, S13, S14). The corresponding text now reads L645: "The annual averaged diffusive fluxes of $CO_2$ ($F_{CO2}$) and $N_2O$ ($F_{N2O}$) in the four ponds in the city of Brussels were also plotted against Chl-*a* concentration, total macrophyte cover in summer, water depth, and lake surface area, as well as DIN for $N_2O$ fluxes (Figs. S12, S13, S14). Annual $F_{CO2}$ did not show significant differences between the four studied ponds (Tukey's HSD test: $p>0.05$ for each comparison), and $F_{CO2}$ did not significantly correlate to the other variables (Chl-*a* concentration, total macrophyte cover, water depth, and lake surface area). This might be surprising since other studies have reported lower $CO_2$ fluxes in more productive lentic systems (*e.g.* Sand-Jensen and Staehr 2007; Borges et al., 2022). We hypothesize that given that the four systems were either phytoplankton-dominated or macrophyte-dominated, in both cases, the ponds had an important submerged productivity resulting in a relatively invariant $F_{CO2}$ as function of either Chl-*a* or macrophyte cover. Annual mean $F_{CO2}$ was also uncorrelated to water depth and lake area (Fig. S12). This might have resulted from the relative similarity of depth and surface area of the four studied ponds, as it is well established that $CO_2$ emissions strongly increase with decreasing size of ponds (Holgerson and Raymond, 2016). Annual $F_{N2O}$ was not significantly different between clear-water and turbid-water ponds. $F_{N2O}$ was significantly lower in the slightly deeper Pêcheries pond than the two slightly shallower Leybeek and Silex ponds (Fig. S13) (Tukey's HSD test $p=0.0012$ for Pêcheries versus Leybeek, and $p=0.0052$ for Pêcheries versus Silex), and $F_{N2O}$ showed a significant negative relationship with water depth (Fig. S13). We hypothesize that this might reflect a larger dilution of $N_2O$ diffusing from sediments in the deeper systems. $F_{N2O}$ did not correlate to DIN, $NH_4^+$, $NO_2^-$, and $NO_3^-$ (Fig. S14). We hypothesize that this reflects the rather narrow range of annual DIN average values in the four studied ponds (~24 to ~29 μmol $L^{-1}$), as DIN, $NH_4^+$, $NO_2^-$, and $NO_3^-$ were not statistically different between ponds (Tukey's HSD test $p>0.05$ for every comparison)."

The manuscript also has 13 figures. While this is fine, readers may end up missing the most crucial part of the results. The unwritten rule of thumb is 6 to a maximum of 8 graphics, which include both tables and figures. I suggest the authors reevaluate the key figures guided by the objectives of the study and then reduce the current number and keep the rest in the suplimentary.

Reply: The reviewer is right that the manuscript has a large number of figures. *Biogeosciences* does not impose a size limit (text and figures) to submissions; the Associate Editors of *Biogeosciences* are required to carefully evaluate the manuscript before they are published in the *Biogeosciences Discussion* forum, which is an indication that for our present submission the Associate Editor decided that the length of text and number of figures were acceptable for an article in *Biogeosciences*. The larger than usual number of figures of our submission reflects the size of the data-set. However, we feel that the length of the manuscript and the number of figures of our submission justifies the scientific merit of keeping the data of the three GHGs in a single manuscript, rather than slicing the data-set into several papers and going down the path of "salami science". As mentioned above, there are relatively few studies that report simultaneously the diffusive emissions of the three GHGs and even fewer with the addition of ebullitive $CH_4$ fluxes. We feel this justifies the scientific merit of keeping this large data-set together in a single manuscript. As suggested by the reviewer, we have carefully reconsidered the usefulness and value of each of the figures, and we have moved former Figure 13 to the supplementals, reducing by one the number of figures in the main text. The other figures were kept because we feel they meet the objectives and hypotheses of the paper that have been added at the end of the Introduction.

Most of the results also lacked tests of significance and I suggest that this should be included in the revised drafts. If differences are not significant, its always acceptable to refer to them as trends

Reply: Most of differences presented in the previous version of the submitted manuscript had been tested, but the related statistical tests were in the supplementary tables and were only referenced in the legends of the figures. We have revised the manuscript to mentioned statistical significance in L357, L366, L368, L370, L371, L386, L388, L389, L413, L437, L464, L493, L553, L554, L576, L577, L578, L596, L598, L599, L611, L623, L625, L647, L657, L661, L672, L676, L694, L734, L735, L736, L737, L738, L739, L745, L767, L771, L772, L790, L793, L796, L797.

Conclusion

It needs to be focused on the objectives of the study and also to have a general outlook on the potential of urban ponds of inland water GHG dynamics.

Reply: We have now listed the objectives and hypotheses of the paper at the end of the Introduction section, and we feel that the Conclusions section addresses these objectives. We do not think that it would be appropriate to provide here, based on only four systems, a general outlook on the potential of urban ponds of inland water GHG emissions. This was to some extent addressed in a previous publication from our group based on a larger data-set in 22 ponds in the city of Brussels (but with more sparse temporal coverage) by a comparison of the urban ponds GHGs emissions with other emissions of GHGs from the city of Brussels (Bauduin et al. 2024; https://doi.org/10.1016/j.watres.2024.121257).Additionally, there are several synthesis papers that extensively address this issue, for example:

- Holgerson and Raymond (2016, https://doi.org/10.1038/ngeo2654)

- Peacock et al. (2021; https://doi.org/10.1111/gcb.15762)

- Deemer and Holgerson (2021, https://doi.org/10.1029/2019JG005600)

- Ray et al. (2023, https://doi.org/10.1002/lno.12362)

We prefer not to repeat the content of these papers, and we feel that the Conclusions reflect the main findings based on our data-set and addresses the objectives stated at the end of the Introduction section.

Detailed comments

Abstract and introduction

Line 15,16: Consider mentioning the direction of the relationship.

Reply: We have removed this sentence because this information was already given elsewhere in the abstract

Line 20: Consider adding a statement relating light availability in the clear ponds to enhance macrophyte growth. The current statement may be unclear at first read.

Reply: Text was modified and now reads L28: "Clear-water (macrophyte-dominated) ponds exhibited higher values of annual ebullitive $CH_4$ fluxes compared to turbid-water (phytoplankton-dominated) ponds, most probably in relation to the delivery to sediments of organic matter from macrophytes."

Line 21: Trim down the statement about pond methane fluxes, for instance, 'Pond methane fluxes to the atmosphere… '.

Reply: Text was modified and now reads L30: "At seasonal scale, $CH_4$ emissions exhibited a temperature dependence in all four ponds, with ebullitive $CH_4$ fluxes having a stronger dependence to temperature than diffusive $CH_4$ fluxes."

Line 37: Consider rephrasing: Greenhouse gas emissions from inland waters to the atmosphere….

Reply: Text was modified and now reads L50: "Greenhouse gas (GHG) emissions from inland water (rivers, lakes, and reservoirs) to the atmosphere such as carbon dioxide ($CO_2$), methane ($CH_4$) and nitrous oxide ($N_2O$) are quantitatively important for global budgets (Lauerwald et al., 2023)."

Line 39: GHG emissions from lakes….

Reply: Text was modified and now reads L52: "GHG emissions from lakes are lower than from rivers for $CO_2$ (Raymond et al., 2013) and $N_2O$ (Lauerwald et al., 2019; Maavara et al., 2019)."

Line 44: Remove the …. You can replace it with such as, which indicates that these are just examples and there could be more.

Reply: Text was modified and now reads L60: "The emissions of GHGs from artificial water bodies such as agricultural reservoirs, urban ponds, and storm-water retention basins could be higher than those from natural systems (Martinez-Cruz et al., 2017; Grinham et al., 2018; Herrero Ortega et al., 2019; Gorsky et al., 2019; Ollivier et al., 2019; Peacock et al., 2019, 2021; Webb et al., 2019; Bauduin et al., 2024)."

Line 46: Noun required after the word this…for example, this finding, this conclusion….check here and everywhere in the manuscript.

Reply: Text was modified and now reads here and elsewhere:

L63: "These higher emissions seem to result from higher external inputs of anthropogenic carbon and nitrogen in artificial systems such as rainfall runoff that brings organic matter and dissolved inorganic nitrogen (DIN), but might also reflect other differences compared to natural systems such as in hydrology (Clifford and Heffernan, 2018)."

L429: "The slopes of these correlations were not significantly different between ponds and were not correlated with surface area, depth, or dominance of type of primary producers (phytoplankton or macrophyte) (Table S6)."

L535: "The resulting calculated ebullitive $CH_4$ fluxes allowed to compare and integrate seasonally both components of $CH_4$ emissions to the atmosphere, and to calculate the relative contribution of ebullition to total (diffusive+ebullitive) $CH_4$ emissions."

L555: "This finding is consistent with other studies showing that ebullitive $CH_4$ fluxes can account for more than half of total $CH_4$ emissions in small and shallow lentic systems (*e.g.* Wik et al., 2013; Deemer and Holgerson, 2021; Ray and Holgerson, 2023; Rabaey and Cotner, 2024)."

L602: "The higher ebullitive $CH_4$ emissions from the clear-water ponds would suggest that the delivery of organic matter to sediments from macrophytes sustained a larger methane production than from phytoplankton."

L604: "This finding is consistent with the notion that vegetated littoral zones of lakes are hot spots of $CH_4$ production and emission (*e.g.* Hyvönen et al., 1998; Huttunen et al., 2003; Juutinen et al., 2003; Desrosiers et al., 2022)."

L615: "An increase in methane production with phytoplankton biomass in turbid-water ponds has also been reported by other studies in lakes (*e.g.* Yan et al., 2019; Bartosiewicz et al., 2021)."

L627: "These patterns are consistent with the idea of an increase of ebullition relative to diffusive $CH_4$ emissions in vegetated sediments compared to unvegetated sediments (e.g. Desrosiers et al., 2022; Ray et al., 2023; Theus et al., 2023)."

L639: "This hypothesis is consistent with the two clear-water ponds in Brussels having higher ebullitive fluxes than in the ponds compiled by Deemer and Holgerson (2021) at equivalent Chl-*a* values (Fig. S11)."

L695: "This pattern of $\delta^{13}C$-$CH_4$ of perturbed sediments could suggest a higher contribution of the hydrogenotrophic methanogenesis pathway compared to the acetoclastic pathway in the clear-water ponds where organic matter for methanogenesis was assumed to be mainly related to macrophytes rather than phytoplankton."

L865: "Years 2022 and 2023 were abnormally dry and wet, respectively, and consequently, the GHG emissions were higher in 2023 mainly due to an increase in the relative contribution of $CO_2$ emissions, probably in response to a strong El Niño event."

Line 52-54. The sentence on runoff comes from nowhere. Did you mean the rainfall runoff gets into the ponds? Consider revising it to make it clearer.

Reply: Text was modified and now reads L63: "These higher emissions seem to result from higher external inputs of anthropogenic carbon and nitrogen in artificial systems such as rainfall runoff that brings organic matter and dissolved inorganic nitrogen (DIN), but might also reflect other differences compared to natural systems such as in hydrology (Clifford and Heffernan, 2018)."

Materials and methods

Line 99. Did you mean the institute laboratory? The use of a home may imply a laboratory located in a personal house/apartment.

Reply: Text was modified and now reads L175: "A 2 liter polyethylene water container was filled with surface water for conditioning the samples for other variables at the laboratory in Université Libre de Bruxelles. "

Line 109. Consider revising from "consistent in" to "consisted of"

Reply: Text was modified and now reads L181: The bubble traps consisted of inverted polypropylene funnels (diameter 23.5 cm) mounted with 60 ml polypropylene syringes, with three way stop valves allowing to collect the gas without contamination from ambient air."

Line 111. The statement is a bit confusing. Consider revising it to make it clearer. How were the gases measured with 60ml syringes?

Reply: Text was modified and now reads L184: "The volume of gas collected in the funnels was sampled with graduated polypropylene 60 ml syringes every 24 hours. The value of the collected volume of gas was logged, and the gas was transferred immediately after collection to pre-evacuated 12 ml vials (Exetainers, Labco, UK) that were stored at ambient temperature protected from direct light prior to the analysis of $CH_4$ concentration and $\delta^{13}C$-$CH_4$ in the laboratory."

Line 113. Consider revising the statement. Did you mean that the measurements at Silex were of a longer frequency?

Reply: Text was modified and now reads L188: "The time-series of measurement were longer at the Silex pond than the other three ponds, because the Silex pond is closed to the public during the week, while the other three ponds are open to the public all the time."

Line 210-211. How were seasonality and pond type considered in your ANOVA analysis? The current statement is too short and lacks details. Was the ANOVA a repeated measures ANOVA as you sampled on the same pond multiple times?

Reply: The reviewer is right; we have expanded the description of the statistical tests, and text now reads L316: "A two-way repeated measures analysis of variance (ANOVA) was used to test for differences in categorical variables, with the four seasons and the four ponds serving as independent factors, pond was set as a random effect to account for repeated measurements. A one-way repeated measures ANOVA was used to test for differences in $\delta^{13}C$-$CH_4$ from "perturbed sediments" with the four ponds serving as independent factors. After conducting an ANOVA and establishing significant differences among at least two groups ($p<0.05$), Tukey's Honestly Significant Difference (HSD) post-hoc test was employed to perform pairwise comparisons across all groups. Statistical outcomes are visually represented on boxplots, where upper- and lower-case letters are used to denote significant differences ($p<0.05$). Different lower- and upper-case letters indicate significant differences between groups."

Results and discussion

Line 223. Wetter and colder

Reply: Text was modified and now reads L338: "Year 2021 had warmer and drier months in June and September, colder and wetter months in July and August, and was overall wetter and colder than 2022 (Fig. 2)."

Line 228-229. Consider adding the reference period at first mention and not at the end of the statement

Reply: Text was modified and now reads L345: "Figure 2: Temperature anomaly (difference between the average annual temperature and the normal annual temperature for the reference period 1991-2020 (11 °C), in °C) plotted against precipitation anomaly (ratio between annual precipitation and normal annual precipitation for the reference period 1991-2020 (837 mm), in %) from 2003 to 2023."

Line 233. Missing article; "with the silex pond"

Reply: Text was modified and now reads L353: "The four sampled ponds are situated in the periphery of the city of Brussels, with the Silex pond bordered by the Sonian Forest (Fig. 1)."

Line 244. In Figure 3, I suggest adding letters to the boxplots to indicate significant differences from the ANOVA test. This will help the reader quickly follow the graphs and avoid looking at an extra table in the supplementary.

Reply: The figures were modified accordingly.

Line 252. Are you reporting significant differences or trends? Check here and everywhere where you report comparisons of means. Also, indicate the level of significance as the information is currently missing in the results.

Reply: We have revised the manuscript to mentioned statistical significance in L357, L366, L368, L370, L371, L386, L388, L389, L413, L437, L464, L493, L553, L554, L576, L577, L578, L596, L598, L599, L611, L623, L625, L647, L657, L661, L672, L676, L694, L734, L735, L736, L737, L738, L739, L745, L767, L771, L772, L790, L793, L796, L797.

Line 253. I would move the explanations to the discussion, i.e., owing to primary production…

Reply: We preferred to keep the format of a joint "Results and Discussion" section. We have contemplated extensively how to present and discuss our data-set that is quite large and varied. We concluded that a joint "Results and Discussion" section was a better option than separated "Results" and "Discussion" sections. We feel that the text was articulated in a logical way that was relatively straightforward to follow by readers, going from relatively simple variables (meteorological and dissolved concentrations) to processes related to $CH_4$ dynamics (ebullition and MOX), and ending an integrative issue with the overall emissions in $CO_2$ equivalents.

Line 256. Consider using low instead of minimal

Reply: Text was modified and now reads L394: "Low values of $pCO_2$ were generally observed in spring and summer probably due to uptake of $CO_2$ by primary production from either phytoplankton or submerged macrophytes."

Line 257. Same comment as 253

Reply: We preferred to keep the format of a joint "Results and Discussion" section, refer to justification given above.

Line 258. Replace Maximal to High

Reply: Text was modified and now reads L396: "High values of $pCO_2$ were observed in fall in the four ponds and probably reflect the release of $CO_2$ from degradation of organic matter due to the senescence of phytoplankton or macrophytes (Fig. 3)."

Line 259-263. Correlation results are important for GHG process information as also discussed in this paragraph. I suggest moving them to the main text and, if possible, using scatterplots for the main relationships and indicating the correlation coefficients in the graphs. Also, always include the direction of the relationship, i.e., how was pco2 related to precipitation? Was it a negative or positive correlation?

Reply: We have added several new supplemental figures (Figs. S3, S4, S5, S6, and S7) showing the correlations for each pond and for the whole data set. Text was modified and now reads L403: "In all four ponds, $pCO_2$ strongly correlated positively to precipitation (Table S3; Figs S3, S4, S5, S6) suggesting a control of external inputs of carbon either as organic carbon sustaining internal degradation of organic matter or as soil $CO_2$ (*e.g.* Marotta et al., 2011)."

Line 264-266. I now see that the results and discussion are combined. While this is fine, the way it's currently written includes a lot of speculative statements that have not been substantiated by the findings of other studies. I suggest taking a closer look at all statements made and trying to support them with other studies. Putting a citation at the end without stating where the authors found similar results is also not encouraged. You can use (e.g., ……) in the citation to make clear that these authors found similar findings.

Reply: We agree and we have carefully revised the text to mention when relevant references with similar results and we have use "e.g." when relevant. Text was amended L114, L399, L405, L500, L507, L509, L517, L530, L551, L556, L581, L582, L605, L607, L618, L628, L643, L650, L834.

Line 269. See comment on line 253

Reply: We preferred to keep the format of a joint "Results and Discussion" section, refer to justification given above.

Line 277. See the comment on the use of "this" above

Reply: Please refer to the reply above, the text was modified in several places (including here) following the above comment.

Line 278. Add a comma between ponds and the

Reply: The comma was added.

Line 279. Were these differences based on the other factors also tested, i.e., the effect of the size of the pond? This analysis would validate the statement. At the moment, it's a bit speculative

Reply: We tested whether the slopes of the relationships differed between ponds (based on an analysis of covariance (ANCOVA)), and whether these slopes showed a relationship to surface area, depth, chl-*a* concentration and macrophyte cover (based on Pearson's linear correlation coefficients, with Fisher's F test and the associated p-value), and added the statistical results in a new supplementary Table S6. Text was modified and now reads L429: "The slopes of these correlations were not significantly different between ponds and were not correlated with surface area, depth, or dominance of type of primary producers (phytoplankton or macrophyte) (Table S6). These results suggest that the effect of precipitation on $pCO_2$ and the impact of temperature on dissolved $CH_4$ concentration outweigh other factors in explaining seasonal variations."

Line 282. Citation of figure or table needed here.

Reply: Text was modified and now reads L436: "The $\%N_2O$ values did not show significant seasonal variations in any of the four sampled ponds (ANOVA $F(3,174)=1,127$, $p=0.4091$) (Fig. 3). In individual ponds, $\%N_2O$ correlated negatively to temperature in the Tenreuken pond and Chl-*a* in the Silex pond, and positively to SRP in the Silex pond and TSM concentration in the Tenreuken pond (Table S3; Fig S3, S4)."

Line 284. Were surprising…

Reply: Text was modified and now reads L440: "The correlations with Chl-*a* and TSM were surprising since they were observed in the two clear-water ponds and might indirectly reflect seasonal variations (with minimal values of these two quantities in summer)."

Line 291-295. This paragraph is a bit confusing. I know what the authors mean, but I suggest it be rephrased in order to explain better the lack of correlation between N2O and DIN and its link to nitrogen deposition. How much is the nitrogen deposition in the region and how does it decrease from the edges of the city to the inner parts? Without this data, the current statement is somehow speculative

Reply: Following the reviewer's comment we have added a new supplemental figure showing the relation between $\%N_2O$ and DIN/atmospheric $NO_2$/distance from the city center (Fig. S8). We have also modified text that now reads L450: "A lower atmospheric nitrogen deposition in the periphery than in the city center is consistent with the correlation between $\%N_2O$ and atmospheric nitrogen dioxide ($NO_2$), and the correlation between $\%N_2O$ and the distance from the city center (Fig. S8)."

Line 297 -298. How do these bubble fluxes compare with other values from similar studies? Are they on the higher end or lower end? I suggest adding a few comparison studies in all fluxes reported to give an idea of where your study stands in terms of the magnitude of the fluxes.

Reply : We compared our data with those of Wik et al (2013), Delsontro et al (2016), and Ray and Holgerson (2023), and we have modified text that now reads L465: "The bubble flux values in the four sampled ponds in the city of Brussels were within the range of values reported in lentic systems of equivalent size by Wik et al. (2013) (0 to 2772 mL m$^{-2}$ d$^{-1}$), Delsontro et al. (2016) (11 to 748 mL m$^{-2}$ d$^{-1}$) and Ray and Holgerson (2023) (0 to 2079 mL m$^{-2}$ d$^{-1}$). The mean $CH_4$ content of the bubbles in the four sampled ponds in the city of Brussels was $31\pm21\%$, and comparable to the values obtained by Wik et al. (2013) ($35\pm25\%$), Delsontro et al. (2016) ($58\pm25\%$) and Ray and Holgerson (2023) ($25\pm13\%$) in lentic systems of equivalent size."

Line 304. I suggest adding the equation of the fit on the graph.

Reply: We preferred not to overcrowd the figure (and for consistency we should have added equations in all the figures). But following the reviewer's comment, we have added the equation in the legend of the figure so that it is easy to access by the readers. Text now reads L477: "Figure 4: Bubble flux (ml m$^{-2}$ d$^{-1}$) as a function of water temperature (°C) and the relative $CH_4$ content in bubbles ($\%CH_4$, in %) in four urban ponds (Leybeek, Pêcheries, Tenreuken, and Silex) in the city of Brussels (Belgium). Bubbles fluxes were measured with three bubble traps in

spring, summer, and fall of 2022 and 2023, totalling 8 days in the Leybeek, Pêcheries, and Tenreuken ponds and 24 days in the Silex pond. Given the shallowness of the sampled systems (<1.5 m, Fig. 1) we assume that sediments experience the same temperature as surface waters. Solid lines represent exponential regression fit of bubble flux as function of temperature ($Y = 28 \cdot e^{0.14 \cdot X}$, n=139), and as function of relative $CH_4$ content in the bubbles ($Y = 164 \cdot e^{0.0.3 \cdot X}$, n=123) (Table S11)."

310-312. This is what I mean by referencing of other studies to support your findings/

Reply: We agree with the reviewer and we have referenced other studies elsewhere in the manuscript (L390, L407, L435, L465, L470, L507, L530, L551).

Line 337. I suggest always using, e.g.,…. Or "similar to what was found…" for every citation quoted in the discussion, particularly those that involve specific findings. This form of citing guides the reader better

Reply: Text was amended L114, L399, L405, L500, L507, L509, L517, L530, L551, L556, L581, L582, L605, L607, L618, L628, L643, L650, L834.

Line 338. I suggest adding the equations of the relationships to Figure 6, which may be useful for future comparisons with other studies and also allow them to be potentially used to estimate ebullition methane fluxes where temperature data is available, as this study has done.

Reply: We preferred not to overcrowd the figure (and for consistency we should have added equations in all of the figures). But following the reviewer's comment, we have added the equation in the legend of the figure so that it is easy accessed by the readers. Text now reads L519: "Figure 6: Measured ebullitive $CH_4$ fluxes (mmol m$^{-2}$ d$^{-1}$) as function of water temperature (°C) in four urban ponds (Leybeek, Pêcheries, Tenreuken, and Silex) in the city of Brussels (Belgium), in spring, summer, and fall of 2022 and 2023, totalling 8 days in the Leybeek, Pêcheries, and Tenreuken ponds and 24 days in the Silex pond, with three bubble traps. Solid lines represent exponential fit for the Leybeek ($Y = 0.01 \cdot e^{0.32 \cdot X}$, n=22), Pêcheries ($Y = 0.16 \cdot e^{0.15 \cdot X}$, n=22), Tenreuken ($Y = 0.10 \cdot e^{0.23 \cdot X}$, n=19), Silex ($Y = 0.54 \cdot e^{0.18 \cdot X}$, n=72) ponds (Table S7) dashed lines represent exponential fit established in similar systems: four small ponds in Québec ($Y = 0.06 \cdot e^{0.25 \cdot X}$) (DelSontro et al., 2016) and a small urban pond in the Netherlands ($Y = 0.51 \cdot e^{0.17 \cdot X}$) (Aben et al., 2017). Each exponential curve allows to determine a $Q_{10}$ of $CH_4$ ebullition, plotted against water depth, solid line represents exponential regression fit ($Y = 92 \cdot e^{-0.02 \cdot X}, n = 6$) (Table S11)."

Line 356. Add letters from posthoc tests to indicate seasonal differences to this figure, similar to my comment on Figure 3

Reply: The figures were modified accordingly.

Line 373. Than for diffusive fluxes…

Reply: Text was modified and now reads L562: "Other studies have also reported higher $Q_{10}$ for $CH_4$ ebullition than for $CH_4$ diffusion in lentic systems (DelSontro et al., 2016; Xun et al., 2024)."

Line 389. How do you explain the polynomial U fit in the first panel?

Reply: The cause of the U shape is discussed further down in the text.

Line 393. Was there a statistical test to show that the differences were significant? Judging by the error bars, which sometimes overlap, it may be that the differences were not significant, but I do agree that the trends are there. In cases where the relationships are not significant, I suggest sticking to … showed trends of being higher in…even though the difference was not significant.

Reply: The error bars reflect the seasonal variability. In order to analyze the differences among the 4 ponds as function of size and phyto/macrophyte dominance it is required to average the data and remove the seasonal variability. For transparency we have nevertheless shown the error bars.

Line 401. I now see the explanation for the polynomial fit, which also makes sense. However, this may not be so clear at first read. Hence, it may help to reference the result first and then link it to the explanation. Also, has the relationship with phytoplankton been found in other turbid pond studies. The current reference talks about lakes.

Reply: We feel that the text is structured conventionally were the results are presented first and the explanations given after. We have added comparison with other turbid ponds and text reads L615: "An increase in methane

production with phytoplankton biomass in turbid-water ponds has also been reported by other studies in lakes (*e.g.* Yan et al., 2019; Bartosiewicz et al., 2021).”

Line 411. Where is the regression done in the results?

Reply: The figure was modified and now includes the results of the regression.

Line 421-423. Lakes and ponds are used as synonyms here, even thou they may have different characteristics. Check here and everywhere to ensure that references made on lakes are assumed to be related also to ponds.

Reply: We have modified text that now reads L642: “Consequently, global scaling of $CH_4$ fluxes in lentic systems using Chl-*a* as a predictor as used in lakes (*e.g.* DelSontro et al., 2018) might under-estimate ebullitive $CH_4$ emissions due to a misrepresentation of macrophyte-dominated clear-water ponds.”

Line 472. “as” not “than”

Reply: We have replaced “than” by “as” (L731).

Line 485. Modify the graph to include posthoc tests showing significance across ponds and seasons

Reply: The figures were modified accordingly.

Conclusion

Line 557-562. Reads more like the results and discussion part. I suggest rewriting the conclusion part to focus more on what were the objectives, what were the main conclusions from each objective, and finally future perspectives on what can be done better.

Reply: We have now listed the objectives and hypotheses of the paper at the end of the Introduction section and we feel that the Conclusions section focusses on these objectives.

**Figure S3: Partial pressure of $CO_2$ (pCO$_2$, ppm), dissolved $CH_4$ concentration ($CH_4$, nmol L$^{-1}$) and N$_2$O saturation level (%N$_2$O, %), versus water temperature (°C), oxygen saturation level (%O$_2$, %), concentration of soluble reactive phosphorus (SRP, μmol L$^{-1}$), concentration of dissolved inorganic nitrogen (DIN= $NH_4^+$ + $NO_2^-$ + $NO_3^-$ , μmol L$^{-1}$), concentration of chlorophyll-*a* (Chl-*a*, μg L$^{-1}$), and total suspended matter (TSM, mg L$^{-1}$) in the clear-water Tenreuken pond in Brussels, sampled from June 2021 to December 2023. Coefficient of determination, r$^2$, and associated *p*-value are indicated in boxes and solid lines indicate significative linear regression lines of the log-transformed data (*p*-value < 0.05).**

[Figure]

**Figure S4:** Partial pressure of $CO_2$ (p$CO_2$, ppm), dissolved $CH_4$ concentration ($CH_4$, nmol $L^{-1}$) and $N_2O$ saturation level (%$N_2O$, %), versus water temperature (°C), oxygen saturation level (%$O_2$, %), concentration of soluble reactive phosphorus (SRP, µmol $L^{-1}$), concentration of dissolved inorganic nitrogen (DIN= $NH_4^+ + NO_2^- + NO_3^-$ , µmol $L^{-1}$), concentration of chlorophyll-*a* (Chl-*a*, µg $L^{-1}$), and total suspended matter (TSM, mg $L^{-1}$) in the clear-water Silex pond in Brussels, sampled from June 2021 to December 2023. Coefficient of determination, $r^2$, and associated *p*-value are indicated in boxes and solid lines indicate significative linear regression lines of the log-transformed data (*p*-value < 0.05).

[Figure]

**Figure S5: Partial pressure of CO$_2$ (pCO$_2$, ppm), dissolved CH$_4$ concentration (CH$_4$, nmol L$^{-1}$) and N$_2$O saturation level (%N$_2$O, %), versus water temperature (°C), oxygen saturation level (%O$_2$, %), concentration of soluble reactive phosphorus (SRP, µmol L$^{-1}$), concentration of dissolved inorganic nitrogen (DIN= NH$_4^+$ + NO$_2^-$ + NO$_3^-$ , µmol L$^{-1}$), concentration of chlorophyll-*a* (Chl-*a*, µg L$^{-1}$), and total suspended matter (TSM, mg L$^{-1}$) in the turbid-water Leybeek pond in Brussels, sampled from June 2021 to December 2023. Coefficient of determination, r$^2$, and associated *p*-value are indicated in boxes and solid lines indicate significative linear regression lines of the log-transformed data (*p*-value < 0.05).**

[Figure]

**Figure S6: Partial pressure of CO$_2$ (pCO$_2$, ppm), dissolved CH$_4$ concentration (CH$_4$, nmol L$^{-1}$) and N$_2$O saturation level (%N$_2$O, %), versus water temperature (°C), oxygen saturation level (%O$_2$, %), concentration of soluble reactive phosphorus (SRP, µmol L$^{-1}$), concentration of dissolved inorganic nitrogen (DIN= NH$_4^+$ + NO$_2^-$ + NO$_3^-$ , µmol L$^{-1}$), concentration of chlorophyll-*a* (Chl-*a*, µg L$^{-1}$), and total suspended matter (TSM, mg L$^{-1}$) in the turbid-water Pêcheries pond in Brussels, sampled from June 2021 to December 2023. Coefficient of determination, r$^2$, and associated *p*-value are indicated in boxes and solid lines indicate significative linear regression lines of the log-transformed data (*p*-value < 0.05).**

[Figure]

**Figure S7: Partial pressure of CO₂ (pCO₂, ppm), dissolved CH₄ concentration (CH₄, nmol L⁻¹) and N₂O saturation level (%N₂O, %), versus water temperature (°C), oxygen saturation level (%O₂, %), concentration of soluble reactive phosphorus (SRP, μmol L⁻¹), concentration of dissolved inorganic nitrogen (DIN= NH₄⁺ + NO₂⁻ + NO₃⁻ , μmol L⁻¹), concentration of chlorophyll-*a* (Chl-*a*, μg L⁻¹), and total suspended matter (TSM, mg L⁻¹) in four ponds in Brussels sampled from June 2021 to December 2023 (Leybeek, Pêcheries, Tenreuken, and Silex). Coefficient of determination, r², and associated *p*-value are indicated in boxes and solid lines indicate significative linear regressions lines of the log-transformed data (*p*-value < 0.05).**

[Figure]

**Figure S8: Mean diffusive N₂O saturation level (%N₂O, %) versus dissolved inorganic nitrogen (DIN= NH₄⁺ + NO₂⁻ + NO₃⁻, in µmol L⁻¹), distance of the pond from city center (km), and atmospheric NO₂ (µg m⁻³) in four ponds in Brussels sampled from June 2021 to December 2023 (Leybeek, Pêcheries, Tenreuken, and Silex), and in other ponds in the city of Brussels sampled in 2021 and 2022 from Bauduin et al. (2024). The atmospheric NO₂ concentration was extracted from the Curieuzenair initiative which analyzed 2483 air samples in September 2021 covering the whole of the city of Brussels with a homogeneous distribution (https://curieuzenair.brussels/en/the-results/). Coefficient of determination, r², and associated *p*-value for data from Bauduin et al. (2024) are indicated in boxes and solid lines indicate significative linear regression lines of the log-transformed data (*p*-value < 0.05).**

[Figure]

**Figure S12: Mean annual diffusive CO₂ flux (F_CO2 in mmol m⁻² d⁻¹) versus chlorophyll-*a* (Chl-*a*, µg L⁻¹), total macrophyte cover in summer (%), water depth (cm), and lake surface area (ha) in four ponds (Leybeek, Pêcheries, Tenreuken, and Silex) in the city of Brussels (Belgium) from June 2021 to December 2023. Error bars indicate the standard deviation.**

[Figure]

**Figure S13: Mean annual N₂O flux (F_N2O, µmol m⁻² d⁻¹) versus chlorophyll-*a* (Chl-*a*, µg L⁻¹), total macrophyte cover in summer (%), water depth (cm), and lake surface area (ha) in four ponds (Leybeek, Pêcheries, Tenreuken, and Silex) in the city of Brussels (Belgium) from June 2021 to December 2023. Error bars indicate the standard deviation. Solid lines indicate trends in relationship between variables.**

[Figure]

**Figure S14: Mean annual diffusive N₂O flux (F_N2O, µmol m⁻² d⁻¹) versus dissolved inorganic nitrogen (DIN= NH₄⁺ + NO₂⁻ + NO₃⁻, µmol L⁻¹), ammonium (NH₄⁺, µmol L⁻¹), nitrite (NO₂⁻, µmol L⁻¹) and nitrate (NO₃⁻, µmol L⁻¹) in four ponds (Leybeek, Pêcheries, Tenreuken, and Silex) in the city of Brussels (Belgium) from June 2021 to December 2023.**

[Figure]

---

## Author Comment (AC2)

General comments

Authors assessed various aspects of GHG dynamics in four urban ponds of either macrophyte or phytoplankton dominated stable states over a 2.5 year period. The authors have produced an impressive and valuable long-term dataset on greenhouse gas (GHG) dynamics in ponds, which notably includes the often-overlooked ebullitive flux and provides insight into various methane pathways. While the data and results are impactful, the manuscript requires major revisions to be considered for publication. First, the writing needs major improvement for clarity and quality, and second, the authors provide insufficient details about methods and statistics, and should reconsider how their gases are presented. Below are my general comments for each section, followed by more specific line comments. I believe these data can make a good contribution to the body of literature on GHG dynamics in urban ponds, and hope these comments will help improve the manuscript.

Reply: We warmly thank the reviewer for the positive evaluation of our work and for the detailed and useful comments for improvement.

Abstract

The abstract was long and wordy with the results. It may be helpful to be more concise and summarize some of these major findings. I provide examples in specific comments below.

Reply: Biogeosciences does not impose a word limit to abstracts, and the length of the abstract reflects the extensive data-set and density of results. Following, the reviewer's comments we have reduced the size of the abstract by focusing on the impactful results, and have considered the various comments described below by the reviewer.

Introduction

The introduction has some good elements to it but overall lacks supporting information for much of the content covered in the study. For example, authors compare macrophyte versus phytoplankton dominated systems but only provide background information on the impact of macrophytes to GHGs. One of the most interesting components of the study to me is methanogenic pathways and methane oxidation, which has not been covered in urban ponds to my knowledge. While authors cover GHG fluxes and drivers, no mention of methane oxidation, methanogenic pathways, and their significance are made in the intro and is only briefly mentioned in the concluding paragraph. I'd like to see some supporting information for phytoplankton dominated ponds, methane oxidation, and methanogenic pathways in the intro.

Reply: We have completed the introduction to take account of the reviewer's comments. We have added general information on the impact of phytoplankton on GHGs that now reads L82: "In phytoplankton-dominated lakes, $CO_2$ concentrations depend in part on the development stage of the phytoplankton, with the growth and peak phases generally coinciding with lower $CO_2$ concentrations due to intense photosynthesis (Grasset et al., 2020; Vachon et al., 2020). $CH_4$ emissions have been reported to increase with the concentration of chlorophyll-*a* (Chl-*a*) in phytoplankton-dominated lakes (DelSontro et al., 2018; Borges et al., 2022)." and L93: "The production of $N_2O$ predominantly occurs through microbial nitrification and denitrification that depend on DIN and $O_2$ levels (Codispoti and Christensen, 1985; Mengis et al., 1997). Competition for DIN between primary producers and $N_2O$-producing microorganisms can impact $N_2O$ production. Additionally, the transfer of labile phytoplankton organic matter to sediments fuels benthic denitrification. Combined, these two processes could explain that some lakes can act as sinks of $N_2O$ under elevated Chl-*a* concentrations (Webb et al., 2019; Borges et al., 2022)."

We have also added a paragraph on the different pathways of methanogenesis in sediments and methanotrophy in the water column at L119: "The two primary metabolic pathways for $CH_4$ production in sediments by methanogenic archaea are the fermentation of acetate (acetoclastic pathway) and the reduction of carbon dioxide by $H_2$ (hydrogenotrophic pathway) (Whiticar et al., 1986; Conrad, 1989). $CH_4$ produced by these two pathways exhibits distinct $^{13}C/^{12}C$ ratios ($\delta^{13}C$-$CH_4$) (Whiticar et al., 1986) and can be used to discriminate which pathway is dominant. When $CH_4$ diffuses from the sediment to the water column, it can be oxidized by methanotrophic bacteria who preferentially consume $CH_4$ with $^{12}C$ over $^{13}C$, resulting in an increase of $\delta^{13}C$-$CH_4$ of the residual $CH_4$ in the water column (Bastviken et al., 2002). Fractionation models then allow estimating methane oxidation (MOX) from measurements of $\delta^{13}C$-$CH_4$ of dissolved $CH_4$ in the water column. Bastviken et al. (2008) report that

30 to 99% of the $CH_4$ produced in sediments of freshwater lakes can be removed by MOX that is as a significant $CH_4$ sink in these water bodies. MOX is known to be inhibited by light (Dumestre et al., 1998) and increases with the presence suspended particles (Abril et al., 2007) so that MOX might vary between clear and turbid waters (Morana et al., 2020)."

In the closing paragraph the authors do not include any objectives or predictions/hypotheses, but rather focus on some of the methods. I strongly suggest focusing less on methods and including objectives and predictions/hypotheses to help guide readers.

Reply: The reviewer is right, and we have added the objectives and working hypothesis at the conclusion of the introduction. Text now reads L137: "We test the hypothesis that the two alternative states in shallow lakes (a clear-water state dominated by macrophytes, or a turbid-water state dominated by phytoplankton) drive differences in the $CO_2$, $CH_4$, and $N_2O$ dissolved concentration and diffusive emissions from the four studied artificial ponds, that have similar depth, surface area, and catchment urban coverage, and that mainly differ by the phytoplankton-macrophyte dominance. We also test the hypothesis that the two alternative states in shallow lakes drive differences in the ebullitive $CH_4$ emissions, water column MOX, and sedimentary methanogenesis pathway (acetoclastic or hydrogenotrophic) in the four studied ponds. The final objective of the present work is to determine the relative contribution of $CO_2$, $CH_4$, and $N_2O$ to the total GHG emissions in $CO_2$-eq and to test the hypothesis that the relative contribution of each GHG differs according to the two alternative states in shallow lakes."

Methods

The methods section requires some reorganization and lacks a lot of details.

The statistics section is grossly lacking in detail and what methods were used appear concerning, but potentially due to no explanation of the approaches used. Authors need to explicitly state when/why they use one-way ANOVAs (we use one-way ANOVAs to test for the effect of X on Y and the effect of A on B) and linear or exponential regression. Exponential regressions were used but no mention was made in methods. I also don't think that one-way ANOVA is appropriate where it is used. First, if your analyses are including all data over time, and there are repeated observations from the same four sites over time, that is a case of pseudoreplication. I suggest looking into generalized linear mixed effects models (GLMM) to account for time as a repeated measure. Second, as an example looking at Table S6 (a bit hard to interpret), I think what I'm seeing are pairwise comparisons for one-way ANOVAs that looked at the effect of pond and season on a variable listed in a column? (i.e., chl-a ~ pond + season)? If so, I think these are instead two-way ANOVAs, and the type of pairwise comparison needs to be stated. Table S3 suggest PERMANOVA was used but again, this is not clear. Stats for pairwise comparisons are provided but nothing for the model itself. Degrees of freedom would be a helpful term to report along with the model stats, not just pairwise stats.

Reply: The reviewer is right that the statistics section lacked detail on our methods. We first examined the relationships between environmental variables and GHGs on a pond-by-pond basis, then looked for general relationships across the dataset using Pearson's coefficient and $R^2$ on log-transformed data. We've updated the statistical table to include exact p-values and degrees of freedom and added supplemental figures to visualize these relationships. After looking at the relationships between variables, we looked at the differences between ponds, and in particular between clear and turbid ponds. To do so, we performed a two-way repeated measures ANOVA with Tukey's HSD post-hoc tests to compare measurements between ponds and seasons, addressing pseudo-replication with the repeated measures. Graphs now display significances, and we've revised the statistical tables for clarity, presenting ANOVA results followed by Tukey HSD tests, and adding degree of freedom in the tables.

We performed LMMs, setting ponds as random effects and environmental variables as fixed effects, following Ray and Holgerson (2023), testing all combinations of different models and keeping only the best model for each GHG. The results are similar to those we obtain with Pearson coefficients. However, we lose some information we find relevant, in particular relationships intrinsic to the ponds, such as $pCO_2$ explained by Chl-*a* in a turbid pond, or the positive relationship with DIN and SRP in some ponds, which demonstrate the general control of

CO$_2$ by biological activity. We therefore prefer to keep the explanation of GHG relationships using Pearson coefficients.

We added information on the statistical methods used in the material and methods section. Text now reads L307: "Statistical analysis was conducted with R version 4.4.1. Pearson's linear correlation coefficients and the r² coefficient were used to assess relationships between log-transformed variables within each pond and across the dataset, to identify potential pond-specific and overall direct relationships between variables and GHGs. Statistical significance was determined using Fisher's F test and the associated *p*-value. This approach was also applied to study the relationships between δ$^{13}$C-CH$_4$, FOX and MOX with Chl-*a* and TSM. To assess the impact of Chl-*a* concentration, macrophyte cover in summer, water depth, and lake surface area on diffusive and ebullitive CH$_4$ fluxes, the ratio of ebullitive CH$_4$ to total CH$_4$ flux, and CO$_2$ and N$_2$O fluxes, both linear and quadratic relationships were applied to log-transformed averaged data. This approach allowed for the observation of trends between explanatory and dependent variables. For N$_2$O fluxes, additional explanatory variables included NO$_2^-$, NO$_3^-$, NH$_4^+$, and DIN concentrations.

A two-way repeated measures analysis of variance (ANOVA) was used to test for differences in categorical variables, with the four seasons and the four ponds serving as independent factors, pond was set as a random effect to account for repeated measurements. A one-way repeated measures ANOVA was used to test for differences in δ$^{13}$C-CH$_4$ from "perturbed sediments" with the four ponds serving as independent factors. After conducting an ANOVA and establishing significant differences among at least two groups (p<0.05), Tukey's Honestly Significant Difference (HSD) post-hoc test was employed to perform pairwise comparisons across all groups. Statistical outcomes are visually represented on boxplots, where upper- and lower-case letters are used to denote significant differences (p<0.05). Different lower- and upper-case letters indicate significant differences between groups."

On another note, I can understand using ANOVA to see significant differences between sites for water chemistry variables (chl-a, TSM, %O2), but later on (e.g., Figure 8, Figure 10, Figure 12) linear regressions are used to test for effects of environmental variables on GHGs, which is redundant. While I don't think it is wrong to use multiple individual regressions to test for the effect of each environmental variable on a gas, multiple regression models may be more informative, and again, you can account for the pseudoreplication of repeated measures over time. Last suggestion here, the point of the paper is to look at differences between macrophyte versus phytoplankton dominated ponds. Have authors tried grouping by stable state type (macrophyte or phytoplankton), and testing for the effect of that? Two sites per level might not be sufficient enough but curious if this was considered. If you went this route and used a GLMM (for example), perhaps individual site could be set as the random effect.

Reply: We believe that the results are not redundant, and we preferred to keep the figures 8, 10 and 12 as they are. The purpose of Figure 8 is to highlight that it is either the macrophytes, the phytoplankton, or a combination of both that explain the variations in diffusive and ebullitive CH$_4$ fluxes, and the figure presents this concisely. It also shows that the usual predictors of CH$_4$ fluxes (surface area and depth) do not explain CH$_4$ fluxes in the four studied ponds. Figure 10 has been revised to present the differences between the isotopic signatures of CH$_4$ of perturbed sediments in the form of boxplots. Figure 12 demonstrates that methanotrophy increases along a turbidity gradient with increasing Chl-*a* and TSM, and presenting the results in this format effectively illustrates our points and supports the text.

Further, I have concerns for how gas concentrations/quantities are presented and equilibrium saturation is calculated. Why present CO2 as a partial pressure, methane as a concentration, and N2O as a percent saturation compared to equilibrium?? I strongly advice all three gases be reported in comparable molar units. In addition to molar concentration, authors should report their deviation from equilibrium, either as a % (like N2O. but values will likely be too high for CH4), or the factor of super/under saturation, and be consistent for all gases (i.e., the concentration of CO2 was X umol/L and 12-fold supersaturated compared to equilibrium). Deviations from equilibrium are more informative for biological changes to gas concentrations. Alternatively, there is so much information in this manuscript that authors should consider removing results for gas concentrations altogether and focus on air-water fluxes, as some seasonal patterns and environmental drivers appear somewhat similar between concentrations and gases.

**Reply:** These units are used in topical literature. pCO2 is usually expressed in ppm and readers can easily determine if the values are above or below atmospheric CO2 of about 400 ppm. In topical literature CH4 is reported in μmol/L or nmol/L, and the values were systematically above saturation. For $N_2O$, given that the values oscillate around saturation, we used percentage of saturation.

But we agree that the mix of different units might be confusing, although used frequently in the publications from our group:

- Borges AV et al. (2019) Biogeosciences, 16, 3801-3834, https://doi.org/10.5194/bg-16-3801-2019
- Borges AV et al. (2022) Sci Adv 8, eabi8716, 1-17, https://doi.org/10.1126/sciadv.abi8716
- Borges AV et al. (2015) Nat Geosci 8, 637-642, https://doi.org/10.1038/NGEO2486
- Borges AV et al. (2023) J Great Lakes Res 49, 229-245, https://doi.org/10.1016/j.jglr.2022.11.010
- Chiriboga G & **AV Borges** (2023) Communications Earth & Environment, 4, 76 https://doi.org/10.1038/s43247-023-00745-1
- Bauduin et al. (2024) Water Research, 253, 121257. https://doi.org/10.1016/j.watres.2024.121257
- Chiriboga, G et al. (2024) Aquatic Sciences, 86(2), 24. https://doi.org/10.1007/s00027-023-01039-6

We recently published a companion paper on GHGs dynamics in Brussels' urban ponds based on an independent data-set using units of $CO_2$, $CH_4$ and $N_2O$ used here (Bauduin et al. 2024; https://doi.org/10.1016/j.watres.2024.121257); if readers want to compare the results and conclusions from both papers, it is preferable that the units are consistent.

Please note that the full data-set is publically available, so the readers can re-use the data in their preferred units.

Results/discussion

This sounds more like a results section than a combined results and discussion section. I recommend keeping results and discussion separate. I provide specific comments up to some of the results, as I imagine results reporting will change when statistics are improved, and discussion will change when these sections are split apart.

Reply: We opted to keep a unified "Results and Discussion" section after thoroughly evaluating how to best present our extensive and varied dataset. We found that integrating the results and discussion into one section was more effective than separating them. This approach allows us to present the data in a logical sequence, beginning with basic variables (meteorological data and dissolved concentrations), advancing to $CH_4$ dynamics (ebullition and MOX), and concluding with a comprehensive analysis of total emissions in $CO_2$ equivalents. We believe this format provides a coherent and accessible narrative for readers. The results section has not been changed, but the related figures and text have been revised to address the reviewer's specific comments regarding the redundancy of certain sentences and the results presented in some figures.

Conclusion

Major results should be broadly summarized here but the significance of the work should also be included. If authors include predictions, they can be circled back on here as well.

Reply: We have now listed the objectives and hypotheses of the paper at the end of the "Introduction" section, and we feel that the Conclusions section addresses these objectives.

Specific comments

ABSTRACT

Line 8-9: Suggest rewording as "…but it is unclear if these two states affect the emission of greenhouse gases carbon dioxide (CO2), methane (CH4), and nitrous oxide (N2O) to the atmosphere."

Reply: Text was modified and now reads L7: "Shallow ponds can occur either in a clear-water state dominated by macrophytes or a turbid-water state dominated by phytoplankton, but it is unclear if and how these two states affect the emission to the atmosphere of greenhouse gases (GHGs) such as carbon dioxide ($CO_2$), methane ($CH_4$) and nitrous oxide ($N_2O$)."

Line 9-12: Suggest rewording to something like the following and including "fluxes": "We measured the saturation and air-water flux of CO2, CH4, and N2O gases, and ancillary variables 46 times over 2.5 years in four urban ponds in Brussels, Belgium: two clear-water macrophyte dominated ponds and two turbid-water phytoplankton dominated ponds."

Reply: Text was modified and now reads L10: "We measured on 46 occasions over 2.5 years (between June 2021 and December 2023) the dissolved concentration of $CO_2$, $CH_4$, and $N_2O$ from which the diffusive air-water fluxes were computed, in four urban ponds in the city of Brussels (Belgium): two clear-water macrophyte-dominated ponds (Silex and Tenreuken), and two turbid-water phytoplankton-dominated ponds (Leybeek and Pêcheries)."

Line 12-15: Here and throughout, I suggest authors use first person instead of passive voice. I reword the next two sentences to include the objective up front and change to passive voice. I also include the method for ebullitive fluxes: "To quantify CH4 ebullitive fluxes we conducted 8 bubble trap deployments totaling 48 cumulated measurements. To characterize methanogenic pathways (acetoclastic or hydrogenotrophic) and quantify water column methane oxidation (MOX) we measured the 13C/12C isotope ratio of CH4 (δ13 13 C-CH4) from bubble traps and sediment bubbles."

Reply: Text was modified and now reads L16: "$CH_4$ ebullitive fluxes were measured with bubble traps in the four ponds during deployments in spring, summer, and fall, totalling 48 days of measurements. To characterize methanogenic pathways (acetoclastic or hydrogenotrophic) and quantify water column methane oxidation (MOX) we measured the $^{13}C/^{12}C$ ratio of $CH_4$ ($\delta^{13}C$-$CH_4$) from gas trapped in the bubble traps, from bubbles deliberately released by the perturbation of the sediments, and in dissolved $CH_4$ in the water column."

Line 15-18: These results could be removed from the abstract. Temperature and precipitation are already touched on later when discussing fluxes.

Reply: We have removed these results from the abstract

Line 18: Remove "The sampled".

Reply: We have removed the two words, text now reads L26: "The turbid-water and clear-water ponds did not differ significantly in terms of diffusive emissions of $CO_2$ and $N_2O$."

Line 22-23: The sentence beginning with "The temperature sensitivity.." could be removed or combined with previous.

Reply: We reduced the length of the sentence, text now reads L32: "The temperature sensitivity of ebullitive $CH_4$ fluxes decreased with increasing water depth."

Line 23-28: These sentences could be combined to reduce wordiness.

Reply: We have reduced the length of this part, text now read L33: "In summer, the $\delta^{13}C$-$CH_4$ values of sediment bubbles indicated that the hydrogenotrophic methanogenesis pathway seemed to dominate in clear-water ponds and acetoclastic methanogenesis pathway seemed to dominate in turbid-water ponds. The $\delta^{13}C$-$CH_4$ values of bubbles traps suggested a seasonal shift from the acetoclastic methanogenesis pathway in spring-summer to the hydrogenotrophic methanogenesis pathway in fall."

Line 35: I suggest adding a concluding sentence to highlight the implications and usefulness of the results.

Reply: We thank the reviewer for the suggestion, but we think that the present version of the abstract achieves the purpose of an abstract that is to list main results/findings of a paper.

INTRODUCTION

Line 39-41: This sentence leads me think you are going to further discuss lentic versus lotic GHGs. I suggest replacing with a sentence highlighting estimated GHG emissions from lakes combined to guide the reader into the next sentence about small pond contributions.

Reply: We feel that it is relevant to contextualize the GHG emissions from lentic systems in broader context of inland water emissions, as such the comparison between lentic and lotic systems is relevant.

Line 42: change "could be" to "are" or "can be"

Reply: Text was modified and now reads L57: "The contribution of $CO_2$ and $CH_4$ emissions from small lentic water bodies (small lakes and ponds) can be disproportionately high compared to large systems (Holgerson and Raymond, 2016) as small lakes and ponds are the most abundant of all water body types in number (Verpoorter et al., 2014, Cael et al., 2017), and flux intensities (per $m^2$) are usually higher in smaller water bodies."

Line 42: shallow lakes are not always ponds (see Richardson et al. (2023) on defining ponds, DOI: 10.1038/s41598-022-14569-0). Maybe cite Downing (2010; DOI: 10.23818/limn.29.02) to make the point in this sentence for small ponds?

Reply: We agree that shallow lakes are not always ponds, and that are differences in functioning that are indeed discussed in detailed by Richardson et al. (2023) Yet, there is so little information and publications on GHGs from ponds that frequently information from publications in lakes are needed to discuss certain aspects of GHG dynamics in ponds. Text was modified and now reads L57: "The contribution of $CO_2$ and $CH_4$ emissions from small lentic water bodies (small lakes and ponds) can be disproportionately high compared to large systems (Holgerson and Raymond, 2016) as small lakes and ponds are the most abundant of all water body types in number (Verpoorter et al., 2014, Cael et al., 2017), and flux intensities (per $m^2$) are usually higher in smaller water bodies."

Line 44-49: Not sure if I would say artificial ponds are "seldom" investigated these days, maybe that the body of literature is growing. Other artificial and/or stormwater pond papers looking at GHGs and carbon inputs: Goeckner et al. 2022 (DOI: 10.1038/s43247-022-00384-y), Ray and Holgerson 2023 (DOI: 10.1029/2023GL104235), and Kalev et al. 2020 for DOC and POC inputs (DOI: 10.1016/j.scitotenv.2020.141773).

Reply: We thank the reviewer for these references. Text was modified and now reads L66: "Among artificial systems, urban ponds are the subject of a growing body of literature (Singh et al., 2000; Natchimuthu et al., 2014; van Bergen et al., 2019; Audet et al., 2020; Peacock et al., 2021; Goeckner et al., 2022; Ray and Holgerson, 2023; Bauduin et al., 2024)."

Line 50: I'm not sure that I agree that urban ponds are mostly in green spaces. If you are referring to a particular region, I would specify that, but this point contradicts what you say in the next sentence that they are surrounded by impervious surfaces.

Reply: The reviewer is right, we have re-worded the sentence and text now read L71:" Urban ponds are generally small, shallow, and usually their catchment consists in majority of impervious surfaces with a smaller contribution from soils (Davidson et al., 2015; Peacock et al., 2021)."

Line 53-54: This sentence is redundant with the sentence on lines 46-48 on C & N inputs. I suggest moving this up to replace that sentence and added a concluding sentence here that highlights a knowledge gap covered in your study.

Reply: We have removed the sentences and added information earlier in the text. Text now read L63: "These higher emissions seem to result from higher external inputs of anthropogenic carbon and nitrogen in artificial systems such as rainfall runoff that brings organic matter and dissolved inorganic nitrogen (DIN), but might also reflect other differences compared to natural systems such as in hydrology (Clifford and Heffernan, 2018)."

Line 55-56: This sentence is a little confusing to me. When you say submerged aquatic primary production, are you referring to the contribution of submerged aquatic vegetation to primary production? If so, I don't think phytoplankton is typically referred to as submerged vegetation. I would simply say primary production or reorganize the beginning of this paragraph to begin with the alternative stable states.

Reply: Text was modified and now reads L75: "In shallow ponds and lakes, including urban ponds, aquatic primary production is either dominated by submerged macrophytes or by phytoplankton, corresponding to two alternate states (Scheffer et al., 1993)."

Line 57: Indicate which stable state is associated to clear or turbid water.

Reply: Text was modified and now reads L76: "These two alternative states correspond to clear waters (macrophyte-dominated) or turbid waters (phytoplankton-dominated), during the productive period of the year (spring and summer in mid-latitudes)."

Line 58: Macrophytes also impact CO2 cycling (e.g., in the Theus et al. 2023 you cite in the next sentence). Further, no background is provided for the effect of stable states on CO2 at all in the introduction. This should be included as it is for CH4 and N2O.

Line 62: Ojala et al. 2011 may also be a relevant paper to check out but they focus on clear versus brown-water lakes (DOI: 10.4319/lo.2011.56.01.0061).

Reply: We have included the suggested references and add information about $CO_2$ cycling in the two stable states. Text now reads L78: "Submerged macrophytes and phytoplankton regulate $CO_2$ dynamic directly through photosynthesis that can be more or less balanced by community respiration in the water column. However, it is not clear whether the presence of macrophytes increases or decreases the $CO_2$ emissions from ponds and lakes. Some studies have shown a decrease of $CO_2$ emissions with increasing macrophyte density (Kosten et al., 2010; Ojala et al., 2011; Davidson et al., 2015), but other studies showed the opposite pattern (Theus et al., 2023). In phytoplankton-dominated lakes, $CO_2$ concentrations depend in part on the development stage of the phytoplankton, with the growth and peak phases generally coinciding with lower $CO_2$ concentrations due to intense photosynthesis (Grasset et al., 2020; Vachon et al., 2020)."

Line 67-70: I would combine these sentences to highlight where positive N2O-macrophyte relationships have been reported and save the details for the discussion.

Reply: Text was modified and now reads L101: " $N_2O$ emissions has been showed to follow diurnal cycles of $O_2$ concentrations in areas dominated by submerged macrophytes in Lake Wuliangsuhai (China) (Ni et al., 2022) and the seasonal cycle of aboveground biomass of emerged macrophytes (*Phragmites*) in Baiyangdian Lake (China) (Yang et al., 2012)."

Line 71: Authors haven't described why denitrification and N2O are associated. I would either include their association (i.e., that N2O can be produced is an intermediate product of denitrification or nitrification), or remove and save this for the discussion.

Reply: The reviewer is right and we have added information about processes leading to N2O production. Text now reads L93: "The production of $N_2O$ predominantly occurs through microbial nitrification and denitrification that depend on DIN and $O_2$ levels (Codispoti and Christensen, 1985; Mengis et al., 1997). Competition for DIN between primary producers and $N_2O$-producing microorganisms can impact $N_2O$ production. Additionally, the transfer of labile phytoplankton organic matter to sediments fuels benthic denitrification. Combined, these two processes could explain that some lakes can act as sinks of $N_2O$ under elevated Chl-*a* concentrations (Webb et al., 2019; Borges et al., 2022)."

Line 72: I suggest concluding with a sentence on the significance of quantifying GHG fluxes in macrophyte vs. phytoplankton dominated systems. Further, no background information is provided on GHG dynamics from phytoplankton dominated ponds, of which there is plenty of literature on.

Reply: We have added concluding sentences to the paragraph that now reads L106: "There have been a very limited number of studies investigating systematically how emissions differ between ponds dominated by phytoplankton and those dominated by macrophytes (Harpenslager et al., 2022; Baliña et al., 2023), and none investigating simultaneously $CO_2$, $CH_4$, and $N_2O$ emissions including both diffusive and ebullitive components."

Line 73 / paragraph 3: This paragraph is good. I suggest adding some support from Ray and Holgerson (2023) on the contribution of ebullition to CH4 fluxes in artificial ponds. Also, you don't mention anything about methanogenesis or methane oxidation until the closing paragraph, whereas this is a profound and really interesting part of your work! The significance of understanding methanogenic pathways, and methane oxidation, should be included in this paragraph.

Reply: We thank the reviewer for pointing this very interesting paper that we have added in the text that now reads L111: "At annual scale, ebullitive $CH_4$ flux usually represents more than half of total (diffusive+ebullitive)

CH$_4$ emissions from shallow lakes (Wik et al., 2013; Deemer and Holgerson, 2021), although the relative contribution of ebullitive and diffusive CH$_4$ emissions is highly variable seasonally (*e.g.* Wik et al., 2023; Ray and Holgerson, 2023).”

We also added a paragraph about methanogenesis and methane oxidation after this paragraph with several references. New paragraph L119: “The two primary metabolic pathways for CH$_4$ production in sediments by methanogenic archaea are the fermentation of acetate (acetoclastic pathway) and the reduction of carbon dioxide by H$_2$ (hydrogenotrophic pathway) (Whiticar et al., 1986; Conrad, 1989). CH$_4$ produced by these two pathways exhibits distinct $^{13}$C/$^{12}$C ratios ($\delta^{13}$C-CH$_4$) (Whiticar et al., 1986) and can be used to discriminate which pathway is dominant. When CH$_4$ diffuses from the sediment to the water column, it can be oxidized by methanotrophic bacteria who preferentially consume CH$_4$ with $^{12}$C over $^{13}$C, resulting in an increase of $\delta^{13}$C-CH$_4$ of the residual CH$_4$ in the water column (Bastviken et al., 2002). Fractionation models then allow estimating methane oxidation (MOX) from measurements of $\delta^{13}$C-CH$_4$ of dissolved CH$_4$ in the water column. Bastviken et al. (2008) report that 30 to 99% of the CH$_4$ produced in sediments of freshwater lakes can be removed by MOX that is as a significant CH$_4$ sink in these water bodies. MOX is known to be inhibited by light (Dumestre et al., 1998) and increases with the presence suspended particles (Abril et al., 2007) so that MOX might vary between clear and turbid waters (Morana et al., 2020). “

Line 81 / paragraph 4: This whole paragraph should be re-written to outline the objectives of this study. To me they were (1) to understand annual variability in saturation/fluxes, (2) characterize and quantify CH4 cycling pathways (methanogenesis/methonotrophy), and (3) identify drivers of these fluxes/pathways including pond type and environmental variables. Then you can briefly say you collect 2.5 years worth of data (so impressive!) on GHG dynamics and environmental conditions in four urban ponds of differing stable states. I also suggest adding predictions based on supporting information provided earlier in the intro. Then conclude with why the study contributes to the body of lit on urban pond GHG dynamics.

Reply: The reviewer is right. We have reduced the size and briefly talk about the 2.5 years of data. Text now reads L130 : “Here, we report a dataset of CO$_2$, CH$_4$, and N$_2$O dissolved concentrations in four shallow and small urban ponds (Leybeek, Pêcheries, Silex, and Tenreuken) in the city of Brussels (Belgium) (Fig. 1), with data collected 46 times at regular intervals (between June 2021 and December 2023) on each pond. The air-water diffusive fluxes of CO$_2$, CH$_4$, and N$_2$O were calculated from dissolved concentrations and the gas transfer velocity, while the ebullitive CH$_4$ fluxes were measured with inverted funnels during 8 deployments (totalling 48 days) in the four ponds. The $\delta^{13}$C-CH$_4$ in the sedimentary bubbles and in the water provides additional information on CH$_4$ dynamics such as the methanogenesis pathway (acetoclastic or hydrogenotrophic) and MOX. We test the hypothesis that the two alternative states in shallow lakes (a clear-water state dominated by macrophytes, or a turbid-water state dominated by phytoplankton) drive differences in the CO$_2$, CH$_4$, and N$_2$O dissolved concentration and diffusive emissions from the four studied artificial ponds, that have similar depth, surface area, and catchment urban coverage, and that mainly differ by the phytoplankton-macrophyte dominance. We also test the hypothesis that the two alternative states in shallow lakes drive differences in the ebullitive CH$_4$ emissions, water column MOX, and sedimentary methanogenesis pathway (acetoclastic or hydrogenotrophic) in the four studied ponds. The final objective of the present work is to determine the relative contribution of CO$_2$, CH$_4$, and N$_2$O to the total GHG emissions in CO$_2$-eq and to test the hypothesis that the relative contribution of each GHG differs according to the two alternative states in shallow lakes.”

METHODS

Line 98: Did you visit each pond on the say day? What time of day (approximate window) did you sample? If you remove the methodological specifics from the conclusion of the introduction, add the details here about the 46 sampling days and period of time you sampled sites from (June 2021 – Dec. 2023). Also, how would you describe the climate/precipitation of Brussels? These can be added before the sentence here.

Reply: We have removed the methodological specifics from conclusion of the introduction to the material and methods section. Text was modified and now reads L155: “Sampling was carried out from a pontoon in the four ponds on the same day between 9am and 11am, 46 times on each pond between June 2021 and December 2023 at a frequency ranging from one (winter) to three (summer) times per month at a single fixed station in each of the four ponds.”

We also added information about climate in Belgium in Results and Discussions, before discussing seasonal variations of GHG. Text now reads L330: "Belgium has a west coast marine climate with mild weather year-round, and evenly distributed abundant rainfall totalling on average 837 mm annually for the reference period 1991-2020. The average annual air temperature was 11°C, with summer average of 17.9 °C and winter average of 4.1 °C for the reference period 1991-2020."

Line 98-99. I would separate GHGs and "other variables" here and focus on the "other variables" first. Then move on to GHG sample collection. In any case, how far below the surface did you collect water from?

Reply: We have modified the text to start with the sampling of GHG and finish with the sampling for "other variables" (see next comments), and we add the distance from the surface for the sampling. Text now reads L157: "Water was sampled 5cm below the surface with 60ml polypropylene syringes for analysis of dissolved concentrations of $CO_2$, $CH_4$, and $N_2O$."

Line 100-103: pCO2 analytical approach needs to be moved to the same section as CH4 and N2O unless it was a portable analyzer (unclear). When you say headspace approach, are you referring to the headspace equilibrium approach? If you used the same approach following the cited Borges et al. (2019) then I think so? Unclear. If so, more information is needed here for the headspace approach. How much water volume versus headspace volume did you equilibrate? How long did you equilibrate? Did you use N2 gas as the headspace or ambient air??

Reply: We have added information on GHG measurements and the headspace technique for $pCO_2$. The three gases were collected with syringes in the field, but $pCO_2$ was measured directly in the field with a portable Li-Cor Li-840 infrared gas analyser. Samples for $CH_4$ and $N_2O$ were collected in the field and measurements were done after in laboratory. Text now reads L158: "Samples for $CH_4$ and $N_2O$ were transferred from the syringes with a silicone tube into 60 ml borosilicate serum bottles (Weathon), preserved with 200 μl of a saturated solution of $HgCl_2$ , sealed with a butyl stopper and crimped with aluminium cap, without a headspace, samples were stored at ambient temperature protected from direct light prior to analysis in laboratory. The partial pressure of $CO_2$ ($pCO_2$) was measured directly in the field, within 5 minutes of sample collection, with a Li-Cor Li-840 infrared gas analyser (IRGA) based on the headspace technique with 4 polypropylene syringes (Borges et al., 2019). A volume of 30 ml of sample water was equilibrated with 30 ml of atmospheric air within the syringe by shaking vigorously for 5 minutes. The headspace of each syringe was then sequentially injected into the IRGA and a fifth syringe was used to measure atmospheric $CO_2$. The final $pCO_2$ value was computed taking into account the partitioning of $CO_2$ between water and the headspace, as well as equilibrium with $HCO_3^-$ (Dickson et al., 2007) using water temperature measured in-situ and after equilibration, and total alkalinity (data not shown). Samples for total alkalinity were conditioned, stored and analysed as described by Borges et al. (2019)."

Line 102: After each cruise? Do you mean when you sampled a pond from the pontoon? I suggest saying "before and after each sampling event". Also, is the Li-Cor Li-840 a portable gas analyzer? I didn't think so but now I'm wondering if it is. If so, measurements of CO2 in the field needs to be explicitly stated and the approach described better."

Reply: Text was modified and now reads L163: "The partial pressure of $CO_2$ ($pCO_2$) was measured directly in the field, within 5 minutes of sample collection, with a Li-Cor Li-840 infrared gas analyser (IRGA) based on the headspace technique with 4 polypropylene syringes (Borges et al., 2019)." and L170: "The IRGA was calibrated in the laboratory with ultrapure $N_2$ and a suite of gas standards (Air Liquide Belgium) with $CO_2$ mixing ratios of 388, 813, 3788 and 8300 ppm."

Line 104: Change "in" to "into" and "poisoned" to "preserved".

Reply: Text was modified and now reads L158: "Samples for $CH_4$ and $N_2O$ were transferred from the syringes with a silicone tube into 60 ml borosilicate serum bottles (Weathon), preserved with 200 μl of a saturated solution of $HgCl_2$ , sealed with a butyl stopper and crimped with aluminium cap, without a headspace, samples were stored at ambient temperature protected from direct light prior to analysis in laboratory."

Line 117: How did you store the gas prior to analysis? Same with other types of samples collected, storage prior to analysis should be included.

Reply: We added this information  and text now reads L158: "Samples for $CH_4$ and $N_2O$ were transferred from the syringes with a silicone tube into 60 ml borosilicate serum bottles (Weathon), preserved with 200 µl of a saturated solution of $HgCl_2$ , sealed with a butyl stopper and crimped with aluminium cap, without a headspace, samples were stored at ambient temperature protected from direct light prior to analysis in laboratory.", L185: "The value of the collected volume of gas was logged, and the gas was transferred immediately after collection to pre-evacuated 12 ml vials (Exetainers, Labco, UK) that were stored at ambient temperature protected from direct light prior to the analysis of $CH_4$ concentration and $\delta^{13}C\text{-}CH_4$ in the laboratory." and L191: "The gas collected in the funnels was stored in pre-evacuated 12 ml vials (Exetainers, Labco, UK) that were stored at ambient temperature protected from direct light prior to the analysis of $\delta^{13}C\text{-}CH_4$ in the laboratory."

Line 125: Ok so for CH4 and N2O you collected water, then used the headspace equilibration approach in the lab? If you also used this approach for CO2 but it was done in the field for CO2, this still should be described earlier.

Reply: Text was modified to clarify that $pCO_2$ was measured directly in the field and that samples for $CH_4$ and $N_2O$ were stored and analyzed in laboratory.

Line 130-131: I strongly advice authors to report each gas as both a concentration and some form of their deviation from equilibrium. Authors say reporting pCO2, nmol/L of CH4, and %N2O is "with convention in existing topical literature", but no references are provided, and I disagree with this approach. Other impactful pond GHG papers focusing on concentrations alone maintain the same units (e.g., Holgerson 2015, DOI: 10.1007/s10533-015-0099-y), and this allows for easier comparison. If N2O is presented as a deviation from equilibrium, I think the same should be included for CO2 and CH4, as deviation from equilibrium is insightful for biological changes to these gases. This helps readers understand to what degree the gases are "systematically and distinctly above saturation".

Reply: We acknowledge that using different units can be confusing. However, the units we have used for CO2 and CH4 are standard in topical literature and used in numerous publications from our group. For $N_2O$, presenting values as percentages around 100% makes it straightforward to determine whether the pond is acting as a source or sink. We recently published a companion paper on GHGs dynamics in Brussels' urban ponds based on an independent data-set using units of $CO_2$, $CH_4$ and $N_2O$ used here (Bauduin et al. 2024; https://doi.org/10.1016/j.watres.2024.121257); if readers want to compare the results and conclusions from both papers, it is preferable that the units are consistent; Please note that the full data-set is publically available, so the readers can re-use the data in their preferred units.

Line 133: How did you calculate the equilibrium solubility of N2O in water?? Did you calculate N2O solubility using the water temperature at the time you collected samples (based on Henrys law)?

Reply: The reviewer is right and indeed we used the temperature at the time we collected our samples. We have included this information in the text that now reads L222: "The $N_2O$ concentrations fluctuated around atmospheric equilibrium, so data are presented as percent of saturation level ($\%N_2O$, where atmospheric equilibrium corresponds to 100%)."

Line 142: I suggest moving this section above the GHG analytical section to improve the flow of methods. (GHG analysis -> GHG calculation)

Reply: We have moved the pre-mentioned section on measurement of other variables above at L201.

Line 154: Is this DIN the sum of NH4-N, NO3-N, and NO2-N? or the sum of the full concentration of each?

Reply: DIN is the sum of concentrations of $NH_4^+$, $NO_3^-$ and $NO_2^-$ expressed in µmol/L (and not as mg/L). This was clarified in text L212: "Concentration of dissolved inorganic nitrogen (DIN) was calculated as the sum $NH_4^+$, $NO_2^-$ and $NO_3^-$ concentrations in µmol $L^{-1}$."

Line 158: Finally I see that CO2 was measured in the field with the Li-Cor Li-840. This needs to be made clear much early on...

Reply: We have included the information about the Li-Cor Li-840 earlier in the text to clearly specify that we measured $pCO_2$ directly in the field.

Line 160: Change "on" to "in".

Reply: Text was modified and now reads L255: "The atmospheric $pCO_2$ was measured in the field with the Li-Cor Li-840."

Line 160: Where did you get this value of 1.9 ppm for CH4?

Reply: We have taken this value from the NOAA Global Monitoring Laboratory measurements, whose data is available online. Text was modified and now reads L255: "For $CH_4$, the global average present day atmospheric mixing ratio of 1.9 ppm was used (Lan et al., 2024)."

> Lan, X., K.W. Thoning, and E.J. Dlugokencky: Trends in globally-averaged CH4, N2O, and SF6 determined from NOAA Global Monitoring Laboratory measurements [data set]. Version 2024-08, https://doi.org/10.15138/P8XG-AA10, 2024.

Line 160-163: I'm a little confused. Because authors use the phrasing "equilibrium with atmosphere for N2O" it sounds like equilibrium solubility in water. Because authors also say air mixing ratios here, I think that this is not what authors intend and instead they mean the atmospheric concentration of N2O.

Reply: The reviewer is right and we have changed the text to beter clarify the sentence. Text was modified and now reads L257: "Atmospheric $N_2O$ concentration was calculated from the average air mixing ratios of $N_2O$ provided by the GMD of the NOAA ESRL (Dutton et al., 2017)."

Line 211 / statistics: This statistics section requires much more detail and I'm a little concerned about the statistics overall but this may be because things are hard to follow. I cover my concerns in the general comment above for this section.

Reply: The reviewer is right; we have expanded the description of the statistical tests used (see the response to the general comment above).

RESULTS & DISCUSSION

Line 237-238: this sentence is redundant with the previous where you already report these values. I would remove and site Figure 3 with the previous sentence.

Reply: We have revised this paragraph to be more concise and explain differences between clear-water and turbid-water ponds. Text now reads L353: "The four studied ponds had significantly different Chl-*a* concentration values during summer, with the Leybeek pond having higher Chl-*a* (78.8±49.5 µg $L^{-1}$), followed by the Pêcheries pond (19.1±13.7 µg $L^{-1}$), the Tenreuken pond (3.3±2.4 µg $L^{-1}$), and the Silex pond (1.0±1.2 µg $L^{-1}$) (Tukey's HSD test p ≤0.0001 for each pair of comparisons, Figs. 1, 3). The Leybeek and Pêcheries ponds with higher summer Chl-*a* concentration had turbid-water (summer TSM = 48.7±36.2 and 13.7±10.7 mg $L^{-1}$, respectively), and undetectable submerged macrophyte cover in summer (Fig. 1, Table S1). The Tenreuken and Silex ponds with lower summer Chl-*a* concentrations had clear-water (summer TSM = 4.9±3.2 and 4.0±3.2 mg $L^{-1}$, respectively), and a high total macrophyte cover during summer (68 and 100%, respectively, Fig. 1, Table S1). Values of Chl-*a* were higher in summer than in winter in the turbid-water Leybeek and Pêcheries ponds (Tukey's HSD test p=0.0107 for the Leybeek pond, p=0.0211 for the Pêcheries pond) related to summer algal blooms. Values of Chl-*a* were higher in winter than in summer in the clear-water Tenreuken and Silex ponds (Tukey's HSD test=0.0296 for the Tenreuken pond, p=0.0056 for the Silex pond), probably related to competition for inorganic nutrients from macrophytes, with the Silex pond showing lower summer Chl-*a* (Tukey's HSD test p<0.0001), lower summer TSM concentrations (Tukey's HSD test p<0.0001) and higher summer total macrophyte cover compared to the Tenreuken pond (Fig. 1)."

Line 252: Were these in the surface of the water? Would be helpful if the depth of measurements are specific in methods.

Reply: We added the information on the depth of measurement in methods. Text now reads L157: "Water was sampled 5cm below the surface with 60ml polypropylene syringes for analysis of dissolved concentrations of $CO_2$, $CH_4$, and $N_2O$." and L174: "Water temperature, specific conductivity, and oxygen saturation level (%$O_2$) were measured in-situ with VWR MU 6100H probe 5cm below the surface."

Line 256: The range is from 40 to 13804 ppm but in the methods authors stated that pCO2 was "systematically and distinctly above saturation level" which was cited at 400 ppm. I'm curious now if authors meant atmospheric CO2, of CO2 dissolved in water, which would differ based on temperature.

Reply: The reviewer is right. However, under-saturation of $pCO_2$ in water relatively to $pCO_2$ measured in-situ in ambient air were only observed on 5 occasions out of 187 measurements. We have added a sentence at the beginning of the paragraph to add this information. Text now reads L391: "Undersaturation of $CO_2$ with respect to atmospheric equilibrium was only observed on five occasions out of the 187 measurements, three times in the turbid-water Leybeek pond in summer (40 ppm on 13 August 2021, 220 ppm on 27 June 2022 and 149 ppm on 13 June 2023), and twice in the clear-water Tenreuken pond in spring and summer (383 ppm on 13 August 2021 and 55 ppm on 2 May 2022)."

Line 257: warmer waters also hold less gas. Including percent or factor of saturation compared to equilibrium may be helpful to understand lower CO2 concentrations in the summer.

Reply: We agree. Yet, the temperature effect is overshadowed by the enormous range of variations of $pCO_2$ (40 to 13804 ppm).

Line 261: What kind of model was used for the results in Table 3? Are these results from a PERMANOVA? If so, nothing about a PERMANOVA was included in the stats section. Again, pseudoreplication from repeated sampling over time. There are also linear regressions of these gases over some of these variables in other analyses, which is redundant.

Reply: We have updated the statistics paragraph in the material and methods to better stated tests that was used in the article. In table S3 are reported Pearson coefficient between one GHG and one environmental variable in one pond to investigate relationships among each pond. ANOVA were used to compare one variable between seasons and ponds. The pseudoreplication problem was assessed in the comparisons using a repeated measures two-way ANOVA.

We believe that the relationships presented in the different figures are useful and support the text and the results we present.

Line 270: What do you mean by "sometimes correlated". As in during certain seasons or only in some ponds?

Reply: We have modified the text that now reads: L410: "In individual ponds, dissolved $CH_4$ concentration was negatively correlated to precipitation and DIN in the Pêcheries pond (Table S3; Fig S6), and positively correlated to SRP in the Silex pond (Table S3; Fig S4)."

Line 283: Not having explanations for correlations included in models is a good example of why a prior hypotheses are useful. There is already a lot of information in this manuscript so maybe some of these analyses for drivers of gas concentrations are not needed in the first place? Just a thought.

Reply: We feel that these explanations are useful.

Line 286: This makes me wonder if this DIN is the N fraction of inorganic nitrogen forms or their full concentration (i.e., NH4-N versus NH4) and what the difference would be for results here between either option, but I think the sum of N fraction in the inorganic forms is more appropriate.

Reply: DIN is given as the sum of concentrations of NH4+, NO3- and NO2- expressed in µmol/L (and not as mg/L). This was clarified in text L213.

Line 295: Would be helpful to summarize at the end of this section what the differences in GHGs for macrophyte versus phytoplankton dominated ponds.

Reply: We added a summarizing paragraph to conclude on differences in relationships between GHG and environmental variables observed between clear-water and turbid-water ponds. Text now reads L455: "The relationships between GHG dissolved concentrations and other variables were similar in clear-water macrophyte-dominated ponds and turbid-water phytoplankton-dominated ponds. $pCO_2$ was positively correlated with precipitation, and dissolved $CH_4$ concentration was positively correlated with temperature, while no significant correlation was found between $\%N_2O$ and other variables in the four ponds taken individually. The negative correlation between $pCO_2$ and $\%O_2$ reflected the photosynthesis-respiration balance independently from the community driving aquatic primary production (macrophytes in clear-water ponds and phytoplankton in turbid-water ponds)."

Line 348-349: The observation of higher CH4 in summer and spring was already noted for CH4 concentration. Maybe a good example of why reporting concentrations and fluxes is not needed?

Reply: We prefer to keep the information because it shows that $CH_4$ fluxes depend on temperature (like CH4 concentration) rather than on wind. It is conceivable that the variability of fluxes depends mainly on the variations of the gas transfer velocity rather than variations on concentrations.

Line 367: See also Ray and Hoglerson (2023), DOI provided above.

Reply: We added this reference in the sentence. Text now reads L545: "This finding is consistent with other studies showing that ebullitive $CH_4$ fluxes can account for more than half of total $CH_4$ emissions in small and shallow lentic systems (*e.g.* Wik et al., 2013; Deemer and Holgerson, 2021; Ray and Holgerson, 2023; Rabaey and Cotner, 2024)."

Line 380: This is an example of where a multiple regression model could be used.

Reply: The reviewer is right. This section aims to compare $CH_4$ fluxes between clear and turbid ponds and identify controlling factors. However, the aim of this figure is to emphasize that the variations in diffusive and ebullitive CH4 fluxes can be attributed to either macrophytes, phytoplankton, or a combination of both. The figure presents this information concisely and clearly conveys the message of the text.

Line 404: No mention was made in statistics of using nonlinear regressions

Reply: The reviewer is right. We have added why used linear and quadratic regressions in the statistics section of the material and methods. Text now reads L311: "To assess the impact of Chl-*a* concentration, macrophyte cover in summer, water depth, and lake surface area on diffusive and ebullitive $CH_4$ fluxes, the ratio of ebullitive $CH_4$ to total $CH_4$ flux, and $CO_2$ and $N_2O$ fluxes, both linear and quadratic relationships were applied to log-transformed averaged data. This approach allowed for the observation of trends between explanatory and dependent variables. For $N_2O$ fluxes, additional explanatory variables included $NO_2^-$, $NO_3^-$, $NH_4^+$, and DIN concentrations."

---

## Author Response (AR2)

Dear Dr. Singer,

Please find enclosed the revised version of manuscript "Methane, carbon dioxide, and nitrous oxide emissions from two clear-water and two turbid-water urban ponds in Brussels (Belgium)"

As recommended by the reviewer we analyzed the data using GLMM to include random effects when appropriate. For some tests, the data-sets were too small to use GLMM (the model did not converge) so we used other methods (Pearson or ANOVA). Please note that we found the same patterns in the data analysis as before, so the new statistical analysis did not change the overall conclusions of the paper.

As recommended, we have also separated the Results and Discussion sections.

As recommended, we simplified and shortened the text, and we have reduced the number of figures.

We have narrowed down the scope of the paper to the GHG emissions and we removed the analysis on methane oxidation and on methanogenesis pathways.

We are grateful for the editorial guidance and the reviewers' suggestions and we sincerely hope that the present version of manuscript meets the requirements for acceptance in Biogeosciences.

Best regards,

Thomas Bauduin, Nathalie Gypens, Alberto Borges

**Associate Editor**

I am sorry for the long time it took to process another round of reviews for your manuscript. Long manuscripts may also need a long time to review.

Reply: We are very grateful for the second round of reviews and your annotated manuscript.

I have received very mixed feedback from the two reviewers who have seen your manuscript also in its first version. Reviewer #1 considers that his/her concerns were well addressed and recommends acceptance of the manuscript upon having taken care of a few minor points. Reviewer #2 acknowledges the work you have put into addressing the raised concerns and into improving the manuscript, in particular with regard to improved flow in the abstract and an improved introduction that now includes hypotheses. This reviewer also still recognizes the value of your dataset on pond GHG dynamics. However, the final verdict of this reviewer was to reject the manuscript because of two reasons: (i) still existing serious deficits with regard to statistical analysis, and (ii) general readability difficulties and complexity of the study presentation given the very long combined results and discussion section. I gave your revised manuscript a careful read once more and have to agree with reviewer #2 regarding the statistical issues, yet I believe these issues can be fixed. I also still believe - as pointed out earlier - that your manuscript is really long and its presentation overly complex, thereby risking only limited attractivity to readers. I don´t think that this issue is critical enough to justify rejection of the paper, however. Once more, I side with reviewer #2´s recommendation to separate results and discussion to create a manuscript that leads the reader through the many results and large number of response variables in a less confusing way.

Reply: We have carefully considered the editorial recommendations. In order to accommodate the request to reduce the size (text length and number of figures) and to re-structure the manuscript into separate Results and Discussion sections, we have removed the data MOX and methanogenesis pathways based on the stable isotope data. The resulting manuscript is solely focused on the GHG emissions, more condensed, reduced in the number of figures, and structured in separate results and discussion. We have also reviewed the statistics (see below and in the response to reviewer 2).

A few more words on statistical issues:

1) Besides giving clear recommendations of how to take care of the temporal nature of your dataset, Reviewer #2 considers your use of the Pearson correlation coefficient inadequate given ignorance of the repeated measures nature of your data. The clear concern is that the current approach inflates significance of some explanatory variables, which is not best practice for temporally repeated measurements (even if you find significance). Please also respect the difference between correlation and regression. For example, in line 370 you argue for slopes of correlations to be significantly different, which is meaningless while also it remains unclear how this was tested. Another example is Fig. 4, where a functional relationship between %CH4 as a driver of bubble flux is implied - I see no justification for a regression in this case, maybe it would be better to plot temperature against %CH4.

Reply: We have used GLMM when the data-set was large enough. For some tests, the data-sets were too small to use GLMM (the model did not converge) so we used other methods (Pearson or ANOVA). Please note that we used ANOVA for repeated measures, so the test does incorporate to some extent the nature of temporally repeated measurements.

We have changed the figure of $\%CH_4$ as suggested but we kept as a supplemental the relation between $\%CH_4$ and bubble flux. We agree that the $\%CH_4$ is not a driver of the bubble flux but there is a justification to plot the data because it shows that $\%CH_4$ increases with bubble flux. So, the increase of $CH_4$ ebullition with temperature results from an increase in both bubble flux and $\%CH_4$. It is conceivable that the $\%CH_4$ of bubbles would remain unchanged and that the ebullition of $CH_4$ was solely a function of bubble flux. The data show that ebullition of $CH_4$ increases with temperature as a result of an increase of both bubble flux and $CH_4$ content.

2) The hypotheses brought forward in lines 117-125 argue for effects of alternative stable states on GHG concentrations, emissions, partitioning into fractions of various GHGs. There is no hypothesis about seasons or any other effect. However, in the statistics section (line 272-279), we can find nothing about how the factor "alternative state" is tested, yet we read "four seasons were serving as independent factor". Also, pond ID seems to have been used as a fixed AND as a random factor, which is confusing and hardly correct. This serious disagreement between hypotheses and methods is then also evident in the R&D section, where at multiple places "significant" effects are stated, yet it remains unclear to which test you refer (examples in lines 370 (already mentioned above) or 506-510).

Statistical analyses were revised:

- We tried to compare the ponds according to 'alternative state'; however, the data-set is only made up of 2 turbid-water ponds and 2 clear-water ponds and this number was insufficient to perform this comparison with GLMM using "alternative state" as fixed variable, and "pond" and "data" as random effects, as you predicted in the final paragraph of your decision letter (below). We therefore compared the 4 ponds with each other (pairwise) and then discussed the differences, considering that 2 were turbid-water and 2 were clear-water, as indicated by Chl-*a* values and macrophyte presence and abundance.

- We compare patterns of GHG versus drivers in the whole data-set (merging the 4 ponds) using GLMM (with sampling date and "pond" as a random factor to take into account repeated measurements over time).
- We investigate seasonal differences by comparing the ponds with each other in each season using repeated measures ANOVA (as GLMM did not converge due to insufficient data).

We have revised the text lines 117-125 (new lines 107-110 in the manuscript without track changes): "We test whether the differences between the four ponds are explained by the two alternative states in terms of (i) $CO_2$, $CH_4$, and $N_2O$ dissolved concentration and diffusive emissions; (ii) ebullitive $CH_4$ emissions; (iii) relative contribution of $CO_2$, $CH_4$, and $N_2O$ to the total GHG emissions in $CO_2$-eq.."). The comparisons described in line 370 have been removed and the comparisons in lines 506-510 (new lines 343-350) are now described in the materials and methods and follow the same methodology as the other comparisons described in the text. We have also revised the entire manuscript to ensure that all the comparisons described are explained in the material and methods, and that the results are fully included in the supplementary materials.

When reading your revised version I also stumbled upon another, likely minor question: You describe usage of an IRGA for a 30 ml headspace sample. I have worked with the Li-840 IRGA myself and wonder whether such a limited volume of gas actually creates a stable plateau reading.

Reply: We compared the measurements of $pCO_2$ with a Li-840 using a 30 ml headspace from discrete measurements with simultaneous $pCO_2$ measurements with a flow-through equilibrator system for a wide range of $pCO_2$ values in diverse environments (rivers and lakes). The agreement is excellent. Please refer to Figure 2 in Abril et al. (https://doi.org/10.5194/bg-12-67-2015) and Figure S22 in Borges et al. (https://doi.org/10.1126/sciadv.abi8716).

I have a few minor comments which I will make available to you as an annotated pdf. In some cases, your text is pretty hard to understand and rewriting suggested.

Reply: We have implemented all of your suggestions on the annotated pdf. Please note that there are no data points for the Netherlands and Québec plot because these are published relationships and are only shown data points for our own measurements. We prefer to keep the units of fluxes in mmol m$^{-2}$ d$^{-1}$ throughout the manuscript (even for the annual averages). This is to avoid mixing different units in the text and figures (which could be confusing for readers). Also the conversion of units from d$^{-1}$ to yr$^{-1}$ is straightforward.

Given the (mixed) opinions by the two reviewers and my own evaluation I am asking you once more to revise your manuscript carefully. I consider this a request for major revision and plan to involve at least one additional reviewer as the reviewers who dealt with your manuscript so far are not willing to provide their expertise once more. I believe that both this fact as well as the long duration of the review process tells you about the challenges an overly long and hard-to-read manuscript creates for your readers. Please carefully revise and clarify usage of statistics. In this respect I see no need to "overdo" it, e.g. **testing an effect of "alternative stable state" may just be deemed impossible given the low sample size of 2 ponds for each factor level**, yet data may still be discussed in this light. However, clearly, your manuscript must be technically correct from a statistical standpoint, and if you bring forward hypotheses to be tested, then this must be reflected in appropriately chosen statistical methods and adequate results. I really urge you to consider shortening your manuscript wherever possible, even if no page limit is enforced by the journal. Splitting the R&D section into separate results and discussion sections is indeed a good advice that will help to make a long manuscript more digestible - you may argue otherwise.

Reply: We have implemented the suggestions of Reviewer#2 on the statistical analysis and have reduced the length of the manuscript and structured it in separate results and discussion.

**#Reviewer 1**

1) Scientific significance

Does the manuscript represent a substantial contribution to scientific progress within the scope of this journal (substantial new concepts, ideas, methods, or data)? Excellent

2) Scientific quality

Are the scientific approach and applied methods valid? Are the results discussed in an appropriate and balanced way (consideration of related work, including appropriate references)? Excellent

3) Presentation quality

Are the scientific results and conclusions presented in a clear, concise, and well structured way (number and quality of figures/tables, appropriate use of English language)? Excellent

For final publication, the manuscript should be accepted subject to minor revisions

Were a revised manuscript to be sent for another round of reviews: I would not be willing to review the revised manuscript.

Suggestions for revision or reasons for rejection (visible to the public if the article is accepted and published)

The authors have done an excellent job addressing the significant concerns I raised in my previous report. After addressing the few minor comments below, I recommend publication.

Line 21: Phytoplankton are also made up of OM. The two OM sources may be distinguished based on quality.

Reply: The reviewer is right. But we did observe higher ebullition in clear-water lakes than turbid-water lakes that should be logically related to macrophyte biomass.

Line 39: Higher or lower or comparable?

Reply: text now reads at lines 35-36 in the manuscript without track changes: "However, reported emissions of $CH_4$ from lakes (Rosentreter et al., 2021; Johnson et al., 2022) are equivalent or even higher compared to rivers (Stanley et al., 2016; Rocher-Ros et al., 2023)"

Line 763. States, "Such a hypothesis was consistent with an overall positive relation between %N2O and DIN in the urban ponds of the city of Brussels irrespective of presence or absence of macrophytes (Bauduin et al., 2024; this study)."However Line 380 states "More surprisingly, %N2O was not correlated with DIN (Table S3; Fig S3, S4, S5, S6) nor with individual forms of DIN (NH4+, NO2-, NO3-) in the four ponds individually or when all the data were pooled together for the individual forms of DIN (Table S3; Fig S7)". It is true to only state that the N2O- DIN positive relationship in urban ponds was found in the previous study alone that tended to be more spatially based rather than including the "this study" citation as it is now.

Reply: Text now reads lines 283-299: "The $\%N_2O$ values ranged from 32 to 826% (Fig. 3). Undersaturation of $N_2O$ with respect to atmospheric equilibrium was observed 66 times out of the 187 measurements. Low values of $\%N_2O$ were generally observed in spring and summer and high values of $\%N_2O$ were generally observed in fall and winter in the four ponds (Fig. 3). During spring, the $\%N_2O$ was lower in the Pêcheries pond (90±11%) than the Leybeek (138±30%, p=0.0043, Table S3) and the Tenreuken (138±41, p=0.0057, Table S3) ponds. During summer, the $\%N_2O$ was lower in the Pêcheries pond (78±17%) than the Leybeek (191±104%, p<0.0001, Table S3) and the Silex (126±49%, p=0.001, Table S3) pond, and lower in the Tenreuken pond (133±106%) than the Leybeek pond (p=0.0219, Table S3). During fall, $\%N_2O$ was lower in the Pêcheries pond (103±33%) than the Leybeek pond (190±70%, p=0.0174, Table S3). For the all sampling period, $\%N_2O$ was lower in the Pêcheries pond (94±28%) than the Leybeek (178±82 %, p<0.0001, Table S7), Tenreuken (140±77%, p<0.0001, Table S7) and Silex (144±113%, p<0.0001, Table S7) ponds, and was lower in the Tenreuken pond than the Leybeek pond (p=0.0038, Table S7). When data were pooled together, $\%N_2O$ was correlated negatively with water temperature and positively with DIN and $NH_4^+$ (Table S4). In individual ponds, $\%N_2O$ was negatively correlated with water temperature in the Leybeek, Pêcheries, and Tenreuken ponds (Table S5). $\%N_2O$ was positively correlated with $NO_3^-$ in the Leybeek pond and with $NH_4^+$ in the Pêcheries and Tenreuken ponds (Table S8). $\%N_2O$ was positively correlated with Chl-*a* and TSM in the Tenreuken pond, and negatively with Chl-*a* in the Leybeek pond (Table S5), probably reflecting the negative correlation of Chl-*a* and TSM with water temperature in the Tenreuken pond and the positive correlation of Chl-*a* with water temperature in the Leybeek pond (Table S6)."

**#Reviewer 2**

1) Scientific significance

Does the manuscript represent a substantial contribution to scientific progress within the scope of this journal (substantial new concepts, ideas, methods, or data)? Good

2) Scientific quality

Are the scientific approach and applied methods valid? Are the results discussed in an appropriate and balanced way (consideration of related work, including appropriate references)? Fair

3) Presentation quality

Are the scientific results and conclusions presented in a clear, concise, and well structured way (number and quality of figures/tables, appropriate use of English language)? Poor

For final publication, the manuscript should be rejected

Were a revised manuscript to be sent for another round of reviews: I would not be willing to review the revised manuscript.

Suggestions for revision or reasons for rejection (visible to the public if the article is accepted and published)

Line comments:

Line 34: I suggest closing the abstract with a statement on how this work contributes to knowledge on carbon cycling from freshwater/ponds.

Reply: In our opinion, the change of annual emissions with precipitation in response to El Niño is the most important result of our work and we prefer to close the abstract with this. We have added new lines 19-22 of the abstract in the manuscript without track changes the following sentence that shows how this work might contribute to knowledge on carbon cycling from freshwater ponds: "These findings imply that it might be necessary to account for the presence of submerged macrophytes when scaling ebullitive $CH_4$ fluxes in ponds at larger scale (regional or global) (particularly if Chl-*a* is used as a descriptor), although possibly less critical for diffusive $CH_4$, $CO_2$, and $N_2O$ fluxes." and in new lines 23-25: "The temperature sensitivity of ebullitive $CH_4$ fluxes decreased with increasing water depth, implying that shallow sediments would respond more strongly to warming (*e.g.* heat waves)."

Line 61: replace "dynamic" with "dynamics".

Reply: Done

Line 61: Can authors cite this? I don't doubt that this is the case for some ecosystems but I'm not confident that this statement can be easily generalized across all freshwaters/ponds.

Reply: We do not see a problem here. Photosynthesis does affect $CO_2$ content in water and will be balanced by respiration (e.g. Sand-Jensen and Staehr 2007). We did not state that photosynthesis was the only process that controlled $CO_2$ in ponds or that phytoplankton/macrophytes occur in all freshwaters/ponds. However, in the context of the paper (urban ponds in a European city) it is unlikely that ponds are devoid of both phytoplankton and macrophytes. It is equally unlikely that photosynthesis from phytoplankton/macrophytes does not affect at all $CO_2$ dynamics of the studied ponds.

Line 91: I think it's great that authors have added context about ebullition and methane oxidation in this paragraph and the next (starting on line 101). This suggestion may not be necessary, but if authors want to reduce the length of the introduction, I recommend combining and shortening these two paragraphs. I'd keep sentences on line 91-95 and remove the sentence on line 106-107 (this sounds like it belongs in methods) with some rewording to make the content flow better.

Reply: We have removed the section on methane oxidation and methanogenesis pathway from the Introduction that is shorter in the present version.

Line 263: I understand that the authors explored an LMM approach to look at environmental drivers of GHGs and methane processes but that they lost significance between variables they expected to correlate to GHGs (i.e., chl-a as a driver of pCO2) and so maintained their current approach. Still, I firmly believe the current statistical approach is inappropriate for the dataset as it does not account for pseudoreplication/repeated measures. When testing out the LMM approach, did authors incorporate sampling date into the formula? If using the glmmTMB package, this can be done by adding the date as a random effect (e.g., pCO2 ~ chla + nutrients + turbidity + (1|pond) + (1|date) ), or better, by using the temporal autocorrelation function ar1(), which accounts for the similarity in samples collected close in time (e.g., pCO2 ~ chla + nutrients + turbidity + (1|pond) + ar1(date|pond); date and pond are in ar1() so that autoregression is applied to data based on the site). I strongly suggest GLMM or LMM because significance is going to be severely inflated due to repeated

measures (Type I error), which inflates degrees of freedom. Any significance seen using the Pearson approach may be misleading, despite that authors expect them to occur.

Reply: The statistics section has been revised to use GLMMs where appropriate. We performed GLMMs to investigate relationships between data and comparisons between ponds. When the model did not converge, we kept Pearson for the relationships between data, and repeated measures ANOVA which, to a certain extent, considers the nature of the repeated measures over time. The following analyses were carried out, as described in the section on statistics new lines 212-226:

"For the data-sets covering the whole sampling period, for $pCO_2$, dissolved $CH_4$ concentration, $\%N_2O$, bubble flux, $\%CH_4$ in bubbles, and both ebullitive and diffusive $CH_4$ fluxes, generalized linear mixed models (GLMMs) were constructed that included water temperature, rainfall, $\%O_2$, Chl-*a*, TSM, DIN, SRP as fixed effects, and "pond" and "sampling date" as a random effect to account for repeated measurements via the *lme4* package (Bates et al., 2015) in R version 4.4.1 (R Core Team, 2021). When comparing data among the four ponds, "sampling date" was used as a random effect and post-hoc tests were performed using estimated marginal means (*emmeans* package) to assess pairwise differences between ponds.

For comparisons between the four seasons, GLMMs did not converge due to insufficient number of data points. Comparisons on log-transformed data were then made using repeated measures Analysis of variance (ANOVA) with Tukey's honestly significant difference (HSD) post-hoc tests.

The relationships between the annual means of $CH_4$, $CO_2$ and $N_2O$ fluxes and the annual means of a subset of variables (Chl-*a*, macrophyte cover, surface area, depth) were tested with Pearson's linear or quadratic regressions. The modelled bubble fluxes in Silex pond were compared to measured values with Pearson's linear regression.

Statistical significance was set at $p < 0.05$ for all analyses. For comparisons presented on boxplots, different lower-case letters indicate a significant difference between groups."

Line 368: I suggest repeating what the correlations were (positive or negative) to remind readers.

Reply: Done

Figures 3, 7, 11: It might help to simplify the boxplots in these figures by making one box for turbid and one for clear ponds and overlay the points from those sites, with points as a unique shape or color for specific sites. That would leave 2 boxplots per season and would make statistical tests simpler since authors seek to look at differences between clear and turbid sites, not individual sites (gas variable ~ stable state type rather than ~ site). Individual site variation can still be seen in the temporal plots on the right side.

Reply: The data-set of only 2 turbid-water ponds and 2 clear-water ponds is insufficient to test variable ~ type. This was also noted by the Editor in the decision letter. The best we can do is test differences among the 4 ponds (pairwise) and then discuss these differences taking into account that two are clear and the other two are turbid. Consequently, the graphs were kept with the boxplots of the 4 ponds separate rather than grouping them 2 by 2.

Figures 2, 4, 5: These can be supplemental figures.

Reply: We have removed 4 figures from the original submission, so the total number of figures is now 9. We preferred to keep these figures as we think they support the core results and conclusions of our work.

---

## Author Response (AR3)

Thank you very much for this carefully done revision of your manuscript. In my opinion, the manuscript has improved substantially. However, there are quite a few inconsistencies which I ask you to still take care of. Most notably, there are inconsistencies in the statistical approach (correlation/regression, see below) and in what is described in the methods section vs. what is actually presented. I urge you to take care of a description of statistical methods that clearly aligns with what you present as results. I also have a number of more language-related minor issues that should be addressed to improve readability of your manuscript. I consider this a request for minor revision.

**Reply:** We warmly thank the Associate Editor for the additional detailed and thorough suggestions for improvement. Text and figures were changed accordingly.

8: move "to the atmosphere" to sentence end.

**Reply:** Text was modified accordingly.

21-22: second half-sentence misses a verb.

**Reply:** Text was modified accordingly.

22: Unclear meaning of "at seasonal scale". Do you mean "across seasons", i.e. based on a comparison among seasons?

**Reply:** "at seasonal scale" was replaced by "across seasons"

30: The abstract would benefit from a concluding sentence (a potential implication or similar ?).

**Reply:** We added a concluding sentence.

34-36: I suggest to refer to all these GHG emissions as decisively "global".

**Reply:** Text was modified accordingly.

41: "from" rather than "in" smaller water bodies.

**Reply:** Text was modified accordingly.

54: insert "they" after but, otherwise subject missing.

**Reply:** Text was modified accordingly.

67: CH4 "production" rather than "cycling"? "CH4 cycling" is a rather sloppy expression, hard to see a meaning of this phrase.

**Reply:** Text was modified accordingly.

91: move "of CO2 and N2O" to after "emissions".

**Reply:** Text was modified accordingly.

107-110: Please rephrase this sentence. It is a rather important one, but has weird syntax. Use "assess" rather than "test" because you do not present a test for differences due to alternative state. Let "in terms of..." follow "differences" immediately.

**Reply:** Text was modified accordingly.

113-115: Rephrase, first site, then time.

**Reply:** Text was modified accordingly.

119: insert "the" before laboratory.

**Reply:** Text was modified accordingly.

121: "of" has wrong font size.

**Reply:** Text was modified accordingly.

141-143: I cannot follow the logic of this sentence. What is the connection between park closure an time series length? May not be very important, but should be understandable.

**Reply:** We agree the explanation was not important, so sentence was shortened.

164-165: Statement is not exact. You did not use the headspace technique for pure gas samples. Reword.

**Reply:** Text was modified accordingly.

183-184: Text repeat, compare lines 173-176. Shorten.

**Reply:** Text was modified accordingly.

195-202: It is unclear what the point of these computations is. Was this done for Silex only? Shouldn´t this be part of the statistics section?

**Reply:** This section describes the model used in Figs. 5 and S4. We think that the readers that want to refer to the model are likely to look in section 2.3.2 rather than section 2.4, so we prefer to keep the text in this place rather than moving it to the statistics section.

211: Make clear here how you use this equation to predict whole time series of ebullition for each pond.

**Reply:** We added L218 the following sentence: "Equation (8) is used to predict $E_{CH4}$ in each pond from time-series of water temperature allowing matching each diffusive $CH_4$ estimate derived from Equation (1)."

212-226: The statistics section is still pretty unclear and hard to follow despite its brevity. GLMMs were used for two different purposes and this cannot be recognized from the text here: (1) to relate GHG variables to their controls across all ponds, (2) to find differences among ponds. Also, in line 219-221, one believes formal comparisons among seaons were done, but this was not the case. Rather, data was compared among ponds but separated by seasons - this needs rewording. I would welcome a try to improve consistency in the statistical approach by NOT using two different methods (GLMMs and ANOVAs) to test for differences among ponds for entire time-series and separated by seasons (this just confuses the reader, I cannot see a benefit in using two different methods). In lines 222-224 you introduce "Pearson´s ... regressions" and thereby start with a confusion of correlation and regression as exchangeable or identical (?) techniques, ignoring their fundamental conceptual difference. This issue of mixing up correlation with regression is prevalent throughout the whole results section and in various figures. I think, in fact, that you almost nowhere use a correlation and thus should not use that word. Pearson may be rightfully considered to be the father of the "Pearson´s correlation coefficient" but I do not know a single statistical text-book that would describe a "Pearson´s regression". All models used are regressions and the text should reflect this (I note, however, that in some cases, see below, a correlation may indeed be more appropriate). For the statement in lines 223-224, I cannot see an objective (Is this part of the GLMM analysis? Was it done only for Silex?).

**Reply**: In the present version of the manuscript, we no longer refer to correlation, throughout the text. We have modified and expanded the Statics section in the Methods to accommodate all of the editor's suggestions and comments. The use of the GLMMs was requested by reviewer 2 during the 2 previous rounds of review to account for the nature of the repeated measures in each system (pseudo-replication). However, when testing differences separating the data by seasons the GLMM did not converge due to the low number of data. This then required using a different test, reverting to the ANOVAs. This is explained clearly in the Methods, so we do not anticipate confusion from the readers.

243: Second half-sentence misses a verb.

**Reply**: Text was modified accordingly.

254: competition "with" macrophytes?

**Reply:** Text was modified accordingly.

266: Reconsider usage of "together" in context with "pooled", can just be dropped. Applies to multiple locations in the manuscript.

**Reply:** Text was modified accordingly, here and elsewhere in text.

267: The inclusion of precipitation (or rainfall in line 215) becomes clear later on, but maybe explain your objective behind including this variable as a predictor at an

earlier point. In fact, it would be nice to have a short justification for the variables considered as predictors in the statistics section.

**Reply**: A short justification of the different tests was added to the statistics section.

275-282 and 292-299: These sections are examples for the confusion of correlation with regression. The word "correlated" appears multiple times but is entirely inappropriate as you always refer to results of regression analysis. This also applies to other text sections, reconsider phrasing using "correlated with" everywhere.

**Reply**: In the present version of the manuscript, we no longer refer to correlation throughout the text.

304: This is a rare example where a correlation is actually more appropriate than a regression. I cannot see why CH4 content of bubbles should depend on bubble flux, especially when CH4 makes up the bulk of the gas in those bubbles. Here, a correlation would be very appropriate, but you use a regression. Reconsider wording but also analysis and figure S3.

**Reply**: The figure and figure legend were updated accordingly.

306: Reword "during more lengthy series" - weird formulation.

**Reply**: Text was modified accordingly.

308: Reword "during events", could just be "following drops...".

**Reply**: Text was modified accordingly.

310: role for what?

**Reply**: Text was modified for clarification.

313-318: Objective behind doing this is unclear to me. Should be part of methods.

**Reply**: Text was modified for clarification.

320: The phrase "positively related to" may serv as an example of proper description of a regression result without using the phrase "correlated with".

**Reply**: In the present version of the manuscript, we no longer refer to correlation throughout the text.

330: Rephrase to "dependency of ...concentration on water temperature". This touches the issue of correlation vs. regression. You mostly use regression techniques, these imply a direction of dependence, thus the phrasing as "dependency between" is not appropriate.

**Reply**: Text was modified accordingly.

332-339: I think this should be described in more detail in the methods section as well. Here, in the results, please acknowledge differences in the temperature ranges between fitted model and usage of the model. For some ponds, you actually extrapolate (!) outside the temperature range of data used for fitting the model. This should be recognizable to the reader as it entails increased chance for bias.

**Reply:** We added a sentence in the Methods section to explain that the ebullitive CH4 fluxes were modelled from water temperature. We added the following sentence "Note that the relations of ebullitive $CH_4$ fluxes as a function of water temperature were established over a temperature range (7.0 to 26.3°C) that is consistent with the range of water temperature values (2.0-25.9°C) over which the ebullitive $CH_4$ fluxes were modelled." in the legend of Figure 7 to address the issue of temperature bounds of the extrapolation.

401: don´t use word "test" as you never do this.

**Reply:** we replaced the word « test » by investigate

414: Sentences misses a subject.

**Reply**: Text was modified for clarification.

422: again missing subject in "when pooled all data together".

**Reply**: Text was modified for clarification.

432: Statements with statistics in brackets may better fit into the results section or at least statistics should be reported there.

**Reply**: Stats were removed from text as they were mentioned in the Results section.

435: This is just a repetition of a result, explanation/interpretation missing here.

**Reply**: Text was shortened accordingly.

443: "similar", not "equivalent".

**Reply**: Text was modified accordingly.

462-463: Reword "of a mitigation....fluxes", weird formulation.

**Reply**: Text was modified accordingly for clarification.

479: Reword "scaled at globally scale".

**Reply**: Word "scaled" was replaced by extrapolated.

483-492: All this only applies if productivity is high, for sure not in oligotrophic clear-water ponds.

**Reply**: We agree, and statements refer to 4 ponds in question, which we think is clear from text, and context of this section.

503: I don´t understand the logic behind the dilution argument here. Flux out of the system represents a rate of production, especially at longer time scale. The volume of water should not make a difference. Reword or explain.

**Reply**: The statement was removed.

506: Match what?

**Reply**: Text was modified accordingly for clarification.

509: Reword "augmentation of".

**Reply**: Text was modified accordingly.

514-523: Plural precipitations seems inappropriate.

**Reply**: Text was modified accordingly.

521: Reconsider sentence syntax.

**Reply**: Text was modified accordingly for clarification.

525-526: Use comma after "clear-water" and "turbid-water".

**Reply**: Text was modified accordingly.

531-532: Rephrase, very awkward wording.

**Reply**: Text was modified accordingly for clarification.

536: Hard to follow phrasing. Can variations be dominated by variations?

**Reply**: Text was modified accordingly for clarification.

539-540: Too simplified. Inorganic carbon contributing to CO2 emission is hardly a result of surface runoff.

**Reply**: Text was modified accordingly for clarification.

Fig 2: why use a regression here? A correlation would be more appropriate. Both the regression line and equation should be deleted.

**Reply**: Figure 2 was updated accordingly to show the linear correlation

561: White and grey need to be Grey and white, I guess. Reorder.

**Reply**: Text was modified accordingly.

Fig 4: This figure left me entirely confused. I cannot find the respective details behind what is presented here. Are these figures representing results for what is described in the methods at all? Equation (3), but then where is the effect of delta_P? Or are these results for application of equation (7) being applied to different variables? Where can I find the methods for the equations presented in the figure legend? These do not look like GLMMs at all. The first figure lets me recognize strong differences among ponds, but these are not represented by the single regression model. Serious clean-up needed here.

**Reply**: We modified the figure to provide an exponential fit as done in other studies for the ebullitive flux.

Figure 5: Is this linked to fig 4 or not?

**Reply:** Figure 5 shows the time-series of bubble flux and temperature in the Silex pond, while figure 4 shows the bubble flux as a function of temperature in all four ponds.

Fig 7: Again reorder colors "white" and "grey" in the figure legend.

**Reply**: Text was modified accordingly.

Fig. 9: Word "evolution" may be reconsidered. The journal is also read by biologists.

**Reply**: Text was modified accordingly.

Table S3: Make clear which base was used for log-transformation, so that original data may be recomputed from values in the table.

**Reply**: Text was modified to indicate it's a $\log_{10}$ tranformation.

Table S6: Details for what is presented here are missing from the statistical methods section.

**Reply**: We updated accordingly the text of the Statistics Methods section.

Table S7: Log-transformation? Base?

**Reply**: Text was modified to indicate it's a $\log_{10}$ transformation.

Table S9: A peculiar function is used to fit the ratio of ebullitive to total fluxes here. Why this function? Whole approach is missing from the methods description. Also, does this model agree with what is presented in Figure S5?

**Reply**: We updated accordingly the text of the Statistics Methods section. We also refer to Fig. S9 in the legend of Table S9. The model in the Table corresponds to the fit of the data in the Figure.

Fig S1: Why is the pressure drop factor not computed continuously, i.e. for a moving time window?

**Reply**: We followed to procedure given by Zhao et al. (2017). The text of the legend of the figure was updated accordingly.

Fig S2: Do not use plural precipations.

**Reply**: Figure S2 was updated accordingly.

Fig S3: Why use a regression here? What is the logic behind this approach?

**Reply**: Here, we used a correlation, and figure and corresponding legend were updated accordingly.

Fig S4: Why separate the 2 temperature ranges? No logic for this approach presented in the methods. The graph seems very inefficient data presentation, we see the same data clouds two times. In fact the upper four panels could be deleted.

**Reply**: We updated the Methods section to explain the logic of using three temperature ranges (<,>15, and full). We agree that there is some redundancy in the data presentation, although this might not be as critical for a supplemental figure than for a figure in the main article. However, it allows to present the goodness of fit for each temperature range in a very clear way. The goodness of fit for each of the three temperature ranges with and without the pressure effect is important in the data presentation and discussion (refer to corresponding text in Results and Discussion). We could have made a slightly more compact figure but very crowded with the stats, making the figure much more difficult to read. We decided to favor clarity over compactness, in particular given this is a Supplemental figure.

Also, no description/justification for the transformation log(X+1) is presentend anywhere. Why +1? The log(X+1) transformation may make sense for count data, but that is not the case here.

**Reply**: We added to the Figure legend the following sentence: "To account for zero values, log10 was computed on values plus 1."

---

## Author Response (AR4)

Thank you for preparing this revision, which I will recommend for publication. I am happy to see this treasure of data to be published!

There are a few minor technical corrections which I ask you to take care of. Note that I refer to line numbers in the pdf with tracked changes.

**Reply:** We very warmly thank the Associate Editor for the additional suggestions for improvement. Text was changed accordingly.

218: Delete statement about observed clustering of points, seems like a Result.

**Reply:** The statement was deleted accordingly.

223-235: The order of statements and equations seems unfortunate here. Please rearrange to allow your reader faster understanding: equ 7 to after equ 8 and after the statement about how equ 8 was used for prediction.

**Reply:** The sequence of equations and related equations were rearranged accordingly.

237-248: I suggest to make a clearer distinction between objectives (1) and (2). First describe comparing ponds using factors, second explain the regression approach with continuous predictors. Separate paragraphs would help. Currently, you write about objectives one a bit like 1-2-1-2-1.

**Reply:** Text was re-arranged accordingly.

Also, lines 245-248 should be reworded, the syntax is hard to understand, better write "the impact of X on Y" and not "the impact on X of Y".

**Reply:** This sentence was reworded accordingly, and now reads L 244 : "This analysis aimed at investigating (1) the impact of photosynthesis-respiration on $CO_2$ concentrations and emissions based the relationships with Chl-*a*, DIN, SRP; (2) the impact of the response of methanogenesis to warming on $CH_4$ concentrations and diffusive and ebullitive emissions based on the relationships to $T_W$; (3) the impact on DIN availability and $T_W$ on $N_2O$ concentrations and diffusive emissions."

344-346: You did not describe analysis of CH4-content anywhere in the data analysis of Methods section. Please add these missing information to Methods. Also applies to Fig 4.

**Reply:** This information was added to the Methods section, text now reads L 212: "The $CH_4$ content in bubbles expressed in % of total gas (%$CH_4$) was fitted with a linear regression model dependent on $T_W$ in the Silex pond. The correlation of %$CH_4$ with $F_{bubble}$ was tested on the merged data of all the four ponds."

Fig 2: I see no need for a regression line and model to be presented here and it makes statistically little sense. Correlation is ok. If you are convinced about using regression, then I would argue it´s rather temperature influencing precipitation.

**Reply:** We have removed the linear regression line from the plot. The Pearson *r* is mentioned in the figure legend and appears on the plot. The same changes were applied to Fig. S3.